# Ecology and spread of the North American H5N1 epizootic

Lambodhar Damodaran[1], Anna S. Jaeger[1] & Louise H. Moncla[1✉]

Since late 2021, a panzootic of highly pathogenic H5N1 has devastated wild birds, agriculture and mammals. Here an analysis of 1,818 haemagglutinin sequences from wild birds, domestic birds and mammals reveals that the North American panzootic was driven by around nine introductions into the Atlantic and Pacific flyways, followed by rapid dissemination through wild, migratory birds. Transmission was primarily driven by Anseriformes, while non-canonical species acted as dead-end hosts. In contrast to the epizootic of 2015 (refs. 1,2), outbreaks in domestic birds were driven by around 46–113 independent introductions from wild birds that persisted for up to 6 months. Backyard birds were infected around 9 days earlier on average than commercial poultry, suggesting potential as early-warning signals for transmission upticks. We pinpoint wild birds as critical drivers of the epizootic, implying that enhanced surveillance in wild birds and strategies that reduce transmission at the wild–agriculture interface will be key for future tracking and outbreak prevention.

Highly pathogenic avian influenza (HPAI) viruses pose persistent challenges for human and animal health. Since emerging in 1996, highly pathogenic H5N1 viruses of the A/goose/Guangdong lineage have spread globally through enzootic transmission in domestic poultry in Asia and Africa, paired with occasional cross-continental movement by wild birds of the Anseriformes (ducks, geese, swans) and Charadriiformes (shorebirds) orders[3–9]. In 2005, introduction of poultry-derived H5N1 viruses into wild birds in China led to viral dispersal across Northern Africa and Asia, establishing new lineages of endemic circulation in poultry[10,11]. In 2014, wild migratory birds carried highly pathogenic H5N8 viruses from Europe to North America, sparking an outbreak in which over 50.5 million commercial birds were culled in the USA[4,12]. As these viruses did not establish persistently within wild birds, the outbreak was extinguished by aggressive culling, and North America remained free of HPAI for years.

In December 2021, clade 2.3.4.4b HPAI H5N1 viruses were introduced and spread across the Americas[13–15], causing a panzootic of considerable morbidity and mortality in wild and domestic animals. In contrast to past North American epizootics, domestic bird culling has not halted detections, and morbidity and mortality has been widespread across wild avian and mammal species not usually impacted by HPAI[15–20], raising the possibility that new reservoir hosts could be established that should be actively surveilled. In Europe, clade 2.3.4.4b virus incursions into wild and domestic birds has led to seasonal outbreaks[21], frequent reassortment[22] and a broader range of affected wild bird species since 2020, and recent analyses suggest that wild birds may now have a greater role in global viral maintenance and dissemination[8,23]. In North America, the broad affected host range and continued agricultural outbreaks suggest that patterns of transmission since 2022 may be distinct from past epizootics. However, the role of wild versus domestic birds in driving transmission in North America has not been robustly or comprehensively studied, limiting informed surveillance and outbreak control.

Viral phylodynamic approaches are emerging as critical tools for outbreak reconstruction. We used Bayesian phylogeographical approaches to trace the introduction and spread of highly pathogenic H5N1 viruses during the first 18 months in North America. We identify multiple incursions into the continent and subsequent spread by wild, migrating birds that drove repeated introductions into agriculture. These data pinpoint wild birds as important drivers of epizootic spread, and implicate enhanced wildlife surveillance and interventions at the wild–domestic interface as key for future viral tracking and spillover prevention.

## Sequences reflect HPAI cases over time

The first detection of HPAI H5N1 in North America was reported in migratory gulls in Newfoundland and Labrador Canada in November 2021 (ref. 13). From January to May 2022, a total of 2,510 total detections was reported across 43 US states and 91 species (Extended Data Fig. 1), followed by a larger epizootic wave from August 2022 to March 2023 (8,001 detections, 48 contiguous US states and Alaska). During the time period analysed (November 2021 to September 2023), most US detections were reported in wild birds (Supplementary Fig. 1a). Case detections peaked in the fall and spring, coinciding roughly with seasonal migration timing for birds migrating between North and South America[24,25]. Continued monitoring is necessary to determine whether these patterns persist in future years.

Although sequencing data from North America are heavily skewed towards the USA and the first 6 months of the outbreak (Supplementary Fig. 2), case detections were modestly correlated with viral effective population size ($N_e$) (highest Spearman rank correlation = 0.65, $P = 4.4 \times 10^{-11}$) (Extended Data Fig. 1c and Supplementary Figs. 3 and 4), a measure of genetic diversity mathematically related to disease transmission and prevalence[26]. Peaks in $N_e$ preceded peaks in detections by around 1 week (Supplementary Fig. 5), probably reflecting the lag

[1]Department of Pathobiology, School of Veterinary Medicine, University of Pennsylvania, Philadelphia, PA, USA. ✉e-mail: lhmoncla@upenn.edu

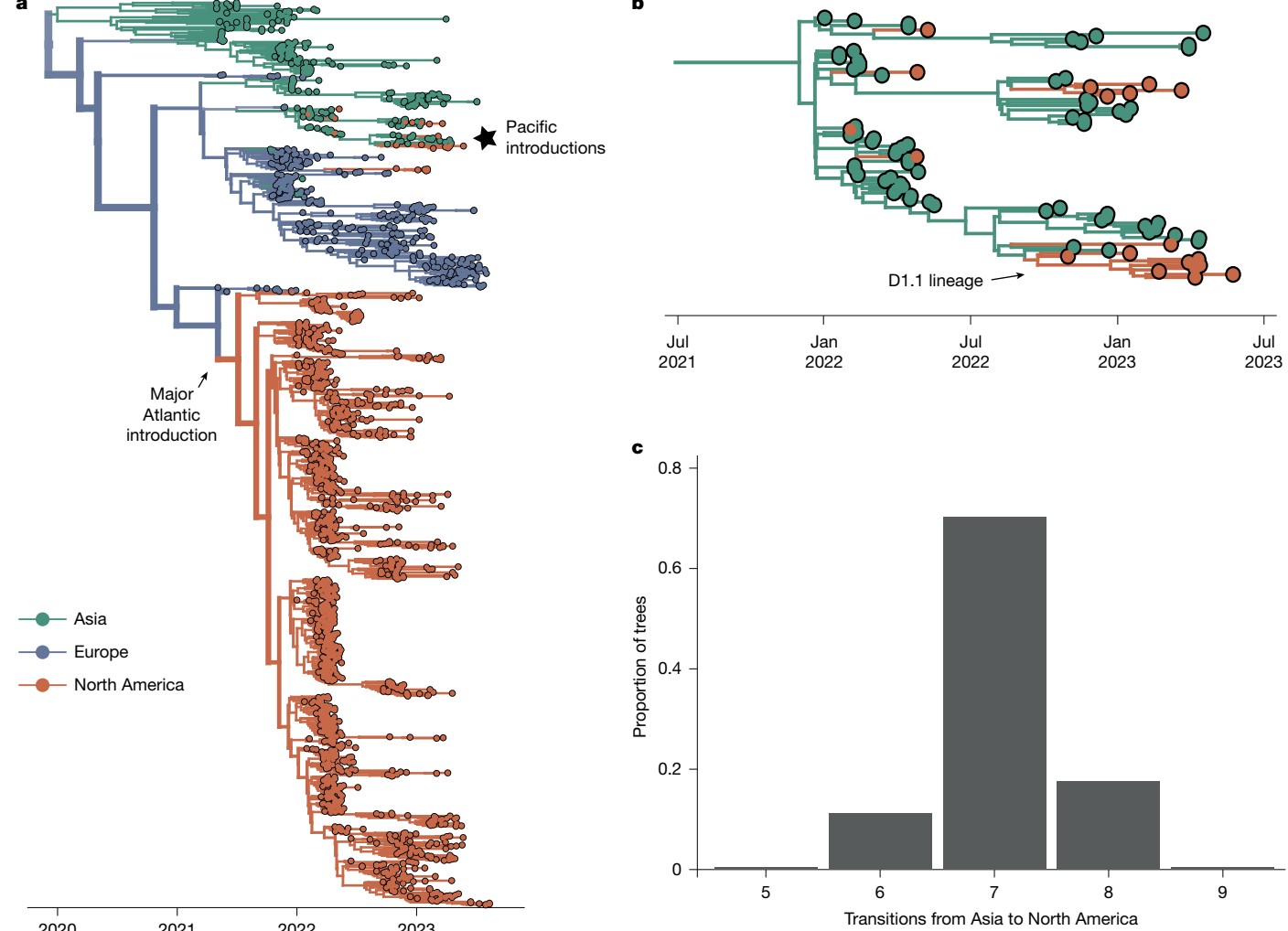

**Fig. 1 | H5N1 viruses were introduced repeatedly from Europe and Asia.**
**a**, Bayesian phylogenetic reconstruction of *n* = 1,927 globally sampled sequences of HPAI clade 2.3.4.4b coloured according to the continent of isolation. The opacity of branches corresponds to posterior support for the discrete trait inferred for a given branch, and the thickness corresponds to the number of descendent tips the given branch produces. The major Atlantic introduction is annotated. **b**, A magnified view of the starred section of the tree in **a**, focusing on introductions from Asia. The introduction that resulted in the D1.1 lineage is also noted. **c**, We inferred the number of transitions from Asia to North America across the posterior set of 9,000 trees. The *x* axis represents the number of introductions, and the *y* axis represents the proportion of trees across the posterior set with that number of inferred transitions.

between viral transmission and case detection. Thus, despite uneven sampling, sequence diversity roughly reflects the amplitude of sampled cases over time.

## Repeated incursions drove the epizootic

Most North American sequences descend from a single introduction from Europe in late 2021 (95% highest posterior density (HPD), 9 September to 7 October 2021; Fig. 1a), consistent with previous reports[13–15] that these viruses may have been introduced as early as 1 to 2 months before the first detection. We recapitulate a second, short-lived introduction from Europe in 2022 (ref. 27), and seven additional (median = 7, 95% HPD = 6–8) introductions between February and September 2022 from Asia (Fig. 1b,c). These introductions persisted briefly (0.024–6.9 months) and represent infections sampled in Alaska, Oregon, California, Wyoming and British Columbia, suggesting introduction through the Pacific flyway[28] (Extended Data Fig. 2). Although none of these Pacific introductions had sampled descendants in the time period analysed, data at the time of writing indicate that one re-emerged in late 2024 as the D1.1 lineage[29] (Fig. 1b). Although it remains unclear why this HA lineage was not detected from

mid-2023 to 2024, the novel introductions documented here and the eventual outgrowth of one of these lineages highlight the importance of surveillance in the Pacific region for capturing viral importations. These data suggest that H5N1 viruses were introduced into North America at least nine times, and that viral flow into the Pacific coast may be far more common than previously documented.

## H5N1 spread across migratory flyways

Recent data from Europe and Asia suggest that wild birds may be increasingly important sources of clade 2.3.4.4b virus evolution and transmission[8]. In the Americas, wild birds migrate across four major flyways: the Atlantic, Mississippi, Central and Pacific[30]. We assigned avian sequences to the migratory flyway matching the US state of sampling and modelled the diffusion between flyways as a proxy for viral movement. To determine whether sequences clustered more strongly by flyway than expected by chance, we calculated the association index (AI)—a measure of how strongly a trait is associated with a phylogenetic tree[31]. To determine whether movement between flyways was better supported than movement across other adjacent geographical regions, we quantified transitions between four North American regions stratified by latitude.

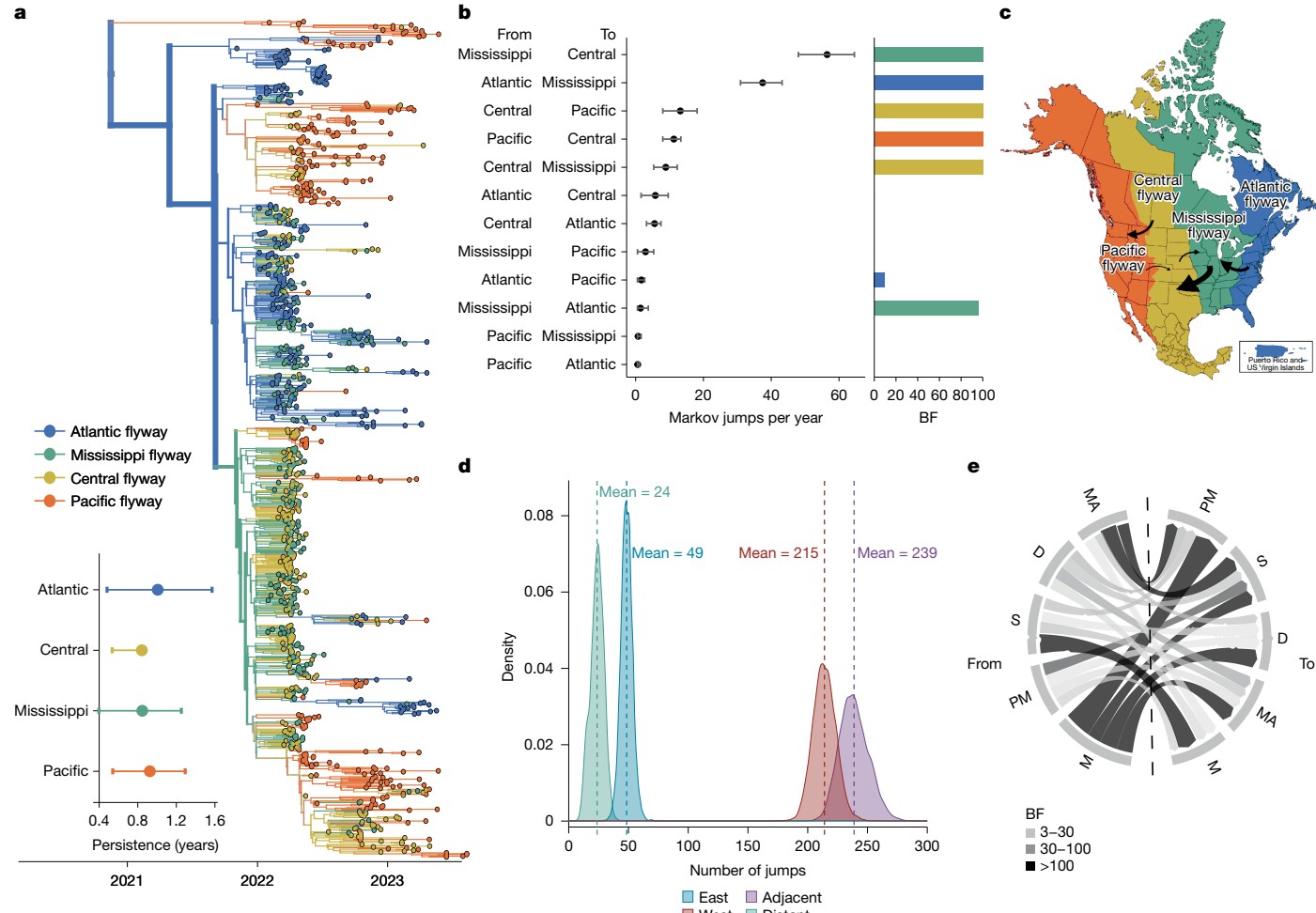

**Fig. 2 | Wild migratory birds drove rapid dissemination across continental migratory flyways. a**, Phylogenetic reconstruction of *n* = 1,000 sequences coloured by migratory flyway. Inset: the results of the PACT analysis quantifying persistence in each flyway (measured as the length of time a tip takes to leave its sampled location, going backwards on the tree), excluding the Pacific clade. **b**, The mean and 95% HPD for the number of Markov jumps per year between US Fish and Wildlife Service (USFWS) flyways. The colour of the bar on the right of each jump pair corresponds to the source population and the height of the bar corresponds to the BF support. **c**, USFWS waterfowl flyways map; arrows are annotated to represent rates with BF support of at least 100. The size of the arrow corresponds to the magnitude of the mean transition rate. **d**, The posterior distribution of the number of Markov jumps between flyways in the eastward or westward direction and between adjacent and distant flyways. **e**, Chord diagram of discrete trait diffusion based on migratory status going from the source population on the left to the sink population on the right. The chord thickness represents the mean transition rate and the colour represents the BF support. D, domestic; M, migratory; MA, mammal; PM, partially migratory; S, sedentary.

Introductions from viruses circulating in Asia (Fig. 1b) form a basal clade inferred in the Pacific flyway (posterior probability (PP) = 0.98). The primary introduction from Europe entered through the Atlantic flyway, and subsequently spread rapidly across North America (Fig. 2a,c). From the inferred time of introduction in the Atlantic coast (9 September to 7 October 2021), viruses descending from this introduction had been sampled in every other flyway within approximately 4.8 months. Sequences clustered strongly by flyway (AI = 10.563, *P* = 0.00199), grouping most closely with those sampled within the same or geographically adjacent flyway (Fig. 2a and Extended Data Table 1). Transitions (inferred as Markov jumps) between adjacent flyways were about 10 times more frequent (mean = 239, 95% HPD = 216–262) than those between distant flyways (mean = 24, 95% HPD = 12–33; Fig. 2d), and 2.8 times more frequent between adjacent latitudinal regions (Extended Data Fig. 4 and Supplementary Table 2), indicating a strong signal of dissemination through geographical proximity. Transitions were predominantly inferred from east to west (Fig. 2c,d and Supplementary Table 1); east to west jumps were inferred around 4.4 times more frequently (mean = 214, 95% HPD = 196–232) than west to east jumps (mean = 49, 95% HPD = 38–57) (Fig. 2d), and 2.3–3.8 times more

frequently than jumps along the north–south axis (Extended Data Fig. 4 and Supplementary Table 2).

Transitions were inferred most frequently from the Mississippi to Central flyway (56.301 Markov jumps per year; 95% HPD = 47.85–64.33), Atlantic to Mississippi flyway (37.34 Markov jumps per year; 95% HPD = 30.84–43.065) and Central to Pacific flyway (13.127 Markov jumps per year; 95% HPD = 7.975–18.077; Fig. 2b, Extended Data Fig. 3 and Supplementary Table 1). Although the Pacific flyway experienced the highest number of introductions, transitions originating from the Pacific flyway were inferred with low magnitude and weak support, with only one statistically supported rate (Pacific to Central, 11.236 Markov jumps per year; 95% HPD = 7.975–13.292). Viral lineages persisted for the longest in the Atlantic and Pacific flyways, although estimates were variable (Fig. 2a). We speculate that this pattern could reflect higher habitat and species richness within coastal flyways[32], or that coastal flyways each only border 1 other flyway.

The strong clustering by flyways is consistent with long-range transmission by wild migratory birds. We next classified sequences into five categories and modelled diffusion among them: wild migratory birds (most ducks and geese), wild partially migratory birds (some ducks,

raptors and vultures), wild sedentary birds (owls crows), domestic birds and non-human mammals. Migratory and partially migratory wild birds are inferred at the root far more frequently than expected from sampling alone (Supplementary Fig. 6 and Extended Data Table 2), indicating a role for these species in sustained transmission across the epizootic. Transitions from wild migratory birds were inferred with the highest number and most strongly supported transition rates (Bayes factor (BF) > 3,000), indicating that migrating wild birds were critical sources of infections in other species (Fig. 2e and Supplementary Table 3). By contrast, transitions from non-migratory wild birds were inferred with low magnitudes and weak support (Fig. 2e and Supplementary Table 3). These results suggest that wild, migratory birds played a pivotal part in transmission, and highlight their capacity to rapidly disseminate novel viral incursions across continental North America.

## Transmission driven by canonical hosts

Previous outbreaks of highly pathogenic H5N1 viruses have been facilitated by wild Anseriformes (waterfowl) and Charadriiformes (shorebirds), and domestic species (Galliformes and Anseriformes)[1,33–36]. While domestic ducks have been critical for bridging wild and domestic populations in Asia, domestic ducks account for only 2% of all detections in the USA, with most cases reported in wild birds and Gallinaceous poultry (turkeys and chickens)[37]. In the current panzootic, die-offs have occurred across a range of wild, non-canonical hosts, including Accipitriformes (raptors, condors, vultures), Strigiformes (owls) and Passeriformes (including sparrows, crows, robins)[15,19,20], raising the possibility that these new species could establish as reservoirs that merit surveillance. To determine whether particular host groups had outsized roles in driving transmission in the epizootic, we classified sequences into seven host order groups (Anseriformes, shorebirds, Strigiformes, Passeriformes, Raptors, Galliformes and non-human mammals), calculated the AI for each group (Extended Data Table 1) and modelled transmission between them. To control for variation in case and sequence acquisition across groups, we performed these analyses under two subsampling regimes (proportional and equal), each with three replicates and report results that were concordant. We also formulated a modified tip-shuffle test to measure the impact of sampling on the inferred host at the root[38] (further details are provided in the Methods).

The first introduction into North America comprised infections from gulls and harbour seals from New England, consistent with migratory shorebirds facilitating transmission from Europe and seeding mammal outbreaks[19,39] (Fig. 3a). Tip-shuffle results indicate mixed evidence for the role of shorebirds in transmission. However, shorebird sequences were highly clustered with each other (AI = 8.008, null = 2.324, $P$ = 0.00999), supporting some degree of separation between viruses circulating in shorebirds and other species[3]. Beyond this early cluster of infections, multiple deep, internal nodes across the phylogeny are inferred in Anseriformes with high posterior support (PP = 0.99), indicating that Anseriformes played an important role in driving sustained transmission and dispersal across North America. Across all replicates in both sampling regimes, Anseriformes are inferred at the root 2–3 times more frequently than in null, shuffled datasets (Extended Data Table 2), providing strong support for Anseriformes as critical drivers of epizootic transmission. We infer Anseriformes as the predominant hosts seeding infections into other species (Fig. 3b,d, Supplementary Fig. 7 and Supplementary Tables 4–11), with the highest rates to Galliformes (17.81 Markov jumps per year; 95% HPD = 9.27–26.02, BF = 1,691, PP = 0.99) and Strigiformes (13.51 Markov jumps per year; 95% HPD = 5.35–22.87, BF = 232, PP = 0.99). Each of these patterns was preserved in each independent subsample in both sampling regimes, indicating high robustness to sampling (Supplementary Figs. 8 and 9).

We also infer support for transmission originating from Galliformes, suggesting that transmission from domestic birds back to wild birds and mammals may have occurred. However, lineages in Galliformes tended to be short-lived, persisting for 0.26 years on average (95% HPD = 0.07–0.33 years). Galliformes were inferred at the root less frequently than expected for their sampling frequency (Extended Data Table 2), and were highly clustered ($P$ = 0.0099; Extended Data Table 1), consistent with transmission confined to localized agricultural outbreaks. By contrast, viral lineages persisted for the longest in Anseriformes and shorebirds (Fig. 3c). These data suggest that, while Anseriformes, shorebirds and Galliformes may all have contributed to infections in other species, Anseriformes were the predominant drivers of longer-term persistence and spread to other hosts.

In the ongoing panzootic, raptors represent the third most prevalent group in wild bird detections in Europe (12% of detections) and the second most detected group in North America (20.3%)[18,40]. Notably, raptors were inferred as a low-frequency but statistically well supported source population to Anseriformes (5.18 Markov jumps per year; 95% HPD = 0.36–9.27, BF = 39, PP = 0.87). Tip-shuffle results indicate that raptors are less probable at the root than expected based on their frequency, supporting a limited role for epizootic transmission. Future work to better establish the reasons for high case numbers among raptors will be necessary for formulating wildlife management strategies.

We found limited support for non-canonical host groups (songbirds, owls and non-human mammals) in seeding infections in other species. Passeriformes (songbirds), Strigiformes (owls) and mammals each primarily served as sinks for viral diversity (Fig. 3b,d), with transitions inferred with low-magnitude and weak support (Fig. 3b). Summing the number of jumps originating from wild canonical (Anseriformes, shorebirds), wild non-canonical (Passeriformes, Strigiformes, raptors, mammals) and Galliforme (domestic) hosts confirm that non-canonical hosts primarily acted sinks that were far likelier to receive virus than propagate it onward (Extended Data Fig. 5), supporting short, terminal transmission chains that did not lead to long-term persistence (Fig. 3c and Supplementary Fig. 10). Mammal sequences cluster across the entire diversity of the phylogeny (Fig. 3a) and are not associated with one particular cluster of viruses, indicating that mammal infections were not confined to a particular viral lineage, supporting very short persistence times of 0.22 years (95% HPD = 0.088–0.328), and only one strongly supported transition rate to Anseriformes (BF = 53, PP = 0.89). Instead, these findings are most compatible with a model in which wild mammals and other non-canonical species are infected by direct interaction with wild birds, possibly related to scavenging and predation behaviour[41]. Taken together, these data suggest that despite high case numbers in several unusual wild hosts, non-canonical species generally had minor roles in transmission. Instead, epizootic transmission was most strongly supported in Anseriformes, supporting surveillance in these species for capturing trends in viral diversity and spread.

## Repeated introductions into agriculture

From 2022 to mid-2025, the USA culled over 160 million domestic birds, with agricultural losses estimated between US$2.5 to US$3 billion[42]. Understanding the extent of agricultural transmission driven by repeated introductions from wild birds versus between-premise spread is critical for formulating biosecurity practices, but challenged by differences in sampling between wild and domestic birds. Domestic birds represent 23.2% of sequences, but only 11% of detections, while wild birds are probably undersampled owing to technical challenges[20,43]. While each detection in wild birds represents a single infection, domestic detections usually represent a single infected farm, with an unknown number of infected animals. To measure the impact of varied sampling on transmission inference between wild and domestic birds, we designed a titration analysis. We first generated a dataset with equal domestic and wild bird sequences, therefore forcing the

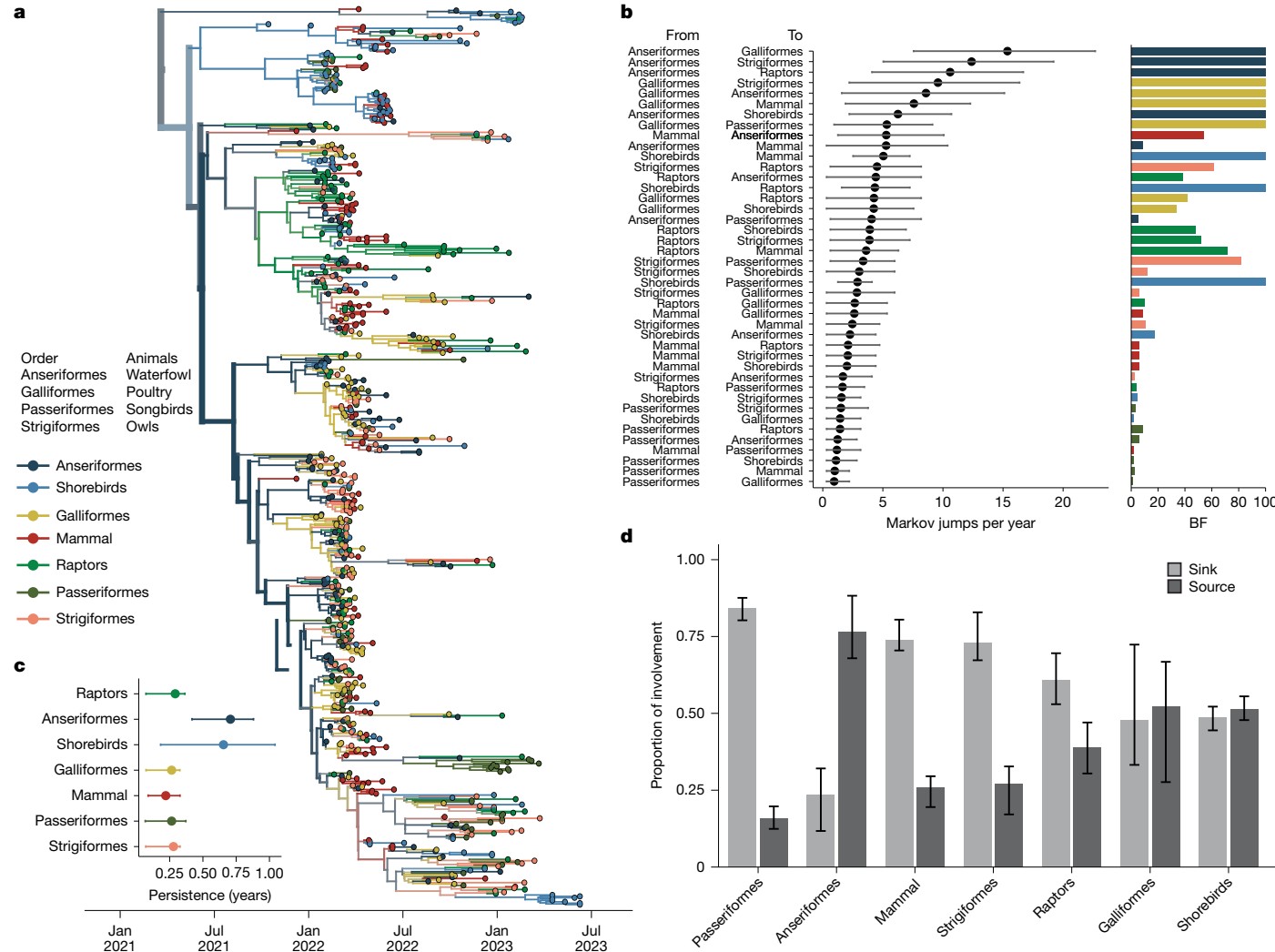

**Fig. 3 | Epizootic transmission was sustained by canonical host species. a**, Bayesian phylogenetic reconstruction of 655 sequences, sampled evenly across host groups. The phylogeny with the highest posterior support is shown; all other replicates are provided in Supplementary Fig. 8 and Supplementary Tables 4–11. Colour represents the taxonomic order of the source host. **b**, The mean number of Markov jumps per year and the 95% HPD from the host group on the left (labelled 'from') to the host on the right (labelled 'to') as inferred from the combined results of three equal sampling replicates. The dot represents the mean, and the lines (whiskers) represent the 95% HPD. The corresponding bar plot shows the BF support for each jump pair, with colour representing the 'from' host. The bar height represents BF support. Values at 100 indicate support of greater than or equal to 100. **c**, Inference of phylogenetic persistence in each host order for the phylogeny shown in **a**. **d**, For each host, we computed the proportion of Markov jumps involving that host order in which that host was inferred as a source (jump coming from that order) or as a sink (jump going to that order). The bars represent the variability across the three replicates of equal orders subsamples.

inference to be driven by the sequencing data rather than sampling. Next, we added in progressively more wild bird sequences until we reached a final ratio of domestic to wild sequences that approximates the ratio of detections (1:3), generating five datasets in total (ratios of domestic to wild bird sequences of 1:1, 1:1.5, 1:2, 1:2.5 and 1:3). For each dataset, we inferred transmission between wild and domestic birds using a discrete trait diffusion model. This analysis was designed to determine whether domestic or wild birds would be inferred as the primary source population, and whether that inference would vary across sampling regimes. Moreover, we hoped to assess whether the inferred number of transitions between hosts stabilized at a certain ratio as a measure of whether currently available data are sufficient for inferring transmission dynamics within this time period.

When domestic/wild sequences were included in equal proportions, wild birds are inferred as the primary source in the outbreak (Supplementary Fig. 11a). Wild birds were inferred at the root of the tree at a far higher probability than expected from their sampling (PP = 0.895 in empirical data versus 0.482 in tip-shuffled data), while domestic

birds were under-represented (Extended Data Table 2). This pattern is consistent with higher genetic diversity among wild bird sequences, supporting a large, source population. Within the background of wild bird sequences, domestic bird sequences form highly clustered groups (AI = 23.096, *P* = 0.0019; Extended Data Table 1), consistent with some transmission between them. However, as wild sequences were progressively added into the tree, most domestic-only clusters became smaller, broken up by wild sequences that interspersed within these clades (Supplementary Fig. 11a–e). The 'breaking up' of these domestic clusters results in more inferred transitions from wild to domestic birds, and fewer transitions from domestic to wild birds (Fig. 4b,c and Extended Data Fig. 6a). The largest changes in the inferred transitions occurred between the 1:1 and 1:2.5 titrations, with minimal to no changes observed between transitions inferred in the 1:2.5 and 1:3 datasets, suggesting stability in the inferred transitions at the end of the experiment (Supplementary Table 12). The phylogeny of the final dataset (1:3 ratio of domestic to wild sequences) shows 106 introductions into domestic birds, and 4 from domestic to wild (Fig. 4a,b, Supplementary Figs. 12 and 13

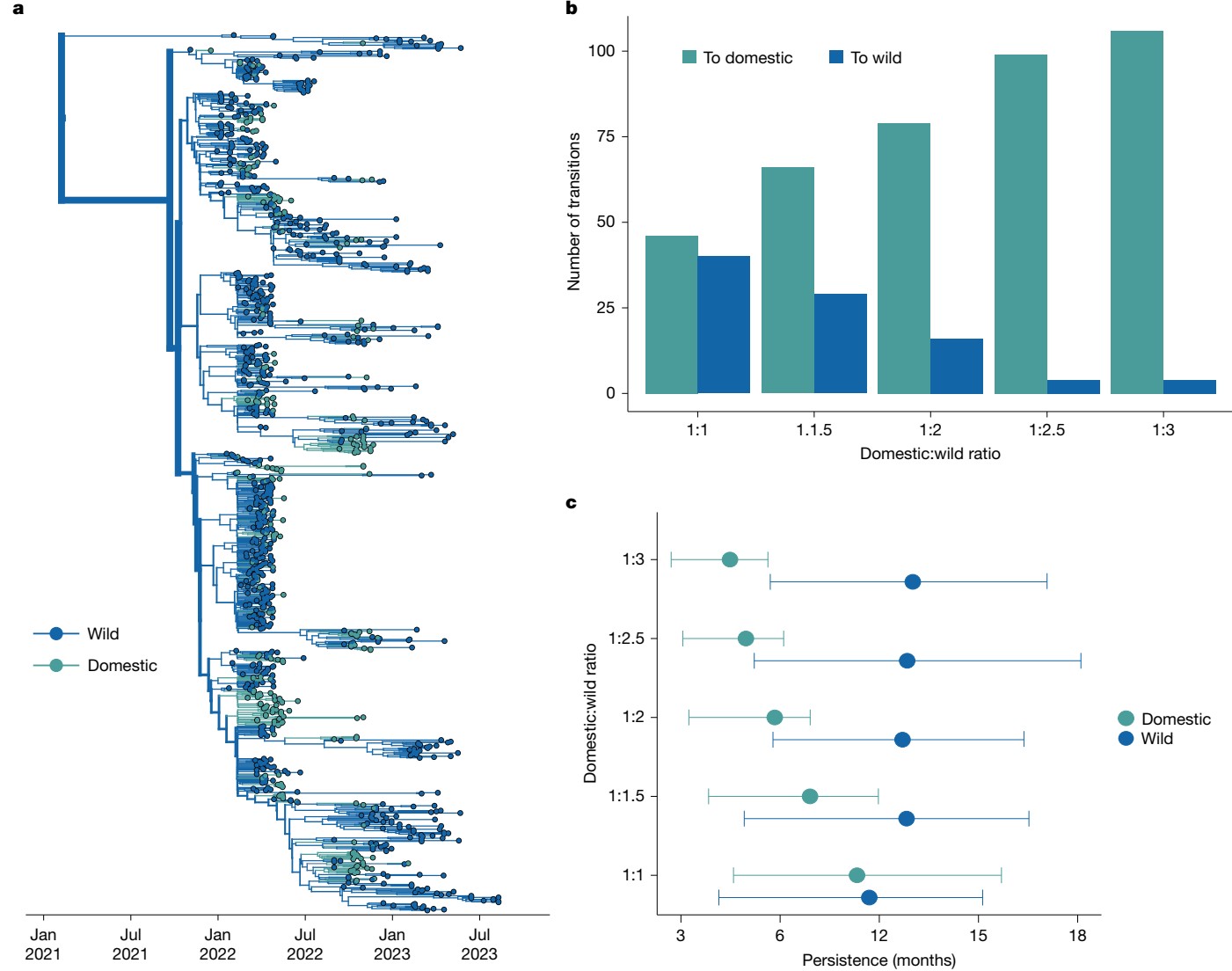

**Fig. 4 | Outbreaks in domestic birds were seeded by repeated introductions from wild birds, with some onward transmission. a**, Phylogenetic reconstruction of transmission between wild and domestic birds. Taxa and branches are coloured by wild or domestic host status containing a 1:3 ratio of domestic to wild bird sequences. *n* = 1,080. **b**, The number of transitions from a given trait to another trait inferred through ancestral state reconstruction for each titration. **c**, The results of the PACT analysis for persistence in domestic and wild birds for each titration.

and Supplementary Table 12). While domestic bird lineages persisted for around 4.5 months on average (95% HPD = 2.7–5.63), viral lineages in wild birds persisted for over twice as long (around 10 months, 95% HPD = 5.7–14.07; Fig. 4c).

Commercial turkey operations have been heavily impacted during the epizootic, comprising 53.7% of all detections on commercial farms[44]. To determine whether excluding turkey sequences (Methods) may have biased our results, we assigned any turkey sequence not labelled as 'wild turkey' as 'domestic' and reran the titration analysis. Turkey sequences did not substantially change the inferred transition rates between wild and domestic birds (Extended Data Fig. 6a and Supplementary Table 12). In both titration experiments, the final number of inferred transmission events from domestic to wild birds was 4 (Supplementary Table 12), indicating minimal transmission back to wild species, regardless of whether turkeys were included (Supplementary Fig. 14 and Supplementary Table 12). Inclusion of turkey sequences did result in a slightly longer inferred domestic bird persistence (1.29 and 1.54 months; Extended Data Fig. 7e) as well as some turkey-only clusters on the tree (Extended Data Fig. 6c–e). Reconstruction using a dataset with equal turkey and domestic (non-turkey) sequences showed that, while most introductions into turkey populations stemmed from

wild birds (42 transitions), transmission events between turkeys and other domestic birds were frequent. We infer around 38 introductions from turkeys to other domestic birds, and 18 in the opposite direction (Extended Data Fig. 7a–d and Supplementary Table 13), suggesting a putative role for turkeys in mediating transmission between wild birds and other poultry production types.

These data suggest a few important conclusions. First, wild birds are inferred as the major source of transmission even when heavily downsampled, and independent of whether turkeys were included in the analysis. Second, regardless of sampling regime, we find that outbreaks in agricultural birds were driven by repeated, independent introductions from wild birds, with some onward transmission between domestic operations. While the exact number of inferred introductions vary across analyses (Supplementary Tables 12 and 13), we infer no fewer than 46, and as many as 113 independent introductions into domestic birds. When allowing sampling frequencies to approximate detections (the 1:3 dataset), we resolve a higher number of introductions into domestic birds with shorter transmission chains, although lineages still persisted for 4–6 months. Together, these results indicate that—while the epizootic of 2014/2015 was started by a small number of introductions that rapidly propagated between commercial

operations[2,12]—intensive and persistent transmission among wild birds since 2022 resulted in continuous incursions into domestic birds. Thus, wild birds had a critical role in agricultural outbreaks in North America from 2021–2023, marking an important departure from past epizootics that may necessitate updates to biosecurity, surveillance and outbreak control.

## Spillovers to backyard/commercial birds

The 2014/2015 H5Nx epizootic in the USA was driven by extensive transmission in commercial poultry[2], prompting a series of biosecurity updates for commercial poultry farms[12,45]. However, not all domestic birds are raised in commercial settings. Rearing domesticated poultry in the home setting has become increasingly popular in the USA, with an estimated 12 million Americans owning 'backyard birds' in 2022 (ref. 46). These birds have been heavily impacted during the ongoing epizootic, with some evidence for distinct transmission chains circulating in backyard birds versus commercial poultry[15]. As backyard birds generally experience less biosecurity than commercial birds and are more likely to be reared outdoors[47], we hypothesized that spillovers into backyard birds may be more frequent than spillovers directly into commercial poultry.

To test this hypothesis, we used a subset of sequences sampled between January and May of 2022, with additional metadata specifying whether they were collected from commercial poultry or from backyard birds. We built a tree with equal sequences from domestic and wild birds, with domestic sequences split between commercial and backyard birds (commercial birds = 85, backyard bird = 85, wild birds = 193). As previously, we infer wild birds as the primary source population, with multiple introductions into commercial and backyard birds (Extended Data Fig. 8a). However, backyard bird sequences clustered more basally than commercial poultry sequences, sometimes falling directly ancestral to clusters of commercial poultry sequences (Extended Data Fig. 8a). While all backyard bird clusters descended from wild birds, 10 out of 26 commercial poultry introductions were inferred from backyard birds (Supplementary Fig. 15a). This pattern was reproducible across multiple independent subsamples, indicating robustness to the exact subset of sequences in the tree. Given the debated link between backyard birds and commercial poultry[48], we further explored two hypotheses that could explain this pattern. The first is that backyard birds mediated transmission between wild birds and commercial birds. Under this model, spillovers into backyard birds (possibly through outdoor rearing) could be spread to commercial populations through shared personnel, clothing or equipment, resulting in backyard bird sequences clustering between wild and commercial bird sequences. Alternatively, backyard birds could have been infected earlier than commercial birds. If backyard birds have a higher risk of exposure (possibly due to lessened biosecurity and increased interactions with wildlife), then a successful spillover event may take less time to occur and be detected in backyard birds, resulting in clustering that is more basal in the tree.

To differentiate between these hypotheses, we performed a second titration analysis. We started with the phylogeny including equal numbers of sequences from commercial and backyard birds, enabling us to directly compare introduction patterns in these two groups. We then added progressively more wild bird sequences into the tree until all available wild bird sequences were added and, for each dataset, inferred the number and timings of transmission events between wild birds, commercial birds and backyard birds. If backyard birds mediated outbreaks in commercial birds (hypothesis 1), then the relationship between backyard birds and commercial birds should remain unchanged. If backyard birds and commercial birds were infected independently (hypothesis 2), then wild bird sequences should intersperse between commercial and backyard bird sequences, resulting in more independent introductions that occur earlier in backyard birds.

Throughout the experiment, wild bird sequences attached throughout the phylogeny, disrupting nearly every backyard bird-commercial bird cluster originally observed (Extended Data Fig. 8). The final tree with all available wild bird sequences resulted in inference of around 82 independent introductions from wild birds to domestic birds, with most clusters containing only commercial (39 clusters) or backyard bird (43 clusters) sequences (Fig. 5a,b, Extended Data Fig. 8 and Supplementary Fig. 15), suggesting that outbreaks in these groups were mostly seeded independently. Of the initial ten transmission events inferred from backyard birds to commercial birds, only two remained undisturbed in the final tree (Fig. 5b and Supplementary Fig. 15), representing outbreaks in the same state and week, which could be plausibly linked. However, all of the other clusters were disrupted. As wild bird sequences were added into the tree, the number of inferred introductions into backyard birds and commercial birds diverged across the posterior trees for each titration (Extended Data Fig. 8), with backyard birds experiencing slightly more introductions (mean = 42 introductions, 95% HPD = 35–49) than commercial poultry (mean = 39 introductions, 95% HPD = 32–44) (Fig. 5c).

To determine whether spillovers into backyard birds occurred earlier than those into commercial poultry, we estimated the number of transitions between hosts across the phylogeny (Markov jumps) and the amount of time that is spent in each host between transitions (Markov rewards)[49,50]. Early in the epizootic, transmission in backyard birds slightly preceded transmission in commercial poultry (Fig. 5d and Supplementary Fig. 15). Enumeration of the cumulative number of transitions between hosts (Markov jumps), showed that backyard birds experienced slightly more jumps than commercial poultry (backyard birds = 43 introductions, 95% HPD = 36–50; commercial birds = 39 introductions, 95% HPD = 32–44), and that these introductions occurred around 9.6 days earlier on average (Fig. 5e). Comparison of detections and sequence availability show no apparent skewing in samples for commercial and backyard birds in that time period, suggesting that this pattern is not simply due to excess earlier cases in backyard birds at that time (Supplementary Fig. 16). Data on testing turnarounds and enrolment in the US indemnity payment register show that commercial and backyard bird farms have nearly identical lag times between case reporting and confirmation (2.15 days for commercial birds, 2.4 days for backyard birds)[51], with testing and depopulation in commercial poultry that is efficient[52] and slightly earlier than in backyard birds. While 511 out of 168,048 commercial operations (0.3%) reported cases and received indemnity payments (a proxy for enrolment in testing programs), only 656 out of around 12 million backyard bird owners (0.0055%) were enrolled[46,51,53]. Thus, the earlier spillovers that we observe cannot be readily explained by systematically earlier case detection, testing or reporting. Future studies using expanded datasets across future epizootic waves are necessary to confirm this pattern more broadly.

## Discussion

Our study collectively supports wild birds as critical sources of the North American H5N1 epizootic. By directly modelling transitions between host groups based on domestic/wild classification, taxonomic order and migratory behaviour, paired with strong dispersal across flyways, we show that wild birds were key drivers of epizootic transmission and introductions into agriculture. These results imply that continuous surveillance in wild birds, particularly Anseriformes[54], may now be critical for viral tracking and outbreak reconstruction. As the primary source of transmission shifts from poultry to wild migratory birds, the ecology of clade 2.3.4.4b viruses in North America may now follow patterns unfolding globally, whereby evolution is increasingly governed by wild bird movement, ecology and reassortment. Recent modelling of HPAI risk in Europe identified Anatinae and Anserinae Anseriformes prevalence as consistent predictors of HPAI detection[54], supporting wildlife surveillance for outbreak forecasting and risk assessment.

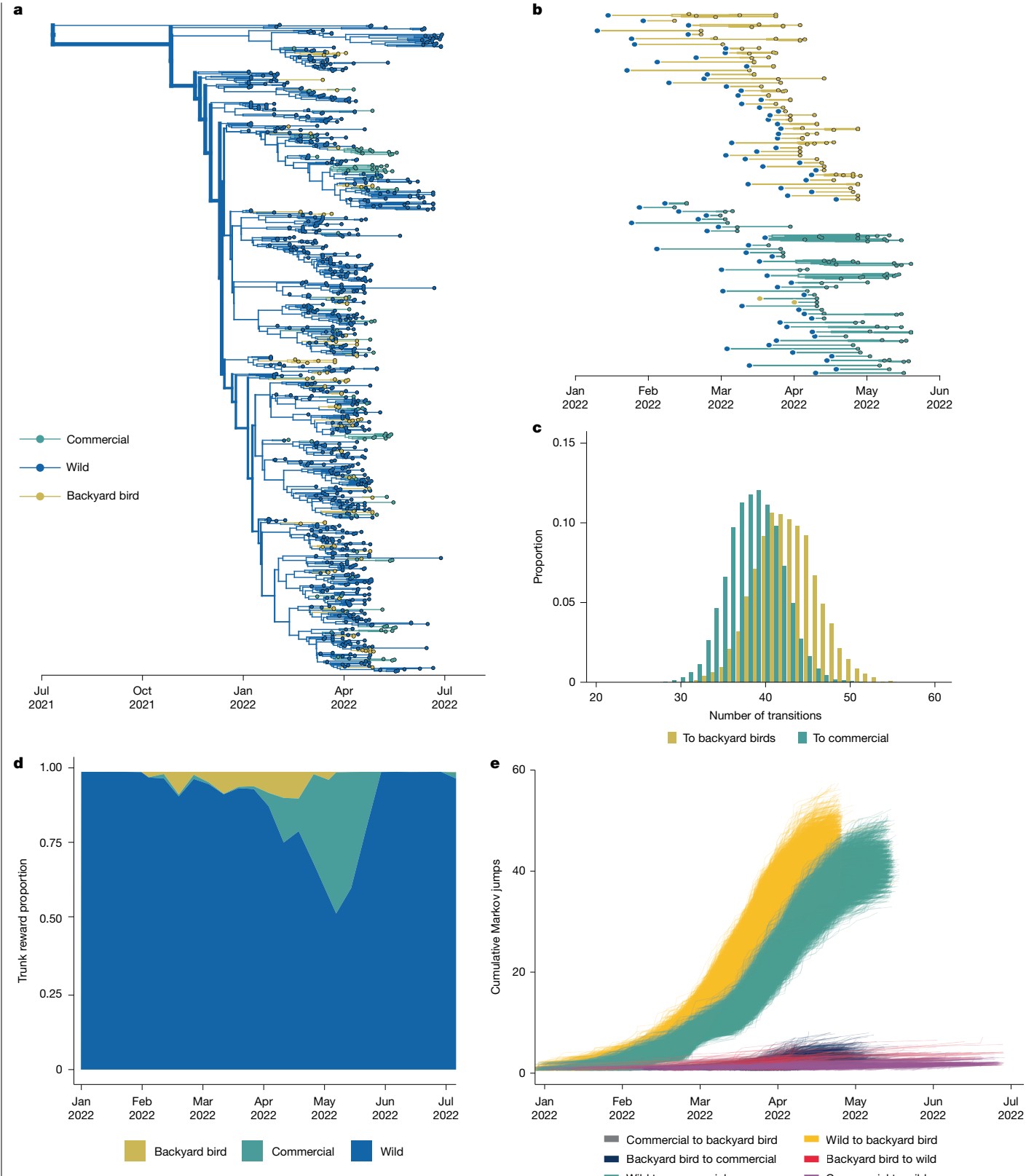

**Fig. 5 | Backyard birds were infected independently, and earlier on average than commercial birds. a**, Phylogenetic reconstruction of sequences collected between Jan 2022 and May 2023, with all available wild bird sequences and equal proportions of commercial and backyard birds. *n* = 942. Taxa and branches are coloured by host domesticity status. **b**, Exploded tree view of the phylogeny showing the branches of transmission in each domestic bird type after transmission from wild birds; subtrees represent the traversal of a tree from the root to the tip, whereby the state is unchanged from the initial state

(given by the large dot on left) to the tips represented by the smaller dots representing continuous chains of transmission within a given state. **c**, The proportion of trees from the posterior tree set with a given number of transitions from wild birds to backyard birds and commercial birds (100% available wild sequences). **d**, The Markov reward trunk proportion for domesticity status showing the waiting time for a given status across branches of the phylogeny over time. **e**, Cumulative Markov jumps from a given bird type to another over time; each line represents a single phylogeny from the posterior sample of trees.

Future work investigating the use of real-time tracking of wild bird abundance and movement for forecasting outbreaks may be useful for formulating new approaches to prevention.

Our study highlights the capacity of migratory birds to rapidly disseminate highly pathogenic H5N1 viruses across North America. We speculate that rapid geographical spread from east to west could be explained by the high inherent transmissibility of clade 2.3.4.4b viruses in wild birds, rapid avian migration or exponential spread among immunologically naive wild birds during early epizootic expansion[55,56]. We infer five incursions[57] into the Pacific that mostly persisted transiently, suggesting frequent viral flow between Asia and the Pacific coast of North America. Limited transmission from the Pacific flyway could be explained by differential fitness of the lineages introduced into the Pacific versus Atlantic flyways, ecological isolation of the Pacific flyway[58–60], differences in host distributions at the locations and times of these incursions or simply due to chance. While future work is necessary to differentiate among these hypotheses, these data support the Pacific coast as an important region for capturing viral transmission between Asia and North America.

We find that outbreaks in agriculture were seeded by repeated introductions from wild birds, a pattern that held true regardless of sampling regime, and that aligns with global observations that clade 2.3.4.4b viruses are increasingly spread by wild birds[8,61]. These findings contrast with the epizootic in 2014/2015, in which a small number of introductions spread efficiently between commercial poultry operations[2,12]. As the viruses circulating in 2014/2015 did not establish in local wild bird populations, that epizootic subsided following aggressive culling. Since 2014/2015, biosecurity plans have improved[12,45] and depopulation occurs more rapidly[12,52], potentially contributing to the shorter domestic persistence and limited transmission back to wild birds we observe. Despite these improvements, efficient transmission in wild birds probably allowed for rapid dispersal and continuous outbreak reseeding, making this epizootic far more challenging to control. US and Canadian policy currently classifies H5N1 as a foreign animal disease, meaning that biosecurity to reduce spread between farms and rapid culling[62,63] are prioritized for outbreak control. Although these control measures will probably remain important, our results suggest that reducing future spillovers into agriculture may now necessitate changes in management priorities. The repeated spillovers that we identify suggest that gaps in farm biosecurity remain that could be enhanced to reduce outbreak risk. Finally, layered approaches, including enhanced wild bird monitoring, new methods to separate wild and domestic birds, and potentially domestic animal vaccination, may necessitate exploration.

Using a small dataset from the first 6 months of the epizootic, we find phylogenetic evidence that spillovers into backyard birds may have occurred slightly earlier and more frequently than those into commercial farms. A large survey of backyard bird populations from 2004 showed that backyard bird flocks often contain multiple species, usually have outdoor access, and that 60–75% regularly interact with wild birds[47]. Biosecurity precautions tend to be much more limited in backyard populations, with 88% of backyard flocks using no precautions (shoe covers, footbaths, clothing changes) at all[47]. Given the enhanced exposure of backyard birds to wild birds, expanded studies to determine whether the patterns of earlier spillovers in these populations hold true more broadly are necessary to investigate backyard birds as potential sentinel species for transmission in wild birds.

Sampling bias is pervasive across viral outbreak datasets, and no modelling approach can completely overcome biases in data acquisition. In the USA, only wild Anseriformes are sampled live or hunter harvested, while all other host groups are sampled sick or dead. Detections in domestic birds depend on producer reporting and testing, which probably varies across production types, locations and premises. To account for this variability, we used multiple subsampling approaches, reported results that were consistent and carried out statistical tests to measure the impact of sampling on our results. The titration tests that we used show that the precise number of transitions between wild and domestic birds depends on sampling numbers, providing a clear argument for continuous surveillance in wildlife, and a warning for overconfidence in estimating the transitions between groups. Still, all phylodynamic inferences are limited by the availability of sequencing data, and the results could change if future data become available. Our analyses use only HA sequences, meaning that differences between reassortants could not be compared[23]. Finally, although we retain data from across North America for all analyses, our results are probably most informative of transmission within the USA during the first 6 months of the epizootic.

Taken together, we show that wild birds played the central role in dispersal of the 2021–2023 H5N1 epizootic. Transmission in wild birds provides an explanation for the rapid cross-continental spread and continued agricultural outbreaks despite aggressive culling. Our results highlight the utility of wild-bird surveillance for accurately distinguishing hypotheses of epizootic spread, and suggest that continuous surveillance is critical for preventing and dissecting future outbreaks. Our data underscore that continued establishment of H5N1 in North American wildlife may necessitate a shift in risk management and mitigation, with interventions focused on reducing risk within the context of enzootic circulation in wild birds. At the time of writing, outbreaks in dairy cattle highlight the critical importance of modelling ecological interactions that drive spillovers between wildlife and domestic production to inform biosecurity, outbreak response and vaccine strain selection.

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

## Methods

### Dataset collection and processing

**Information on case detections in North America.** In this study, a detection is defined as a positive PCR test from a collected sample. In Canada, year-round surveillance in wild and domestic populations is coordinated by the Canadian Food Inspection Agency, Environment Canada, the Public Health Agency of Canada and the Canadian Wildlife Health Centre[64]. In the USA, the United State Department of Agriculture Animal and Plant Health Inspection Service (APHIS) manages HPAI surveillance and testing in wild birds through investigation of reported morbidity and mortality events, hunter-collected game birds/waterfowl, sentinel species/live bird collection, and environmental sampling of water bodies and surfaces[43,65]. USDA APHIS also surveilles domestic birds using several reporting methods: mandatory testing through the National Poultry Improvement Plan, coordination with state agricultural agencies, routine testing in high-risk areas and backyard flock surveillance[66].

Data on detections of HPAI in the USA used in analyses for this study were collected from USDA APHIS. Reports for mammals, wild birds and domestic poultry were all downloaded in November 2023 (download date: 25 November 2023)[40]. During the time period analysed in this study (November 2021-September 2023), most HPAI detections in the USA were reported in wild birds (Supplementary Fig. 1a). Data on domestic bird detections are reported with information on poultry type (such as duck, chicken) and by whether the farm is classified as a commercial operation or backyard flock. Backyard flocks are categorized by the USDA as operations with fewer than 1,000 birds[47,67] and by the World Organization for Animal Health (WOAH) as any birds kept in captivity for reasons other than for commercial production[68]. Among domestic birds, detections (1,177 total) came predominantly from commercial chickens (9.3%), commercial turkeys (28.5%), commercial breeding operations (species unspecified) (15.3%) and birds designated WOAH non-poultry, which refers to backyard birds (42.3%) (Supplementary Fig. 1b). Other domestic bird detections occurred in game bird raising operations (2.5%) and commercial ducks (2.0%). The North American epizootic has impacted a broad range of mammalian hosts, with detections (399) reported in red foxes (24.3%), mice (24.1%), skunks (12.2%) and domestic cats (13.2%). Other mammalian hosts (26.2%) represent a wide range of species including harbour seals, bobcats, fishers and bears (Supplementary Fig. 1c).

**Genomic data processing and initial phylogenetics.** We downloaded all available nucleotide sequencing data and associated metadata for the haemagglutinin protein of all HPAI clade 2.3.4.4b H5Nx viruses from the GISAID database on 25 November 2023 (ref. 69). For each subset of the data described for further phylodynamic modelling, the following process was followed. We first aligned sequences using MAFFT v.7.5.20, sequence alignments were visually inspected using Geneious and sequences causing significant gaps were removed and nucleotides before the start codon and after the stop codon were removed[70,71]. We deduplicated identical sequences collected on the same day (retaining identical sequences that occurred on different days). We identified and removed temporal outliers for all genomic datasets by performing initial phylogenetic reconstruction in a maximum-likelihood framework using IQtree v.1.6.12 and the program TimeTree v.0.11.2 was used to remove temporal outliers and to assess the clockliness of the dataset before Bayesian phylogenetic reconstruction[72,73]. This resulted in a dataset of 1,824 sequences that were used in further analyses (Supplementary Fig. 17).

**Biases in genomic data and $N_e$ inference.** Sequencing data sampled in North America are heavily skewed toward sequences from the USA (USA, 1,590; Canada, 224; Central America, 8), and from the first 6 months of the outbreak, with 74% of all available sequences sampled from January to July 2022 (Supplementary Fig. 2). To evaluate whether sequencing data reflect case detections, we inferred the viral $N_e$—a measure of viral genetic diversity shown to be mathematically related to disease prevalence and the disease transmission rate[26]. We inferred $N_e$ using a nonparametric population model (Skygrid), which captures relative changes in genetic diversity and the variability of growth rate in the virus population over time, providing a proxy for epidemic dynamics as previously described. $N_e$ is modestly correlated with detections (highest Spearman rank correlation: 0.65, $P = 4.4 \times 10^{-11}$) (Fig. 1c and Supplementary Figs. 3 and 4), with peaks in $N_e$ preceding peaks in detections by about 1 week (Supplementary Fig. 5), probably reflecting the lag between viral transmission and case detection. We interpret these results to suggest that, despite uneven sequence acquisition across time, the diversity of sampled sequences roughly reflect the amplitude of H5N1 cases. Given these results, we opted to use sequencing data for the entire sampling period for broad inferences on introductions and geographical spread, but supplement these analyses with controls for sampling differences between groups. For more-intensive reconstructions of transmission patterns between wild birds, commercial poultry and backyard birds, we focus on the initial 6-month period with the most densely sampled data, coupled with experiments to assess the impacts of sampling on results. Finally, although we retained data from Canada and Central America for all subsequent analyses, our results are probably most informative about transmission within the USA due to the heavy skewing of data towards the USA.

**AVONET database.** We downloaded the AVONET database for avian ecology data and merged it to available host metadata from GISAID for each sequence[74]. We used the species if provided to match the species indicated in the AVONET database. If host metadata in GISAID was defined using common name for a bird, we determined the taxonomic species name and used that for further merging with the AVONET data (for example, 'mallard' was replaced with *Anas platyrhynchos*) for the given region to match the species to its respective ecological data. Domesticity status (whether a sequence was isolated from a wild host or a domestic host) was determined using available metadata downloaded from GISAID using the 'Note' and 'Domestic_Status' fields in sequence associated metadata. Moreover, if a given sequence strain name (in the field 'Isolate_Name') indicated domestic status (for example, A/domestic_duck/2022) these sequences were labelled as belonging to domestic hosts.

### Phylodynamic analysis

The following Bayesian phylogenetic reconstructions and analyses were performed using BEAST (v.1.10.4)[75].

**Empirical tree set estimation and coalescent analysis.** We performed Bayesian phylogenetic reconstruction for each dataset before discrete trait diffusion modelling to estimate a posterior set of empirical trees. The following priors and settings were used for each subset of the sequencing data. We used the HKY nucleotide substitution model with gamma-distributed rate variation among sites and log-normal relaxed molecular clock model[76,77]. The Bayesian SkyGrid coalescent was used with the number of grid points corresponding to the number of weeks between the earliest and latest collected sample (for example, for a dataset collected between 4 November 2021 and 11 August 2023, we would set 92 grid points)[78]. We initially ran four independent MCMC chains with a chain-length of 100 million states logging every 10,000 states. We diagnosed the combined results of the independent runs diagnosed Tracer v1.7.2. to ensure an adequate effective sample size (ESS > 200) and reasonable estimates for parameters[75]. If ESS was inadequate additional independent MCMC runs were run increasing chain length to 150 million states, sampling every 15,000 states were performed. We combined the tree files from each independent MCMC run removing 10–30% burn-in and resampling to get a tree file with

between 9,000 and 10,000 posterior trees using Logcombiner v.1.10.4. A posterior sample of 500 trees was extracted and used as empirical tree sets in discrete trait diffusion modelling.

**Discrete trait modelling framework.** For each discrete trait dataset, we used an asymmetric continuous time Markov chain discrete trait diffusion model and implemented the Bayesian stochastic search variable selection (BSSVS) to determine the most parsimonious diffusion network[79]. We inferred the history of changes from a given trait to another across branches of the phylogeny, providing a rate of transitions from A to B per year for each pair of trait states. When reporting these results, we refer to state A as the source population/state and B as the sink population/state. We implemented the BSSVS, which enables us to determine which rates have the highest posterior support by using a stochastic binary operator which turns on and off rates to determine their contribution to the diffusion network. In addition to the discrete trait diffusion rate, we used a Markov Jump analysis to observe the number of jumps between discrete states across the posterior set of trees and estimated the Markov rewards to determine the waiting time for a given discrete trait state in the phylogeny[49,50]. The Markov reward proportion is calculated as the proportion of the phylogeny at a given time being a given discrete state. By looking at the proportion of a given state over time across the phylogeny, we can provide a proxy for how long transmission has occurred in each group between transition events. We calculate the transition rate as a realization of the CTMC process by dividing the number of Markov jumps by the tree height (branch length from the earliest tip to the root of the tree), and separately, by tree length (sum of all branch lengths). For each pairwise transition rate, we calculate the level of BF support that the given rate has. The BF represents the support of a given rate, and is calculated as the ratio of the posterior odds of the given rate being non-zero divided by the equivalent prior odds, which is set as a Poisson prior with a 50% prior probability on the minimal number of rates possible[79]. We use the support definitions by Kass and Rafferty to interpret the BF support where BF > 3 indicates little support, a BF between 3 and 10 indicates substantial support, a BF between 10 and 100 indicates strong support, and a BF of greater than 100 indicates very strong support[80].

Empirical tree sets were used with the discrete traits defined for each sequence to perform discrete trait diffusion modelling. Each discrete trait model was implemented using three independent MCMC chains with a chain length of 10 million states, logging every 1,000 states. Runs were combined using LogCombiner v.1.10.4, subsampling a posterior sample of 10,000 trees/states. The BF support for transition rates were calculated using the program SPREAD3 (ref. 81). Maximum clade credibility trees were constructed using TreeAnnotator v.1.10.4.

**Extraction of phylogenetic metrics.** We calculated the transitions between states across branches of phylogenies estimated from ancestral state reconstructions using the Baltic python package[82]. To calculate the persistence of a given discrete trait, we used the program PACT v.0.9.5, which calculates the persistence of a trait by traversing the phylogenetic tree backwards and measuring the amount of time that a tip takes to leave its sampled state[83].

**Dataset subsampling and definition of discrete traits**

**Geographical introductions analysis.** We characterized the geographical introduction of HPAI into North America by randomly sampling 100 sequences from Europe and Asia for each year between 2021 and 2023 (total, 300 non-North American) and all available North American sequences across the study period. After removal of temporal outliers, this resulted in a dataset of $n$ = 1,927 sequences annotated by continent of origin. The sequencing data available from North America broken down by country are as follows: USA (1,590), Canada (224), Honduras (2), Costa Rica (5) and Panama (1).

**Migratory flyways analysis.** To characterize geographical transmission within North America after introduction, we constructed a dataset of sequences subsampled based on migratory flyway. We used place-of-isolation data to match the US state or Canadian province that the sequence was collected from with the respective US Fish and Wildlife Service Migratory Bird Program Administrative Flyway[30]. We subsampled 250 sequences for each flyway (Atlantic, Mississippi, Central and Pacific) to create a dataset of 1,000 sequences collected between November 2021 and August 2023. In addition to USFWS flyways, we defined four geographical regions going north to south based on latitude lines, with the following delineations for each group. We divided North America into four regions segregated by latitude, with the northernmost group above the 49° N parallel and the southernmost group below the 36° N parallel. We then sampled 916 sequences uniformly across these categories and inferred transitions between these regions.

**Host order analysis.** We classified sequences by host taxonomic order, inferring the host species using designations in the strain name and/or metadata to match species records in AVONET[74]. To ensure that each discrete trait had an adequate number of samples for the discrete trait analysis of host orders, we combined orders in two instances based on taxonomic and behavioural similarity. The order Falconiformes ($n$ = 14), representing falcons, was added to Accipitriformes ($n$ = 363), which includes other raptors such as eagles, hawks and vultures. Pelecaniformes ($n$ = 34), including pelicans, were grouped with Charadriiformes ($n$ = 74, shorebirds and waders) due to their similar aquatic lifestyles and behaviours. Mammals were kept as a broad non-human classification as most samples were of the order carnivora (foxes, skunks, bobcats), apart from samples of dolphins (Artiodactyla) and Virginia opossum (Didelphimorphia). The following orders were omitted due to a low number of sequences: Rheaforimes ($n$ = 2), Casuariiformes ($n$ = 1), Apodiformes ($n$ = 2), Suliformes ($n$ = 7), Gaviiformes ($n$ = 1), Gruiformes ($n$ = 1) and Podicipediformes ($n$ = 1).

Discrete trait approaches assume that the number of sequences in a dataset are representative of the underlying distribution of cases in an outbreak, resulting in faulty inference when this assumption is violated[27,39,53] and bias when groups are unevenly sampled[82,84,85]. To account for differential sampling among these host order groups, we considered two distinct subsampling approaches. The first is a proportional sampling regime in which sequences are sampled proportional to the detections in each host group each month. This common sampling regime assumes that case detections in each group are the closest proxy for the case distribution in the outbreak, and attempts to align sampling with underlying model assumptions. However, this approach may not be appropriate if case detection is heavily biased between groups. For HPAI H5N1 in North America, detections in wild birds are primarily identified when humans report sick or dead birds to wildlife health authorities or wildlife rescues (Supplementary Fig. 1a), which may skew detections towards birds with dedicated rescue services or birds that reside in closer proximity to humans. For example, Anseriformes and raptors comprised 50.2% and 20.3% of all sequences, respectively, which could arise from high case intensity or a higher rate of case acquisition. A second, complementary subsampling approach is to sample sequences equally, meaning that sequences are sampled from each group in perfectly equal numbers. By forcing the number of sequences from each group to be equal, the transmission inference must be driven by the underlying sequence diversity in each group rather than by sampling differences. Given the high variation among detections within each host group, we opted to pursue both sampling regimes and focus on results that were concordant in both. We first performed an AI test to confirm that clustering was sufficient for discrete trait inference (Extended Data Table 1). Next, for each regime, we performed three independent subsamples, where the dataset was sampled either proportional to cases or equally. For the equal sampling

regime, each dataset included 100 randomly sampled sequences per host group, except for Passeriformes, for which only 57 sequences were available. To account for variation across subsampled datasets, we combined the results for the three independent subsamples to summarize statistical support (Supplementary Fig. 12 and Supplementary Tables 4 and 5). Owing to similar tree topologies across replicates, we visualized the phylogeny of the dataset with the highest posterior support (equal order subsample 1) in the main text and make the results of all analyses available in supplement (Supplementary Fig. 8 and Supplementary Tables 4–11). Finally, to measure the effects of potential sampling bias on the inferred transition rates, we performed a modified tip-shuffle analysis. We generated 100 datasets in which the host tip assignments were randomly shuffled, re-inferred the host group at internal nodes and infer a mean root state probability for each host across the 100 shuffled datasets. We then compared the root state probability in the empirical data to that inferred in the shuffled data as a measure of the impact of sampling on the results as previously described (see the 'Tip-shuffle analysis' section for further details)[38].

For the equal sampling regime, we randomly subsampled 100 sequences for each host order between 4 November 2021 and 11 August 2023, resulting in a dataset of $n = 655$ sequences whereby all isolates for host orders with less than 100 samples, Passeriformes ($n = 57$) and Strigiformes ($n = 99$) (removing one temporal outlier), were used (Supplementary Fig. 18). We repeated this random subsampling three times, resulting in three separate datasets. For the proportional sampling regime, we performed three subsamples of sequences based on the proportion of detections in each host order group, which were collected between 4 November 2021 and 11 August 2023. Three random proportional samples were taken each with the following number of sequences for each group: Accipitriformes (133), Anseriformes (342), Passeriformes (12), non-human-mammal (16), Galliformes (83), Charadriiformes (40), Strigiformes (29) (total $n = 655$ sequences).

**Migratory behaviour analysis.** We defined discrete traits for use in discrete trait diffusion modelling based on the available sequence metadata and merged AVONET data. In addition to taxonomic order, we defined migratory behaviour. Birds were classified as sedentary (staying in each location and not showing any major migration behaviour), partially migratory (for example, small proportion of population migrates long distances, or population undergoes short-distance migration, nomadic movements, distinct altitudinal migration) or migratory (the majority of population undertakes long-distance migration). We subsampled sequences based on migratory behaviour including non-human-mammals and domestic birds to create a subsample of 500 sequences with equal sampling across behaviour groups.

**Rationale for inclusion of turkeys as domestic birds.** While commercial turkey operations represent 53.7% of all detections on commercial farms[44], the presence of wild turkeys throughout North America makes categorizing turkey sequences as domestic or wild status ambiguous. 98% of all turkey sequences are not associated with metadata on domestic/wild status, and were therefore excluded from the first analysis of domestic/wild bird diffusion. However, epidemiological data suggest that most deposited turkey sequences probably stem from domestic outbreaks. Among case detections during the study period, only 139 were reported in wild turkeys, representing 1.5% of all wild bird detections. By contrast, commercial turkey outbreaks comprised 28.5% of all domestic detections in the study period, suggesting that unlabelled turkey sequences are most likely to have come from domestic birds. While these data are not conclusive, we opted to perform an additional analysis to determine whether our exclusion of turkey sequences (that are most likely domestic) may have biased our results. In the analyses detailed below, turkeys are assumed to be domestic.

**Domestic/wild titration analysis.** To study the impact of sampling of wild birds on the estimation of rates between domestic and wild birds, we created five separate datasets with varying numbers of wild birds for sequences collected between 2021 and 2023. We randomly sampled 270 domestic sequences and 270 wild sequences as the initial 1:1 ratio dataset. We then made four more datasets increasing the number of wild sequences by a factor of 0.5 (adding 135 wild sequences), resulting in a final titration of 1:3 domestic to wild sequences ($n = 1,080$). We applied a two-state asymmetric CTMC discrete trait diffusion model in which sequences were labelled as domestic or wild. All priors and model parameters selected are the same as those described in the empirical tree set description above. To study the impact of the inclusion of turkeys in the transmission between domestic and wild populations, we annotated all unannotated sequences collected from turkeys as domestic (see the rationale in section above). We then created three datasets starting with 525 domestic and 525 wild bird sequences, adding 263 sequences to successive titrations resulting in 1:1, 1:1.5 and 1:2 (domestic:wild) sequencing datasets with a final titration size of 1,575 sequences. We again applied a two-state asymmetric CTMC discrete trait diffusion model in which sequences were labelled as domestic or wild, and all priors and model parameters selected are the same as those described in the empirical tree set description above. To determine whether the proportion of turkeys to other domestic birds would impact the results of the previously described titration analysis we built a dataset in which the domestic bird group had equal numbers of turkey and domestic (non-turkey) sequences. This dataset included 173 turkey, 173 domestic bird and 692 wild bird sequences, totalling 1,038 sequences. Given that turkeys comprised 53.7% of commercial poultry outbreaks in the study period, this sampling regime conforms to both equal and proportional sampling regimes. We applied an asymmetric CTMC discrete trait diffusion model using a BSSVS for a three-trait model with the following states: wild birds, domestic birds (not turkey) and turkey. We performed three independent runs of this analysis using the models and parameters described in the empirical tree analysis section above. All titration replicates were performed using an MCMC chain length of 100 million states sampling every 10,000 states.

**Commercial, backyard, wild-bird titration analysis.** Metadata and annotated sequences were made available describing sequences as being from backyard birds for sequences collected in early 2022 which distinguished them from commercial poultry (previously all sequences being determined domestic)[15]. We used these metadata to create a dataset with equally sampled backyard birds and commercial birds ($n = 85$ for each bird type) and then added all available wild birds ($n = 722$) in 25% increments creating four separate datasets for sequences collected between Jan 2022 and June 2022. This resulted in a final dataset of $n = 942$ sequences. We performed discrete trait diffusion modelling using an asymmetric CTMC diffusion model described in the previous section for sequences labelled as backyard bird, commercial bird and wild bird. Calculation of the lag time between the cumulative Markov Jumps for backyard birds and commercial birds was calculated as the average length of time between points where cumulative Markov jumps are equal between backyard birds and commercial birds. This was calculated for each tree in the posterior.

## Assessment of sampling bias

**BaTs analysis.** To determine whether the discrete traits analysed correlated with shared ancestry in the phylogeny, we employed tip trait association tests implemented in the Bayesian Tip-Association Significance (BaTs) program (v.1.0)[31]. This program assesses the phylogenetic structure of discrete traits across viral lineages using three metrics: the AI, parsimony score (PS) and maximum monophyletic clade size (MC). The AI measures the imbalance of internal nodes of a phylogeny for a given set of traits. The PS calculates the number of state changes in the

phylogeny. The MC measures the maximum number of tips belonging to a monophyletic clade for each discrete trait of interest. These metrics are calculated for the phylogeny as tips are randomly swapped to create a null distribution to compare against. Taken together, these metrics quantify the degree of clustering within the phylogeny, with lower AI and PS values indicating stronger phylogenetic structure, suggesting that closely related taxa tend to share the same trait, whereas higher values indicate weaker structure and more-frequent transitions between trait states. Statistical significance was assessed by comparing observed values against a null distribution generated through randomization, with P values reported for each test. All discrete trait groupings showed evidence for clustering by trait, supporting the use of trait modelling across the tree. The results of BaTS analyses for each discrete trait in this study are provided in Extended Data Table 1.

**Tip-shuffle analysis.** To assess the sensitivity of each of our discrete trait reconstructions to differences in sampling between groups, we implemented a modified version of a tip-swap analysis[38]. As originally developed, a tip swap analysis attempts to assess the impact of trait sampling on discrete trait measurements. An operator is implemented within the MCMC chain that randomly picks pairs of tips and swaps their trait values, thus generating a posterior set of trees among which pairs of trait assignments have been randomly swapped. The probability of each state at the root is then computed, and compared to the inferred root state probabilities in the empirical data. As the root state probabilities in randomized datasets should primarily reflect the frequency of each trait in the analysis, empirical results that differ substantially from this null distribution are interpreted as evidence that the sequencing data are informing the analysis beyond what is expected based on trait frequencies alone. Thus, traits for which the root state probability differs considerably from the root state probability in the null data are frequently interpreted as being informed by the data, rather than sampling bias. While this approach has been shown to perform well on small phylogenies[1,86], the strategy of swapping single pairs of tips poses challenges for larger trees. In our flyways dataset, which includes around 1,000 tips, we found that, even with extremely high operator values (4,000), the traditional tip-swap analysis resulted in a posterior set of trees in which the majority of tips (93.3%) remained assigned to their true state at least 50% of the time, resulting in a null dataset that was only partially randomized. We believe that this is due to the high number of tips in our analysis, resulting in only an extremely small fraction of tips randomized at any given step in the MCMC chain. To overcome this limitation, we instead performed a randomized tip-shuffle analysis. Using the empirical set of trees inferred for each discrete trait analysis, we generated 100 null datasets in which we shuffled the trait assignments randomly across the tips. In this approach, we preserve the phylogenetic tree topology and the ratio of samples from each group, but shuffle their assignments at the tips. For each discrete trait analysis, we generated 100 distinct shuffled versions of the empirical trees, reran the analysis and summarized the resulting posterior distribution by inferring a maximum clade credibility tree. We then computed the root state probabilities for each trait for each MCC tree, and computed the mean root state probability across all 100 replicates. This computed mean is reported in Extended Data Table 2, in the column labelled 'mean root state probability across 100 datasets with randomly shuffled tip states'. We then compared the root state probabilities in the empirical data (reported as 'root state probability in empirical data' in Extended Data Table 2) to the shuffled data as a measure of the impact of sampling on the results. As expected, the root state probabilities inferred in the shuffled datasets are proportional to the number of sequences included for each group. For the analyses using an equal sampling regime (migration, flyway, host orders equal and initial titration tests), this leads to approximately equal expected root state probabilities across groups. By contrast, the root state probabilities in the empirical data generally

differ significantly from expectation, suggesting that the phylogenetic results are informed by the genetic data rather than from sampling alone. The results of tip shuffling analyses for each discrete trait in this study are provided in Supplementary Figs. 19–26 and Extended Data Table 2.

### Reporting summary

Further information on research design is available in the Nature Portfolio Reporting Summary linked to this article.

## Data availability

All data that were used in this analysis were sourced from public databases. The acknowledgement table for GISAID isolates used in this analysis is provided in Supplementary Table 18, which can also be found at GitHub (https://github.com/moncla-lab/North-American-HPAI). Several of the analyses presented have also been publicly made available using a maximum-likelihood framework through the Nextstrain pipeline and a narrative of this work can be found online (https://nextstrain.org/community/narratives/moncla-lab/nextstrain-narrative-hpai-north-america@main/HPAI-in-North-America).

## Code availability

All analytical scripts, metadata annotations and BEAST XMLs used in this analysis are available at GitHub (https://github.com/moncla-lab/North-American-HPAI). This code is also tracked and freely available at Zenodo[87] (https://doi.org/10.5281/zenodo.17259872).

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

**Acknowledgements** We thank M. K. Torchetti for her feedback and discussion about our results. This work was supported by NIH R00-AI147029-05 and by funding from the Centers of Excellence for Influenza Research and Response (CEIRR), funded by NIH 75N93021C00015. L.H.M. is a Pew Biomedical Scholar and is supported by NIH R00-AI147029-05. L.D. is supported by NIH 75N93021C00015 and A.S.J. is supported by NIH R00-AI147029-05.

**Author contributions** L.D. and L.H.M. conceived the project. L.D. and A.S.J. curated data and performed analyses. L.D. and L.H.M. wrote the manuscript. L.D., A.S.J. and L.H.M. edited the manuscript text and figures. All of the authors discussed and approved the manuscript.

**Competing interests** The authors declare no competing interests.

**Additional information**
**Correspondence and requests for materials** should be addressed to Louise H. Moncla.

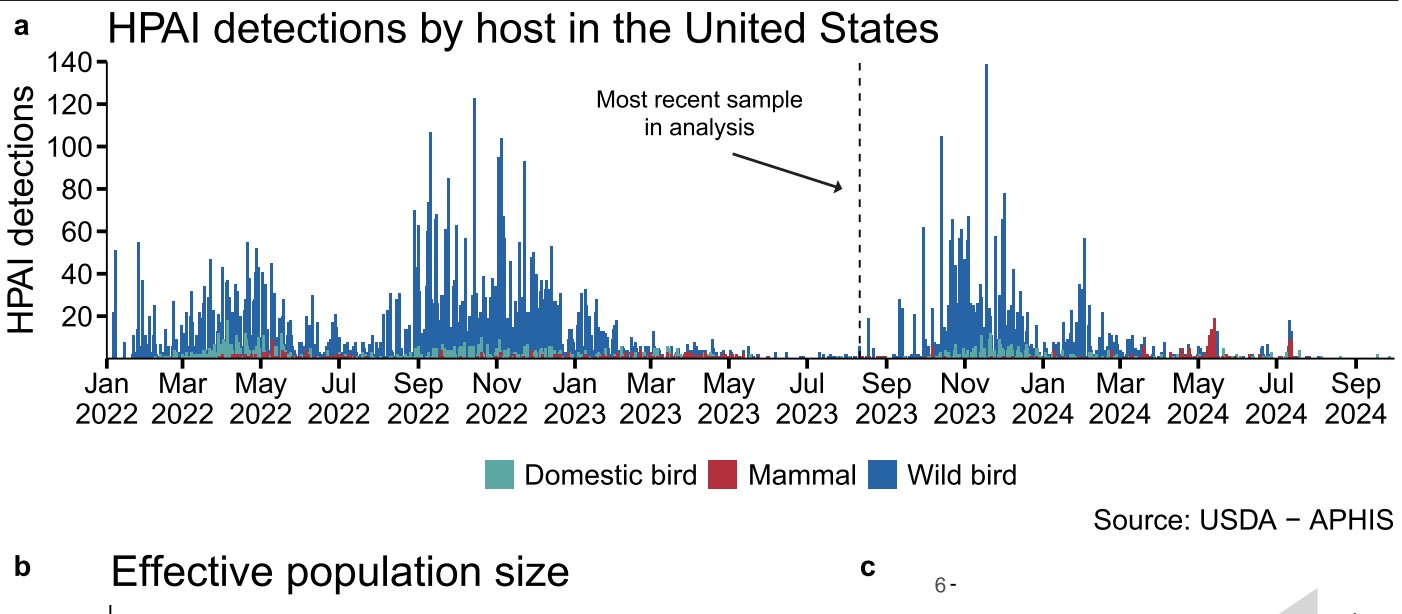

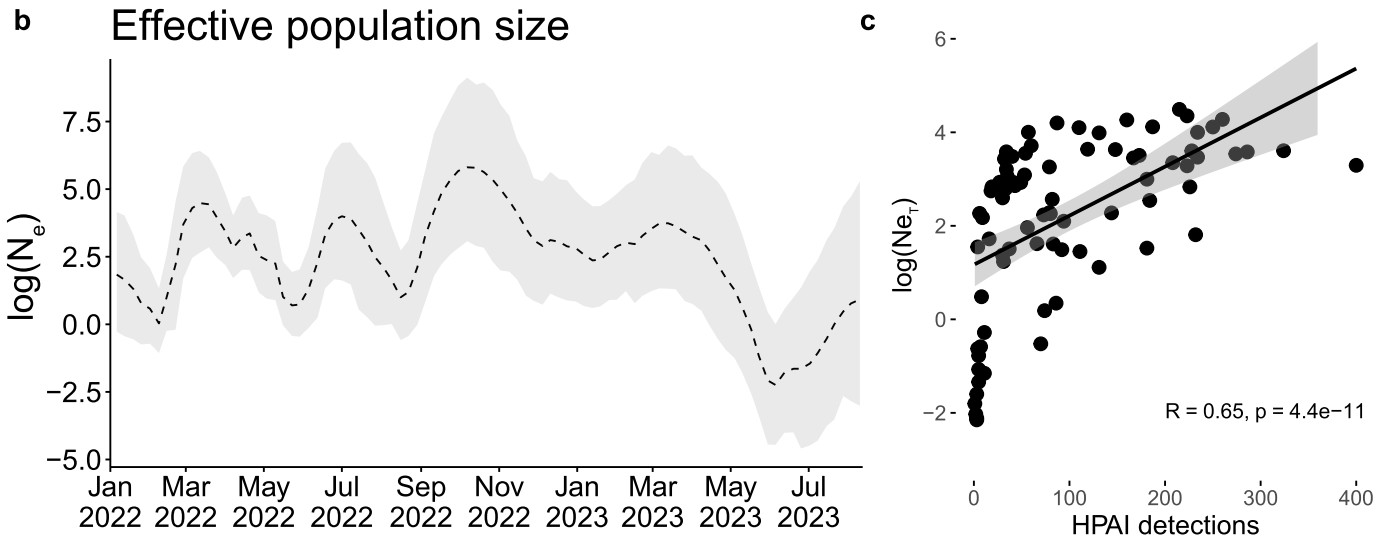

**Extended Data Fig. 1 | Detections of HPAI in North America over time show distinct epidemic waves following introduction events in late 2021.** A) Detections of HPAI in wild birds, domestic birds, and non-human mammals over time. B) The Log-scaled Effective population size ($N_e$) estimates estimated in BEAST using the Bayesian SkyGrid coalescent for sequences collected between Sep 2021 and Aug 2023. C) Correlation plot of log($N_e$) vs HPAI detections by week, spearman correlation displayed.

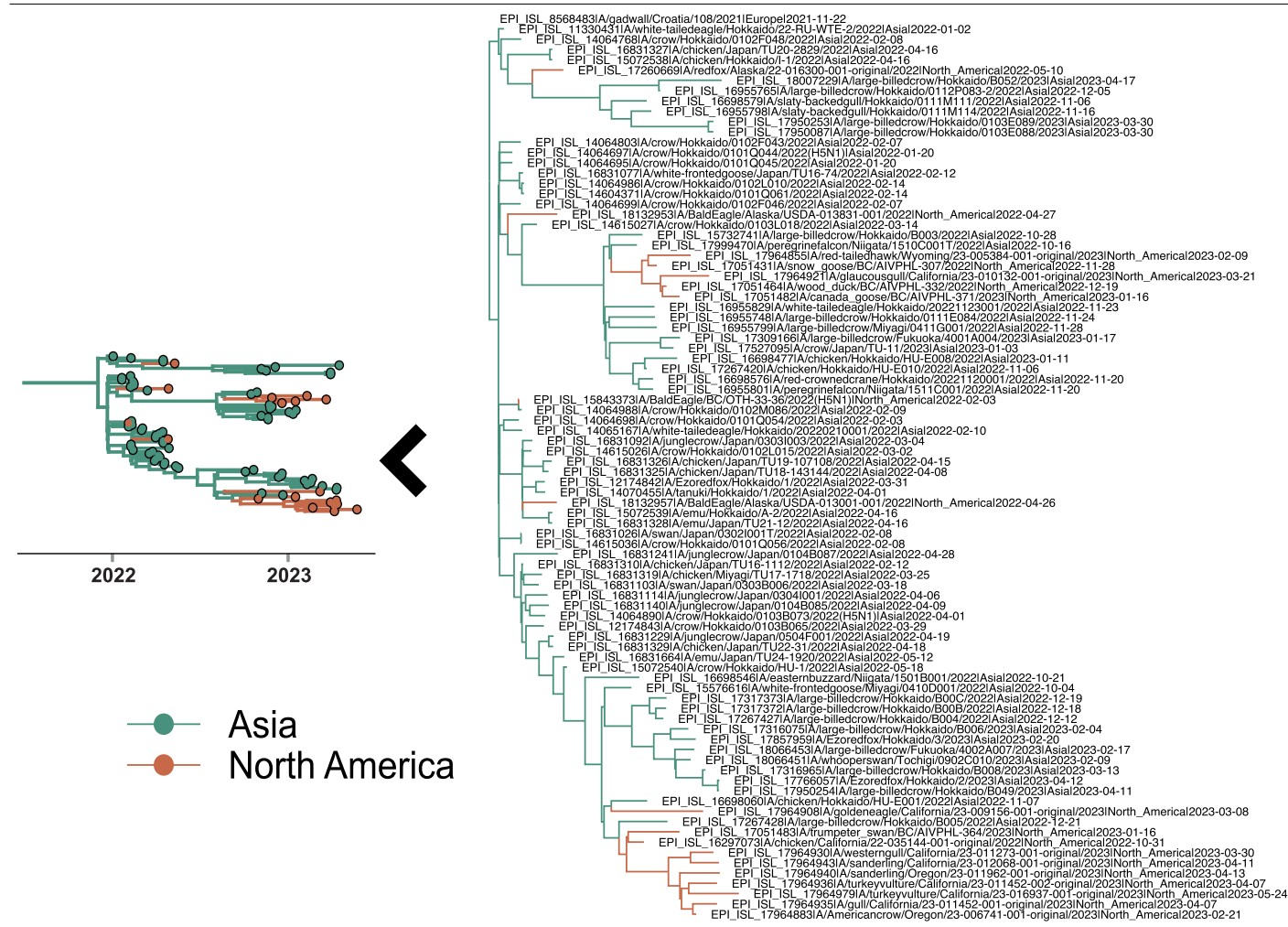

**Extended Data Fig. 2 | Multiple introductions of HPAI into North America from Asia.** An enlarged view of the clade associated with Asian introductions of HPAI into North America is shown, with taxa labelled.

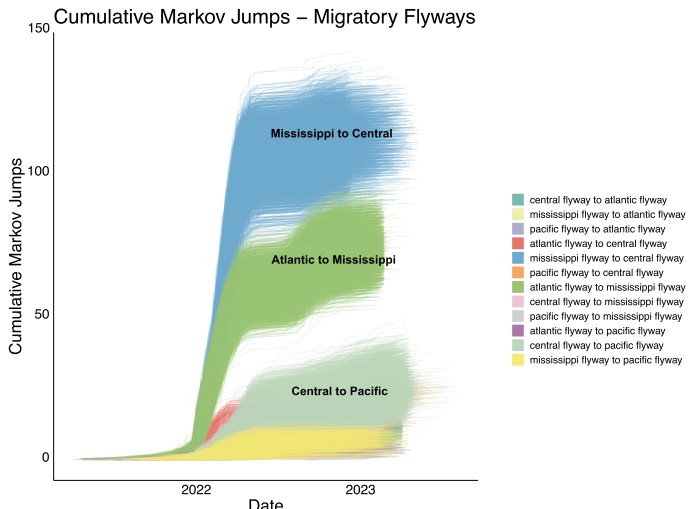

**Extended Data Fig. 3 | Cumulative Markov jumps over time between USFWS flyways.** The cumulative number of markov jumps over time for each transition pair between each flyway with the three largest transition rate pairs labelled. Each line represents a single posterior sample.

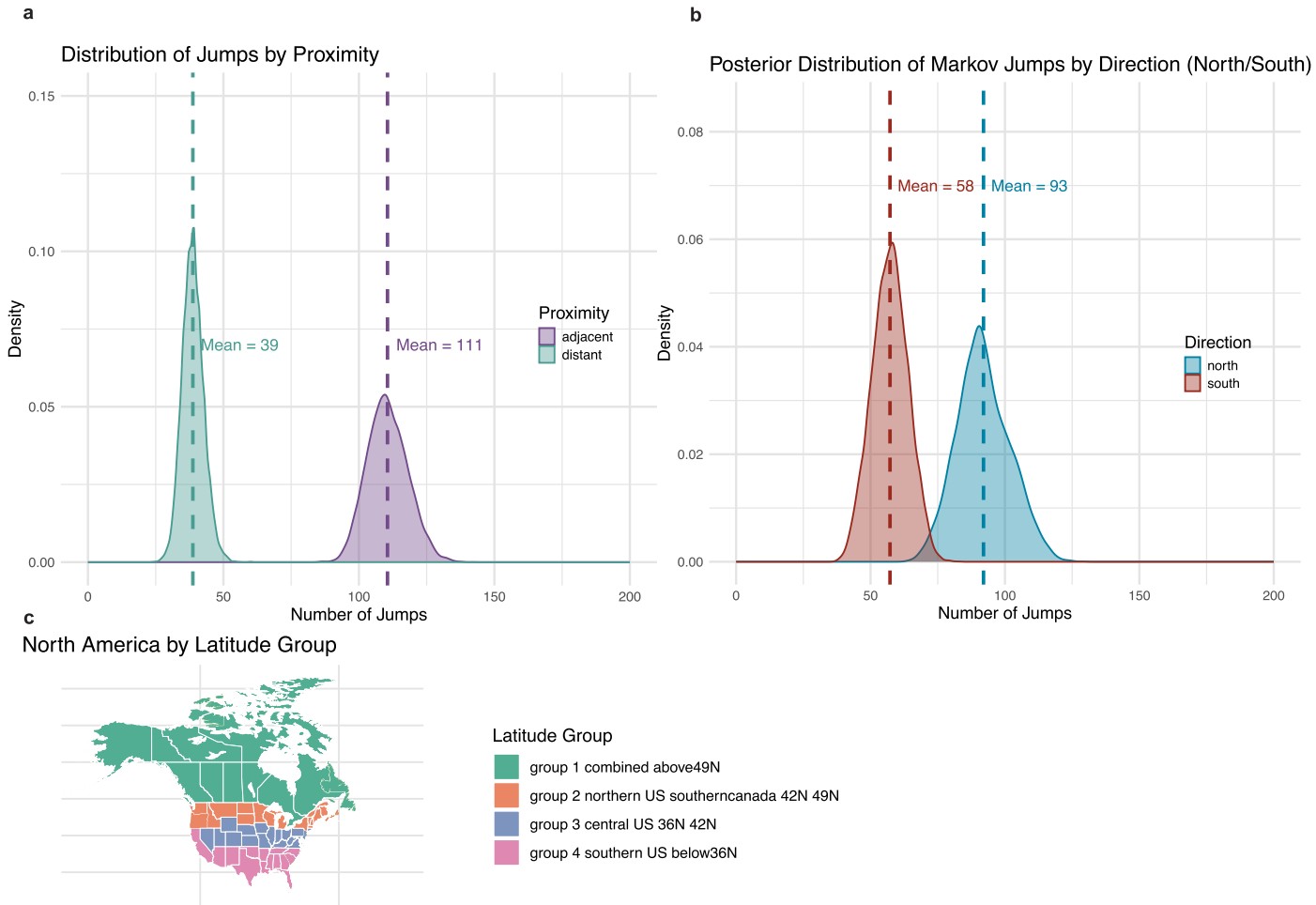

**Extended Data Fig. 4 | Markov Jumps based on direction and adjacency.**
A) Mean number of Markov jumps across the posterior distribution of trees between adjacent (directly next to each other) and distant geographic (not next to each other) groups based on latitude. B) Mean number of Markov jumps across the posterior distribution of trees based on direction of jump (North or South between geographic groups based on latitude). C) Map of North America with states coloured by latitude groups.

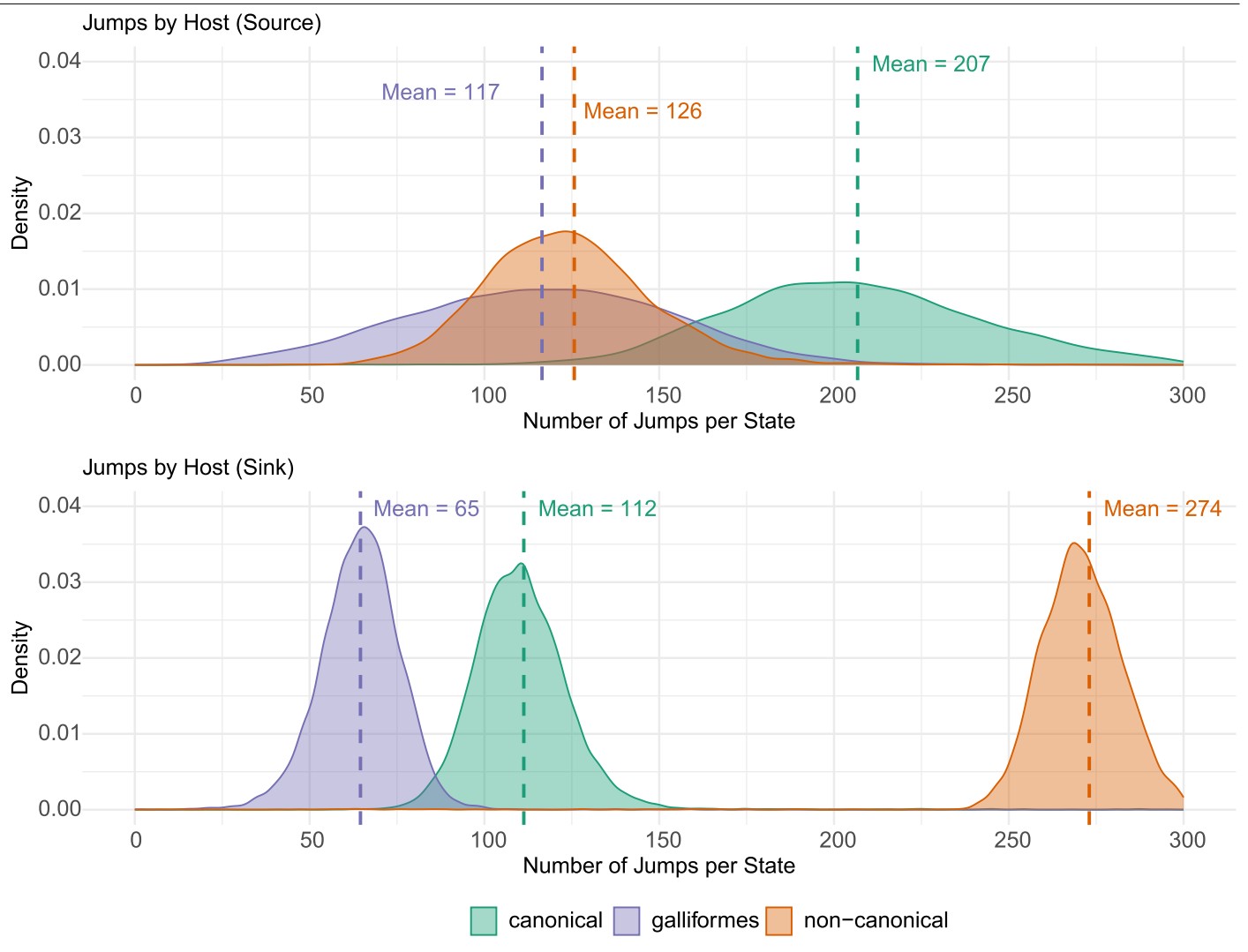

**Extended Data Fig. 5 | Source and sink behaviour by host type.** The top panel shows density plots for the mean number of Markov jumps across the posterior distribution of trees for jumps where the host acts as source. The bottom panel shows density plots for the mean number of Markov jumps across the posterior distribution of trees for jumps where the host acts as sink.

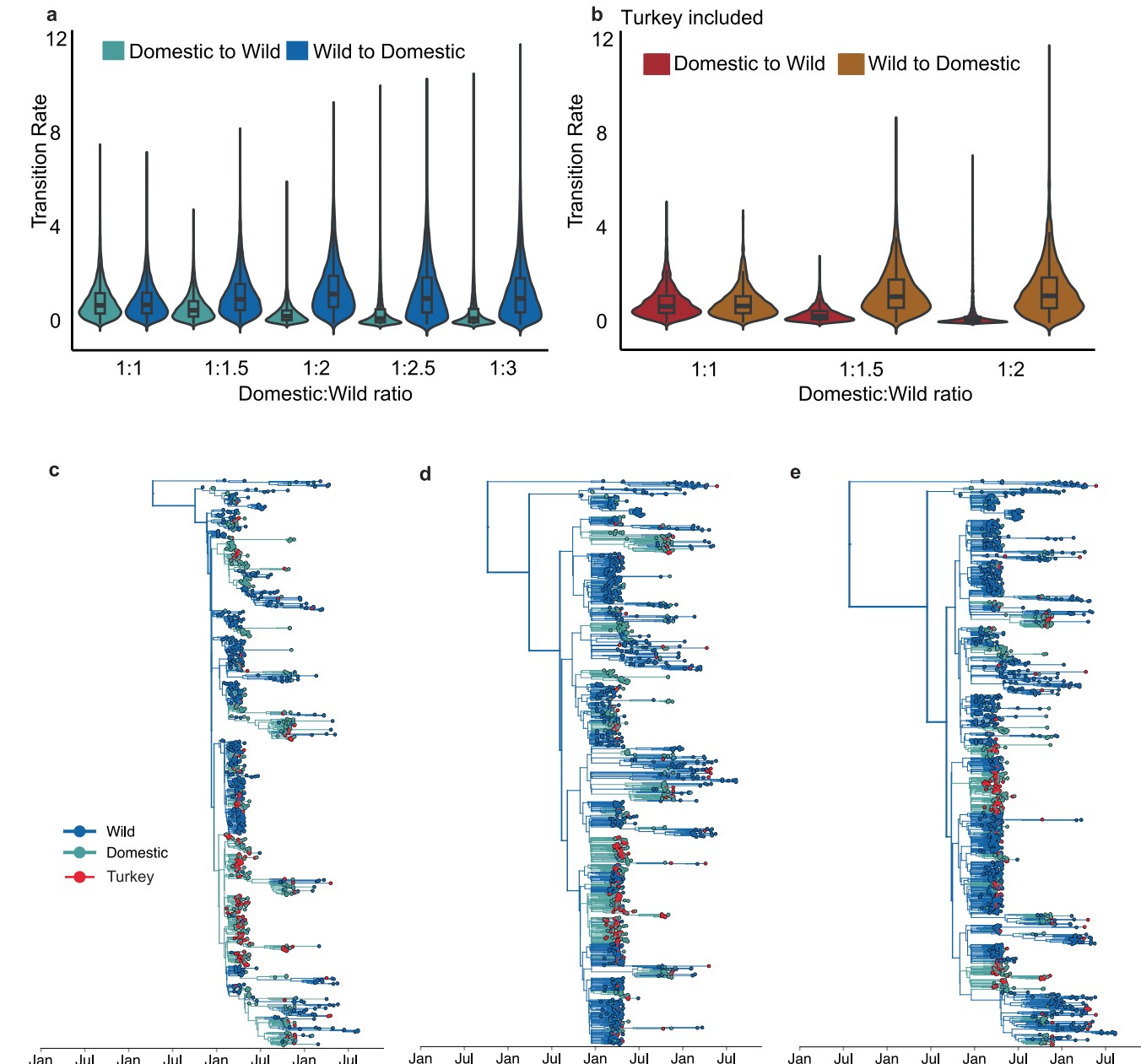

**Extended Data Fig. 6 | Two state rarefaction analysis with turkey sequences.**
A) Violin plots of discrete trait transition rates from domestic to wild birds and from wild to domestic birds for two state titration test. B) Violin plots of discrete trait transition rates from domestic to wild and wild to domestic for the two state titration test that includes turkey sequences. All transition rates had a posterior probability of 0.99. C-E) MCC trees of the ratios of domestic (including turkey) to wild sequences across the titration test, shown in increasing order based on the number of wild bird sequences. Tips and branches are coloured by the state (wild or domestic) of the sample and inferred state respectively with turkey sequences coloured red.

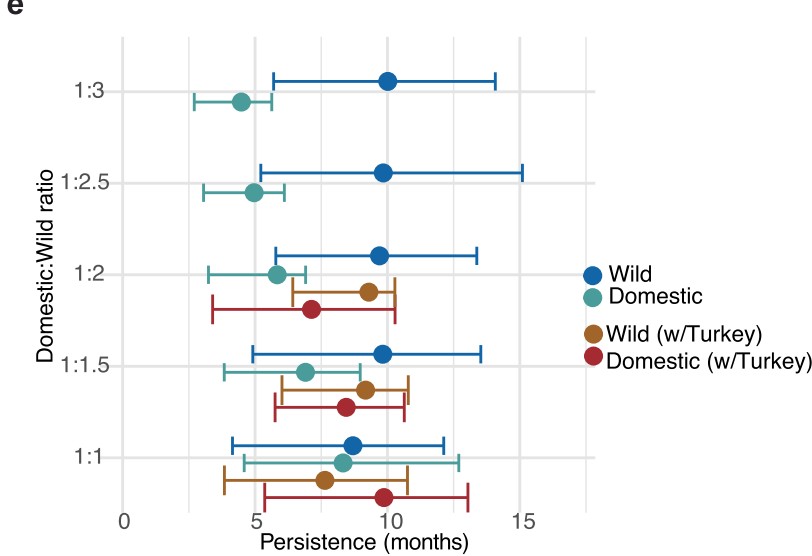

**Extended Data Fig. 7 | Final titration of three state rarefaction analysis with turkey sequences.** A) MCC tree of three state analysis of 1:2 (domestic:wild) sequence dataset with equal numbers of turkey and domestic bird sequences. B) Exploded tree view of the three state 1:2 MCC tree for transitions into domestic. C) Exploded tree view of the three state 1:2 MCC tree for transitions into turkey. D) Exploded tree view of the three state 1:2 MCC tree for transitions into wild birds. E) Combined results of PACT analysis for each titration in the 2-state rarefaction and 2-state rarefaction including turkey. Colours correspond to analysis.

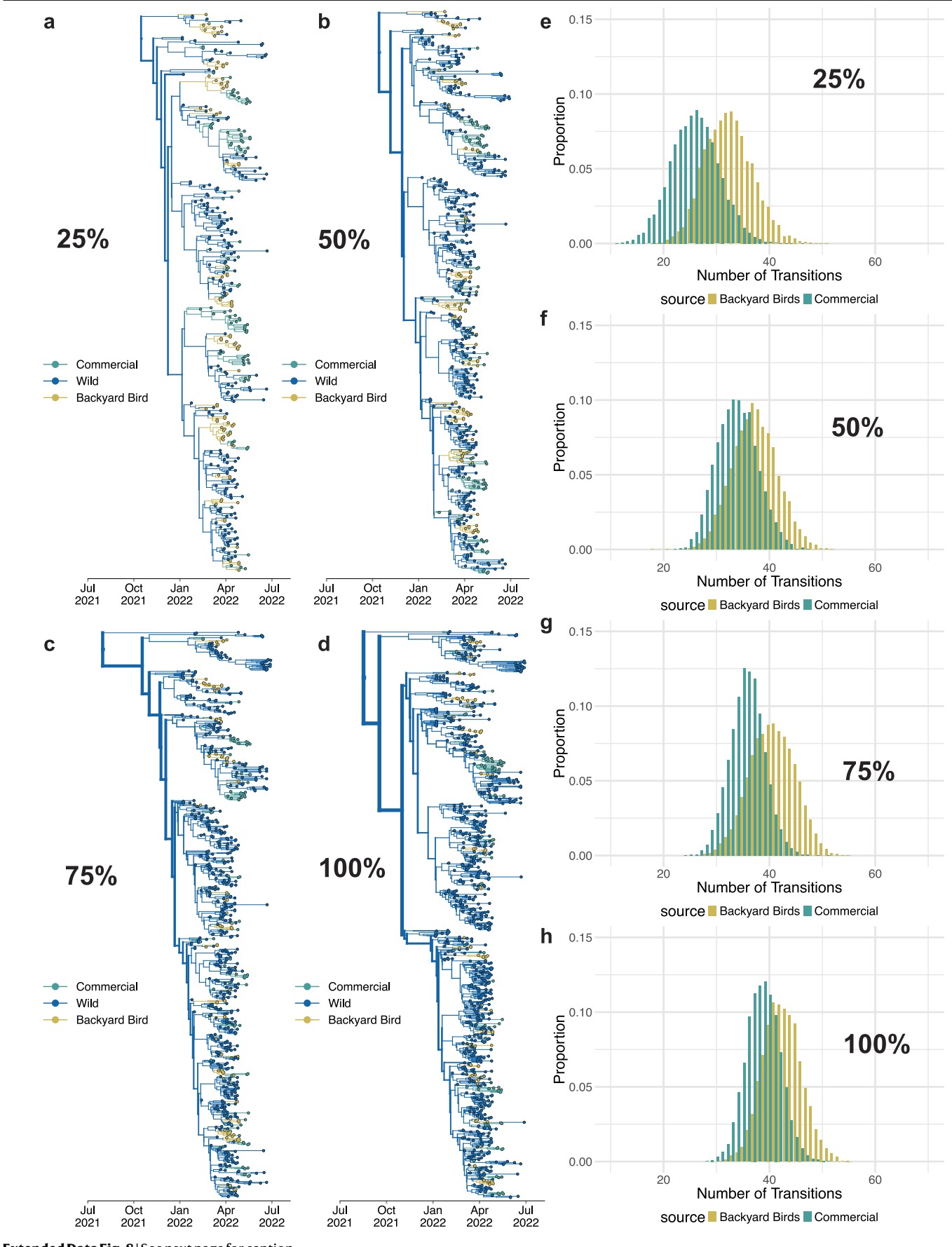

**Extended Data Fig. 8** | See next page for caption.

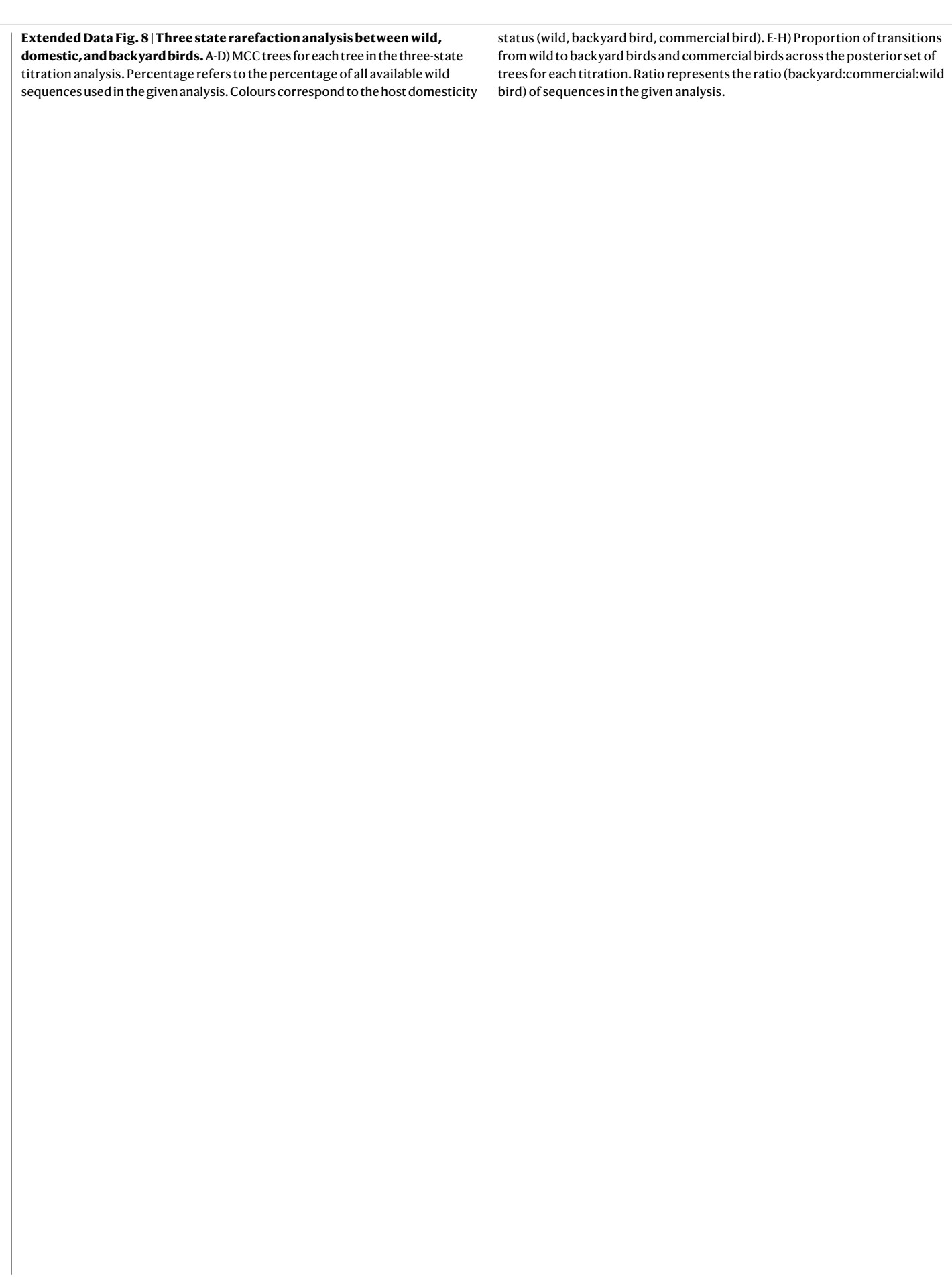

**Extended Data Fig. 8 | Three state rarefaction analysis between wild, domestic, and backyard birds.** A-D) MCC trees for each tree in the three-state titration analysis. Percentage refers to the percentage of all available wild sequences used in the given analysis. Colours correspond to the host domesticity status (wild, backyard bird, commercial bird). E-H) Proportion of transitions from wild to backyard birds and commercial birds across the posterior set of trees for each titration. Ratio represents the ratio (backyard:commercial:wild bird) of sequences in the given analysis.

**Extended Data Table 1 | Results of BaTs analysis for each discrete trait set used**

| | Obs Mean | L95 Obs | U95 Obs | Null Mean | L95 Null | U95 Null | p |
|---|---|---|---|---|---|---|---|
| **Geographic introduction** | | | | | | | |
| AI | 2.221 | 1.585 | 2.946 | 105.122 | 101.926 | 108.689 | 0.001 |
| PS | 19.596 | 19.000 | 21.000 | 550.334 | 544.913 | 556.562 | 0.001 |
| Europe | 162.900 | 160.000 | 180.000 | 2.921 | 2.486 | 3.426 | 0.001 |
| Asia | 82.552 | 80.000 | 80.000 | 2.778 | 2.379 | 3.181 | 0.001 |
| North America | 437.691 | 268.000 | 550.000 | 11.443 | 10.113 | 13.497 | 0.001 |
| **Flyway** | | | | | | | |
| AI | 10.563 | 9.345 | 11.880 | 78.911 | 76.250 | 82.111 | 0.002 |
| PS | 95.375 | 92.000 | 100.000 | 540.419 | 528.531 | 551.669 | 0.002 |
| Atlantic flyway | 41.305 | 41.000 | 43.000 | 3.288 | 2.808 | 4.048 | 0.002 |
| Mississippi flyway | 26.024 | 18.000 | 38.000 | 3.332 | 2.796 | 4.202 | 0.002 |
| central flyway | 18.707 | 11.000 | 23.000 | 3.316 | 2.792 | 4.056 | 0.002 |
| pacific flyway | 27.798 | 22.000 | 42.000 | 3.164 | 2.597 | 4.000 | 0.002 |
| **Migration** | | | | | | | |
| AI | 59.277 | 55.857 | 62.574 | 80.351 | 78.342 | 82.289 | 0.004 |
| PS | 407.488 | 397.000 | 417.000 | 510.881 | 502.818 | 519.850 | 0.004 |
| domestic | 8.310 | 8.000 | 10.000 | 3.784 | 3.278 | 4.916 | 0.004 |
| migratory | 6.544 | 6.000 | 8.000 | 4.487 | 3.924 | 5.414 | 0.004 |
| nonhuman-mammal | 3.232 | 3.000 | 5.000 | 1.667 | 1.324 | 2.030 | 0.004 |
| partially migratory | 5.656 | 5.000 | 8.000 | 2.816 | 2.378 | 3.348 | 0.004 |
| sedentary | 3.036 | 3.000 | 3.000 | 1.785 | 1.458 | 2.082 | 0.004 |
| **Host Order** | | | | | | | |
| AI | 42.505 | 39.799 | 45.411 | 61.018 | 59.291 | 62.755 | 0.010 |
| PS | 334.122 | 325.000 | 342.000 | 437.302 | 429.890 | 445.370 | 0.010 |
| Galliformes | 6.000 | 6.000 | 6.000 | 2.321 | 2.052 | 3.020 | 0.010 |
| Anseriformes | 2.592 | 2.000 | 4.000 | 2.318 | 2.058 | 3.006 | 0.010 |
| nonhuman-mammal | 4.922 | 4.000 | 6.000 | 2.306 | 2.050 | 3.012 | 0.010 |
| raptors | 4.106 | 3.000 | 6.000 | 2.321 | 2.056 | 3.018 | 0.010 |
| shorebirds | 8.008 | 8.000 | 8.000 | 2.324 | 2.054 | 3.030 | 0.010 |
| Strigiformes | 3.664 | 3.000 | 5.000 | 2.316 | 2.048 | 3.020 | 0.010 |
| Passeriformes | 10.000 | 10.000 | 10.000 | 1.847 | 1.414 | 2.156 | 0.010 |
| **domestic, wild, turkey** | | | | | | | |
| AI | 20.709 | 18.436 | 23.018 | 57.149 | 54.852 | 59.336 | 0.002 |
| PS | 144.599 | 135.000 | 155.000 | 318.740 | 313.372 | 323.543 | 0.002 |
| turkey | 9.784 | 7.000 | 15.000 | 2.631 | 2.294 | 3.097 | 0.002 |
| domestic | 11.565 | 11.000 | 14.000 | 2.613 | 2.283 | 3.123 | 0.002 |
| wild | 27.688 | 19.000 | 37.000 | 9.602 | 8.294 | 12.082 | 0.002 |
| **domestic, wild** | | | | | | | |
| AI | 15.148 | 13.665 | 16.980 | 43.950 | 41.232 | 46.378 | 0.002 |
| PS | 108.882 | 104.000 | 115.000 | 241.071 | 234.363 | 246.802 | 0.002 |
| Wild | 38.178 | 37.000 | 44.000 | 12.628 | 10.511 | 16.301 | 0.002 |
| domestic | 23.096 | 19.000 | 28.000 | 3.390 | 2.924 | 4.156 | 0.002 |
| **domestic, wild, backyard bird** | | | | | | | |
| AI | 10.854 | 9.014 | 12.775 | 30.252 | 28.751 | 31.921 | 0.010 |
| PS | 84.263 | 78.000 | 93.000 | 164.462 | 161.920 | 166.440 | 0.010 |
| backyard bird | 7.297 | 6.000 | 9.000 | 2.007 | 1.693 | 2.267 | 0.010 |
| domestic | 11.243 | 10.000 | 13.000 | 2.012 | 1.693 | 2.320 | 0.010 |
| wild | 64.840 | 63.000 | 68.000 | 16.674 | 14.143 | 20.300 | 0.010 |

Association Index (AI), Parsimony score (PS) and the maximum monophyletic clade size for each trait in the analysis are listed with their mean and 95% CI. The mean and confidence intervals for the null model are provided for each trait.

**Extended Data Table 2 | Root state probabilities for discrete traits of each analysis in the study using the discrete trait shuffling test**

| Trait | Original | Tip shuffle | Number of taxa |
|---|---|---|---|
| **Migration** | | | |
| Domestic | 0.1194 | 0.1891 | 100 |
| Nonhuman Mammal | 0.0141 | 0.2071 | 100 |
| Migratory | 0.4569 | 0.2171 | 100 |
| Part Migratory | 0.4065 | 0.1917 | 100 |
| Sedentary | 0.0031 | 0.1948 | 100 |
| **Global** | | | |
| Asia | 0.2117 | 0.0010 | 294 |
| North America | 0.6759 | 0.9980 | 1333 |
| Europe | 0.1124 | 0.0010 | 300 |
| **Flyway** | | | |
| Atlantic | 0.9503 | 0.3010 | 250 |
| Central | 0.0232 | 0.2530 | 250 |
| Mississippi | 0.0252 | 0.2470 | 250 |
| Pacific | 0.0011 | 0.1970 | 250 |
| **Domestic, Wild, Turkey (1:1:1)** | | | |
| Domestic | 0.0263 | 0.0630 | 173 |
| Turkey | 0.0113 | 0.0630 | 173 |
| Wild | 0.9623 | 0.8720 | 346 |
| **Domestic, Wild (1:1)** | | | |
| Domestic | 0.1140 | 0.5170 | 270 |
| Wild | 0.8950 | 0.4830 | 270 |
| **Domestic, Wild, Backyard bird** | | | |
| Wild | 0.9992 | 0.9190 | 193 |
| Domestic | 0.0040 | 0.0037 | 85 |
| Backyard bird | 0.0003 | 0.0043 | 85 |
| **Host orders – Equal 1** | | | |
| Galliformes | 0.1189 | 0.1309 | 100 |
| Strigiformes | 0.0290 | 0.1568 | 99 |
| Raptors | 0.1019 | 0.1838 | 100 |
| Nonhuman mammal | 0.1409 | 0.1568 | 100 |
| Shorebird | 0.1499 | 0.1548 | 100 |
| Passeriformes | 0.0130 | 0.0200 | 57 |
| Anseriformes | 0.4466 | 0.1968 | 100 |
| **Host orders – Equal 2** | | | |
| Galliformes | 0.0789 | 0.1580 | 100 |
| Strigiformes | 0.0260 | 0.1540 | 99 |
| Raptors | 0.1069 | 0.1580 | 100 |
| Nonhuman mammal | 0.1099 | 0.1780 | 100 |
| Shorebird | 0.1269 | 0.1690 | 100 |
| Passeriformes | 0.0180 | 0.0240 | 57 |
| Anseriformes | 0.5335 | 0.1540 | 100 |
| **Host orders – Equal 3** | | | |
| Galliformes | 0.0920 | 0.1620 | 100 |
| Strigiformes | 0.0421 | 0.1570 | 99 |
| Raptors | 0.1494 | 0.1550 | 100 |
| Nonhuman mammal | 0.1696 | 0.1790 | 100 |
| Shorebird | 0.0613 | 0.1560 | 100 |
| Passeriformes | 0.0192 | 0.0240 | 57 |
| Anseriformes | 0.4674 | 0.1640 | 100 |
| **Host orders – Proportional 1** | | | |
| Galliformes | 0.0030 | 0.0460 | 65 |
| Strigiformes | 0.0010 | 0.0290 | 44 |
| Raptors | 0.0759 | 0.4690 | 167 |
| Nonhuman mammal | 0.0010 | 0.0330 | 33 |
| Shorebird | 0.0529 | 0.0090 | 83 |
| Passeriformes | 0.0010 | 0.0120 | 31 |
| Anseriformes | 0.9691 | 0.3990 | 232 |
| **Host orders – Proportional 2** | | | |
| Galliformes | 0.0080 | 0.0870 | 65 |
| Strigiformes | 0.0010 | 0.0550 | 44 |
| Raptors | 0.0709 | 0.3840 | 167 |
| Nonhuman mammal | 0.0010 | 0.0940 | 33 |
| Shorebird | 0.0519 | 0.0210 | 83 |
| Passeriformes | 0.0010 | 0.0340 | 31 |
| Anseriformes | 0.9681 | 0.3210 | 232 |
| **Host orders – Proportional 3** | | | |
| Galliformes | 0.0010 | 0.0800 | 65 |
| Strigiformes | 0.0020 | 0.0430 | 44 |
| Raptors | 0.3387 | 0.3680 | 167 |
| Nonhuman mammal | 0.0709 | 0.1280 | 33 |
| Shorebird | 0.0010 | 0.0340 | 83 |
| Passeriformes | 0.0060 | 0.0250 | 31 |
| Anseriformes | 0.5904 | 0.3180 | 232 |

The root state probability before and after the shuffling test are displayed.

# Reporting Summary

## Statistics

For all statistical analyses, confirm that the following items are present in the figure legend, table legend, main text, or Methods section.

| n/a | Confirmed | |
|---|---|---|
| ☐ | ☒ | The exact sample size (*n*) for each experimental group/condition, given as a discrete number and unit of measurement |
| ☒ | ☐ | A statement on whether measurements were taken from distinct samples or whether the same sample was measured repeatedly |
| ☐ | ☒ | The statistical test(s) used AND whether they are one- or two-sided *Only common tests should be described solely by name; describe more complex techniques in the Methods section.* |
| ☐ | ☒ | A description of all covariates tested |
| ☐ | ☒ | A description of any assumptions or corrections, such as tests of normality and adjustment for multiple comparisons |
| ☐ | ☒ | A full description of the statistical parameters including central tendency (e.g. means) or other basic estimates (e.g. regression coefficient) AND variation (e.g. standard deviation) or associated estimates of uncertainty (e.g. confidence intervals) |
| ☐ | ☒ | For null hypothesis testing, the test statistic (e.g. *F*, *t*, *r*) with confidence intervals, effect sizes, degrees of freedom and *P* value noted *Give P values as exact values whenever suitable.* |
| ☐ | ☒ | For Bayesian analysis, information on the choice of priors and Markov chain Monte Carlo settings |
| ☒ | ☐ | For hierarchical and complex designs, identification of the appropriate level for tests and full reporting of outcomes |
| ☐ | ☒ | Estimates of effect sizes (e.g. Cohen's *d*, Pearson's *r*), indicating how they were calculated |

*Our web collection on statistics for biologists contains articles on many of the points above.*

## Software and code

Policy information about availability of computer code

Data collection
: We downloaded all available nucleotide sequence data and associated meta-data for the Hemagglutinin protein of all HPAI clade 2.3.4.4b H5Nx viruses from the GISAID database on 2023-11-25. We downloaded the AVONET database for avian ecology data and merged it to available host metadata from GISAID for each sequence. We used the species if provided to match the species indicated in the AVONET database. If host metadata in GISAID was defined using common name for a bird, we determined the taxonomic species name and used that for further merging with the AVONET data. Data for detections of HPAI in North America were collected from USDA APHIS. Reports for mammals, wild birds, and domestic poultry were all downloaded (download date: 2023-11-25).

Data analysis
: For each subset of the data described for further phylodynamic modeling the following process was followed. We first aligned sequences using MAFFT v7.5, sequence alignments were visually inspected using Geneious and sequences causing significant gaps were removed and nucleotides before the start codon and after the stop codon were removed. We de-duplicated identical sequences collected on the same day (retaining identical sequences that occurred on different days). We identified and removed temporal outliers for all genomic datasets by performing initial phylogenetic reconstruction in a maximum likelihood framework using IQtree v.1.6.12 and used the program TimeTree v 0.11.2 was used to remove temporal outliers. Bayesian phylogenetic reconstructions and analyses were performed using BEAST v.1.10.4. We performed Bayesian phylogenetic reconstruction for each dataset prior to discrete trait diffusion modeling to estimate a posterior set of empirical trees. The following priors and settings were used for each subset of the sequence data. We used the HKY nucleotide substitution model with gamma-distributed rate variation among sites and lognormal relaxed molecular clock model. The Bayesian SkyGrid coalescent was used with the number of grid points corresponding to the number of weeks between the earliest and latest collected sample (e.g for a dataset collected between 2021-11-04 and 2023-08-11 we would set 92 grid points). We initially ran four independent MCMC chains with a chain-length of 100 million states logging every 10000 states. We diagnosed the combined results of the independent runs diagnosed Tracer v1.7.2. to ensure adequate ESS (ESS > 200) and reasonable estimates for parameters. If ESS was inadequate additional independent MCMC runs were run increasing chain length to 150 million states, sampling every 15000 states were performed. We combined the tree files from each

independent MCMC run removing 10-30% burn-in and resampling to get a tree file with between 9000 and 10000 posterior trees using Logcombiner v1.10.4. A posterior sample of 500 trees was extracted and used as empirical tree sets in discrete trait diffusion modeling. We defined discrete traits for use in discrete trait diffusion modeling based on the available sequence metadata and merged AVONET data. For each discrete trait dataset, we used an asymmetric continuous time Markov chain discrete trait diffusion model and implemented the Bayesian stochastic search variable selection (BSSVS) to determine the most parsimonious diffusion network. We calculate the transition rate as a realization of the CTMC process by dividing the number of markov jumps by the tree height (branch length from the earliest tip to the root of the tree) , and separately, by tree length (sum of all branch lengths). For each pairwise transition rate, we calculate the level of Bayes Factor (BF) support that the given rate has.

For manuscripts utilizing custom algorithms or software that are central to the research but not yet described in published literature, software must be made available to editors and reviewers. We strongly encourage code deposition in a community repository (e.g. GitHub). See the Nature Portfolio guidelines for submitting code & software for further information.

## Data

Policy information about availability of data

All manuscripts must include a data availability statement. This statement should provide the following information, where applicable:
- Accession codes, unique identifiers, or web links for publicly available datasets
- A description of any restrictions on data availability
- For clinical datasets or third party data, please ensure that the statement adheres to our policy

All analytical scripts, metadata annotations, and BEAST XMLs used in this analysis can be found at the following GitHub repository: https://github.com/moncla-lab/North-American-HPAI

All data that was used in this analysis were sourced from public databases. Acknowledgement table for GISAID isolates used in this analysis can be found in Table S20.

Several of the analyses presented have also been publicly made available using a maximum likelihood framework through the Nextstrain pipeline and a narrative of this work can be found in the following link: https://nextstrain.org/community/narratives/moncla-lab/nextstrain-narrative-hpai-north-america@main/HPAI-in-North-America

## Research involving human participants, their data, or biological material

Policy information about studies with human participants or human data. See also policy information about sex, gender (identity/presentation), and sexual orientation and race, ethnicity and racism.

| | |
|---|---|
| Reporting on sex and gender | N/A |
| Reporting on race, ethnicity, or other socially relevant groupings | N/A |
| Population characteristics | N/A |
| Recruitment | N/A |
| Ethics oversight | N/A |

Note that full information on the approval of the study protocol must also be provided in the manuscript.

# Field-specific reporting

Please select the one below that is the best fit for your research. If you are not sure, read the appropriate sections before making your selection.

☒ Life sciences ☐ Behavioural & social sciences ☐ Ecological, evolutionary & environmental sciences

For a reference copy of the document with all sections, see nature.com/documents/nr-reporting-summary-flat.pdf

# Life sciences study design

All studies must disclose on these points even when the disclosure is negative.

Sample size: We downloaded all available nucleotide sequence data and associated meta-data for the Hemagglutinin protein of all HPAI clade 2.3.4.4b H5Nx viruses from the GISAID database on 2023-11-2589. For each subset of the data described for further phylodynamic modeling the following process was followed. We first aligned sequences using MAFFT v7.5.20, sequence alignments were visually inspected using Geneious and sequences causing significant gaps were removed and nucleotides before the start codon and after the stop codon were removed. We de-duplicated identical sequences collected on the same day (retaining identical sequences that occurred on different days). We identified and removed temporal outliers for all genomic datasets by performing initial phylogenetic reconstruction in a maximum

likelihood framework using IQtree v.1.6.12 and used the program TimeTree v 0.11.2 was used to remove temporal outliers and to assess the clockliness of the dataset prior to Bayesian phylogenetic reconstruction. This resulted in a dataset of 1824 sequences that were used in further analyses (Figure S24).

| | |
|---|---|
| Data exclusions | Temporal outliers determined by initial phylogenetic analysis were removed. We de-duplicated identical sequences collected on the same day (retaining identical sequences that occurred on different days). We identified and removed temporal outliers for all genomic datasets by performing initial phylogenetic reconstruction in a maximum likelihood framework using IQtree v.1.6.12 and used the program TimeTree v 0.11.2 was used to remove temporal outliers and to assess the clockliness of the dataset prior to Bayesian phylogenetic reconstruction Data for discrete trait diffusion models were subsampled based on availability of data for given discrete traits. |
| Replication | We performed at least three independent runs of analyses using the models and parameters described in the empirical tree analysis section above. All titration replicates were performed using an MCMC chain length of 100 million states sampling every 10,000 states. |
| Randomization | Random sub-sampling of data was performed for discrete trait datasets as well as random case proportional subsamplig for analyses of host order transmission. |
| Blinding | N/A |

# Reporting for specific materials, systems and methods

We require information from authors about some types of materials, experimental systems and methods used in many studies. Here, indicate whether each material, system or method listed is relevant to your study. If you are not sure if a list item applies to your research, read the appropriate section before selecting a response.

## Materials & experimental systems

| n/a | Involved in the study |
|---|---|
| ☒ | ☐ Antibodies |
| ☒ | ☐ Eukaryotic cell lines |
| ☒ | ☐ Palaeontology and archaeology |
| ☒ | ☐ Animals and other organisms |
| ☒ | ☐ Clinical data |
| ☒ | ☐ Dual use research of concern |
| ☒ | ☐ Plants |

## Methods

| n/a | Involved in the study |
|---|---|
| ☒ | ☐ ChIP-seq |
| ☒ | ☐ Flow cytometry |
| ☒ | ☐ MRI-based neuroimaging |

## Plants

| | |
|---|---|
| Seed stocks | N/A |
| Novel plant genotypes | N/A |
| Authentication | N/A |

