## [Peer Review file · Nature]

Ecology and spread of the North American H5N1 epizootic

Corresponding Author: Dr Louise Moncla

Version 0:

Reviewer comments:

Referee #1

(Remarks to the Author)

In this paper the authors analyze available HPAI H5N1 viruses and provide evidence for wild bird mediated spread and spillover maintaining the ongoing HPAI H5 epidemic in North America. The authors also highlight that detections in backyard poultry are earlier than detections in commercial poultry. They highlight a role for enhanced surveillance in backyard poultry to support pandemic preparedness and prevention efforts. I do have some concerns, that I have highlighted below.

1) Skyline interpretation - The proportional relationship between $Ne(t)$ and $I(t)$ has been challenged. Frost and Volz 2012 showed that $Ne(t)$ depends on transmission rates (λ) as well as prevalence ($I(t)$), challenging the assumption of proportionality. However - I am unsure if this criticism is valid towards the SkyGrid. Given the poor correlation between HPAI detections and the $\text{Log}(Ne)$ I wonder if the underlying assumption in this analysis and interpretation may be spurious

2) The molecular api described in Figure 2 and spread in Figure 3 has been shown previously, Specifically the Atlantic introduction and establishment in North America and subsequent patterns of spread (See Gass et al 2023 for example, ref 6 and 7). Multiple introductions to North America along the pacific coast have also been described (Alkie et al 2022 ref 34). The relationship linking between bird behavior (migratory to sedentary, domestic or mammal) appears to be a novel result. However, the authors should determine the impact of trait bias sampling on this result. This could be tested with a tip-trait swap test (see Edwards et al Current Biology, 2011 - also relevant to analyses presented in Figure 4,5 and 6).

3) The role of wild birds in the spread and maintenance of HPAI in this current HPAI outbreak is consistent with other observations. This has been assumed to be true. Certainly, most researchers working on influenza would believe this to be an accepted fact, and therefore, this conclusion will not be controversial. However, the speculations at Line 293-299, within the middle of the results, touches on the mechanistic questions that could be impactful on how preparedness and surveillance efforts. Since the results do not touch on this, this section belongs in the discussion.

Note - authors state "Previous work has shown that within-flyway transmission occurs far more rapidly than transmission between flyways, which may occur over longer time spans (> 5 years) (38– 40)." Reference 38 contradicts this statement.

Line 286 please correct "statically"

4) the titration analysis is an elegant test and certainly should serve as a warning against inferences made from non-representative datasets. Rather than assuming Turkeys not labeled as wild, did you consider classifying this host as ambiguous where it could be either wild or domestic? Regardless, determining that the outbreak in domestic poultry has been maintained through repeated introductions is an important result.

5) The earlier infection of backyard poultry compared to commercial poultry highlight a potential role for enhanced surveillance in this population. The use of sentinel birds is a key prevention tool. The suggestion of a backyard poultry sentinel network as an early warning is a nice conclusion from this work.

J Bahl

Referee #2

(Remarks to the Author)

This manuscript provides a comprehensive genomic analysis of the 2021–2023 HPAI H5N1 panzootic in North America, focusing on virus introductions, migration patterns, and agricultural spillovers. Using Bayesian phylodynamic analysis of HA gene sequences, the authors reveal key findings, including multiple introductions in the US, a dominant persistent lineage, rapid westward spread facilitated by wild birds (Anseriformes, shorebirds, Galliformes), numerous wild bird-to-domestic poultry spillovers, limited onward transmission within poultry, and the potential role of backyard flocks as an early warning system. These findings highlight the increasing persistence of HPAI in wild birds, emphasizing the necessity for targeted surveillance and mitigation measures at the wildlife-agriculture interface.

While offering a valuable synthesis of existing data, the manuscript requires substantial revisions to clarify its novelty, strengthen methodological justifications, improve data presentation, and accurately reflect the existing literature.

1. The justification used (both in abstract and introduction) feels outdated. While mentioning novel species infections, it underemphasizes the established understanding that current poultry outbreaks are largely driven by shifted H5N1 ecology (as seen in Eurasia/Africa and previous North/South American studies), increased wild bird adaptation, and repeated introductions. Instead of implying a need to "re-examine" outbreak causes, it might be better to focus on how these known drivers manifest in the 2021-2023 panzootic and their implications for surveillance/control.

2. While providing a strong overview, the Introduction needs clarification on several points:

- o It underemphasizes recent evidence of the increased role of wild birds in H5N1 dissemination, particularly major ecological/evolutionary changes driving global outbreaks. It would be better to align paragraph 3 with the 2020/2021 Eurasian wild bird resurgence (e.g., Ref 35 for instance).
- o Clarify lines 64-69 regarding reassortment with LPAI H5Nx. How was it determined that the gene sources were LPAI H5, and why not other LPAI subtypes?
- o The claim of altered mammalian tissue tropism (lines 66-68) requires specific genomic evidence/references. It is unclear how Ref 8 supports "altered tissue tropism." If it is a hypothesis, rephrase as such.
- o The introduction suggests culling halted the 2014–2015 epidemic but overlooks that those viruses failed to sustain in wild birds. Revise to avoid implying a causal link between poultry culling and HPAI's disappearance from North America (also see comments on Discussion below).

3. The claim that wild birds constitute an "emerging reservoir" needs careful framing. Is this a long-term HPAI reservoir, or primarily a consequence of ongoing introductions and rapid transmission? A clear definition of "reservoir" (lines 90-92), including any genomic or epidemiological criteria used, will help.

Results

4. While the authors attempt to clarify the geographic origin of the data (North America vs. U.S.), some parts are unclear. For example, in the Methods, "Data for detections of HPAI in North America were collected from USDA APHIS" implies a US focus. If the data is primarily from the US, please revise the claims to reflect this. It would be useful if the numbers from each country were explicitly stated.

5. The term "detections" is used frequently at the start. While generally understood, it would be beneficial to explicitly define what a detection constitutes in this context (lines 185-193) (e.g., a positive test result from a single animal/sample) to avoid ambiguity. Although, to be fair, the authors provide more information in a subsequent Results section (lines 462–464): "while each detection in wild birds represents a single infected animal, domestic bird detections usually represent a single infected farm, where the true..."

6. Line 233 "failed to disseminate widely". The duration of persistence (0.024-6.9 months) is quite variable. "Limited dissemination" or "localized persistence" might be more appropriate, as "failed" could imply a lack of any spread.

7. Line 249. The text refers to 'transition rates' between flyways as a "proxy for transmission." While transition rates reflect viral movement, they don't necessarily equate to successful onward transmission... acknowledge that transition rates are a proxy for, but not equivalent to, successful onward transmission.

8. Lines 323-325. Domestic ducks are missing here, which play a crucial role in facilitating outbreaks, especially in Asia.

9. An earlier study on LPAI phylogeography (<https://journals.plos.org/plospathogens/article?id=10.1371/journal.ppat.1003570>) suggested transition across flyways usually occurs via congregating sites.

Discussion

10. The Discussion extensively reiterates results, adding little new interpretation. It needs to be significantly condensed, focusing on broader implications and avoiding excessive detail already presented in the Results. The current length dilutes the key messages.

11. Lines 818–823: "Combined with our analyses showing that most agricultural outbreaks were driven by repeated introductions from wild birds, these findings suggest that infections in these domestic populations were generally transient." This is likely due to extensive culling, and these patterns are bound to change once vaccination strategies are adopted. This

needs to be acknowledged.

12. "Given backyard birds' enhanced likelihood of interaction with wild birds, and our inference of earlier spillovers in those groups, backyard bird populations could be investigated as early warning sentinels for increasing transmission in local wild bird populations." The "early warning signal" claim is weak and potentially misleading. The earlier detection in backyard birds could be due to differences in surveillance rather than a true early warning. Frame this point much more cautiously and acknowledge the limitations of the data. The Mali/Egypt comparison feels speculative and should be removed or significantly toned down. "Putative links between infected locally reared birds and human infections have been made in Mali and Egypt where backyard flocks are common, serving as an indicator for the potential for exposure to occur in North America (85,86)."

13. "Sampling bias is pervasive across viral outbreak datasets, and no modeling approach can completely overcome inherent biases in data acquisition." While acknowledging sampling bias is important, the discussion of its impact on the results is limited. Discuss how the chosen sampling regimes and experiments address these biases and what limitations remain. "Even so, an important caveat of our work is that our inferences are limited by the availability of sequence data, and results could change if future data become available."

14. The calls for future research are numerous and often generic. It would be better to focus on the most promising and feasible research directions.

(Remarks on code availability)

This repository provides the scripts and data necessary to reproduce the analyses presented in the associated publication. A preliminary assessment suggests that the code is well-structured and includes clear comments, which facilitates understanding of the workflow. Further investigation is needed to fully assess the ease of installation and execution across different computing environments. However, the presence of a README and the overall organization of the repository suggest a commitment to reproducibility and code sharing, making this a useful resource. Optionally, future work might include a more detailed documentation to further enhance usability for researchers less familiar with the specific analysis pipeline.

Referee #3

(Remarks to the Author)

The manuscript by Damodaran and colleagues presents a comprehensive evolutionary analysis of Hemagglutinin gene sequences, sourced from public databases, to document the spread of H5N1 in North American birds, both wild and domestic, and mammals. The study provides valuable insights into patterns of introduction, the role of specific bird orders in the rapid dissemination of the virus in North America, and how outbreaks in domestic birds have been seeded. This information may prove useful for efforts to prevent agricultural outbreaks. The authors thoroughly analyze the available data and the findings are generally well-presented. However, I have some concerns about specific strategies.

The Authors claim to test how virus diversity is structured according to flyways (line 245), but no hypothesis testing is eventually being performed. The discrete trait reconstruction indeed shows an appreciable degree of clustering by flyway, but no null hypothesis is being rejected, nor do we get an idea how much better the flyway hypothesis is compared to others. There are however ways for evaluating this more formally. Using metrics such as the association index (10.1128/JVI.75.23.11686-11699.2001), one could test whether the null hypothesis of random clustering by trait can be rejected, and approaches exist to apply this to a posterior set for trees (<https://doi.org/10.1016/j.meegid.2007.08.001>). To further appreciate the degree of clustering by flyway, one could use these metrics for comparison to other partitions of the data. For instance, a partition in 4 groups of equal size based on latitude could provide an interesting point of comparison. Alternatively, a generalized linear model extension of the discrete diffusion model could be used (10.1371/journal.ppat.1003932.) with a more fine-grained spatial partitioning and including predictors such as pairwise rates representing within or between flyway transitions, and a latitude-based predictor and/or a geographic distance predictor as alternatives.

The transition rate estimates (reported in Fig.3 and Fig. 4) are not very useful in my opinion. First it is not clear whether they are appropriately scaled in units per time using the overall rate scalar of the CTMC model. The scale of the estimates in both figures seems to suggest that only the relative rates are used (and the Methods do not offer further insight about this). Second, credible intervals are only shown in Fig.4 and not in Fig.3. This brings me to the most important issue; such rates are poorly informed by the data (a single discrete trait observation), and they are strongly impacted by (default) prior specification – this is clearly shown by the uncertainty represented for the estimates in Fig.4. It would be more useful to report rates based on the realizations of the CTMC process, which can be obtained by Markov jump counts divided by tree length. Furthermore, Markov jump counts would be useful to more strongly or more directly support specific claims, for example by summarizing the posterior ratio of jumps from East-to-West over jumps from West-to-East, to quantify the asymmetry, or by comparing jumps between adjacent versus non- adjacent flyways in support of transmission being more efficient between adjacent flyways. In the same way, Markov jumps could be useful to demonstrate sink-like behavior for particular host orders (e.g. for Strigiformes, Fig. 4).

I find the use of the 'titration experiments' to examine transitions between wild, backyard and commercial birds somewhat convoluted. I appreciate that such an approach can be useful to examine sensitivity to sampling bias/heterogeneity, but in

this case, hypotheses are being put forward based on the analysis of an incomplete data set that are subsequently tested based on the more complete data set. If the more complete data set (942 sequences) is deemed to be the more realistic/representative, than inference should focus on this, avoiding confusion over first analysis incomplete data.

Minor comments

Could the Authors clarify what they consider to be the trunk in the phylogeny (Fig.6A&D)? This concept has been useful for ladder-like trees with strong lineage turnover, but that is not the case here as there are many co-circulating lineages. I also do not really understand how Markov rewards along the trunk would be proxies for when transmission occurred in each group.

Line 288-290: "Quantification of the length of times that lineages persisted in each flyway showed slightly longer persistence within the Atlantic and Pacific flyways, potentially due to the habitat and species richness in each flyway allowing for greater interaction of hosts". This could also have a simple technical explanation in terms of border effects: in these two flyways, viruses can only transition to one adjacent flyway, whereas in the Central and Mississippi flyway, they can transition to two adjacent flyways.

Using metrics of association would also be helpful in demonstrating that there is sufficient viral genetic structure by host order group for performing discrete trait reconstructions (Fig. 4).

Line 204: that including -> that includes

Version 1:

Reviewer comments:

Referee #1

(Remarks to the Author)

In this revision the authors have addressed my previous concerns. My opinion on the novelty of the study remains. We have known the role of wild birds in the spread and persistence of HPAI since it was identified in bar-headed geese in 2005, since lineage 2.3 was described in 2006(?). Spillover events in Asia, the Middle East, and Europe have been linked to migratory bird populations. The current outbreak was mediated by migratory birds, and multiple outbreaks have been linked to spill-over events from wild birds. That said, this paper does provide a comprehensive analysis summarizing the multiple continental incursions and spill-overs to domestic populations and spread by wild birds.

- "If the epizootic were spread predominantly by wild, migratory birds, we reasoned that viruses sampled from the same, or neighboring flyways, should cluster together more closely than viruses sampled from non-adjacent flyways." Why? Flyways are administrative units, informed by a few species (primarily Mallards). It is questionable that they could truly be generalized to all birds within an Order (Buhnerkempe et al 2016).
 - o "Alternatively, the failure of Pacific incursions to spread onward could be explained by ecological isolation of the Pacific flyway, potentially due to land features like the Rocky Mountains." While this assumption/observation is intuitive, it should be tested
- "Together, our results suggest a critical shift in the ecology of highly pathogenic avian influenza viruses in North America." I disagree with this statement. The outbreak in 2015 seems to be atypical. In that outbreak, the introduction failed to become established in the wild bird population. In contrast, the outbreak did become entrenched in domestic animals and spread among farms. Stamping out practices controlled this outbreak in a relatively short time. For the current outbreak, the spread is more typical of avian influenza in wild birds. This has been the story since the virus became established in wild birds.
- "Our study develops multiple lines of evidence that collectively support wild birds as a critical emerging source of highly pathogenic avian influenza transmission in North America. By directly modeling transitions between host groups based on domestic/wild classification, taxonomic order, and migratory behavior, paired with geographic analyses showing strong dispersal across flyways, we show that wild birds were key drivers of the epizootic. These results imply that continuous surveillance in wild birds may now be key for viral tracking and outbreak reconstruction, with our data pointing to Anseriformes as good potential targets." 100% agree with this general conclusion. This has been stated repeatedly. Unfortunately, this has not been supported in policy. Could this point be strengthened?
- "Finally, our results provide explanations for the rapid expansion across the continent and for why culling domestic birds may no longer be sufficient for preventing outbreaks in agriculture. Instead, layered interventions, like improved biosecurity, separation of wild and domestic birds, and domestic animal vaccination may now be necessary for reducing future spillovers to agriculture, and by extension, humans" The study does not support this conclusion. No work has been done to support the effects of implementing control strategies. A statement like "culling domestic birds may no longer be sufficient for preventing outbreaks..." needs to be tempered and stated very carefully so that is not taken out of context. I would suggest a statement like "current control strategies are not sufficient" rather than calling out culling alone.

Referee #2

(Remarks to the Author)

The authors have substantially improved the manuscript. The dataset is impressive, with extensive genomic and metadata

coverage across host species and geographies. The analysis is rigorous and well-interpreted. This work provides valuable insight into the role of wild birds in the North American H5N1 epizootic, particularly in contrast to earlier outbreaks.

However, some framing issues remain. The manuscript would benefit from a clearer distinction between well-established knowledge and novel contributions. A few suggestions:

1. The concluding statement that “wild birds are an emerging source” is misleading. Wild birds have long been recognized as key vectors of HPAI. The main contribution here is the evidence for sustained transmission and repeated spillovers from wild birds into poultry during this outbreak, unlike the more limited role in 2014–15. Consider rephrasing accordingly.
2. Introduction, Lines 46–50. The claim that reassortment led to “altered neurotropism” is not well-supported by Ref. 8. Moreover, neurotropism and neuropathogenesis have been observed across a wide range of influenza viruses, including seasonal human strains, and are not unique to clade 2.3.4.4b. If retained, this claim needs stronger support and clearer wording. Please see Bauer, Lisa et al. Trends in Neurosciences, Volume 46, Issue 11, 953 - 970 The neuropathogenesis of highly pathogenic avian influenza H5Nx viruses in mammalian species including humans.
3. Lines 67–70. Consider citing Xie et al. (Ref. 17), which directly supports the point about increased transmission potential in wild birds by formally comparing clade 2.3.4.4b with other H5Nx clades.
4. The Introduction covers many topics—from viral evolution to mammalian spillover to policy implications—but this breadth sometimes obscures the paper’s central narrative. The key question seems to be: To what extent did wild birds sustain the North American H5N1 epizootic of 2021–23, and how does this differ from past outbreaks? A more focused structure around this aim would help orient readers, especially those less familiar with HPAI ecology.

Results

5. Section: “Viral sequence data capture seasonal variation of HPAI detections”. The observed epidemic waves in Figure 1 are described as “seasonal,” but it’s unclear whether these represent true seasonality or reflect episodic viral incursions. Given the relatively short timeframe (18–24 months), caution is warranted in interpreting these patterns as seasonal.
6. Section: “Highly pathogenic H5N1 was introduced multiple times from Europe and Asia” This section is mostly descriptive and could be tightened. Lines 219–222, for example, could be removed or condensed to focus on core findings. While the number of introductions is informative, its significance is not clearly framed in the Introduction, and the implications remain underexplored. Briefly clarifying why this matters, e.g., for understanding barriers to establishment or guiding surveillance priorities, would strengthen the narrative.

Discussion

7. The Discussion is generally well-structured and thoughtfully contextualizes the findings. However, some claims overstate the novelty of the work. The global role of wild birds in the dissemination of HPAI has been well established for nearly two decades. What this study adds is the scale and resolution of genomic surveillance during the North American epizootic, and the insight that repeated wild bird-to-poultry spillovers, rather than sustained poultry-to-poultry transmission, characterized the 2021–23 outbreak. This distinction should be more clearly emphasized.
8. Some broad statements, such as “a critical shift in the ecology of highly pathogenic avian influenza viruses in North America”, are difficult to interpret and may be misleading. For example, the 2014–15 clade 2.3.4.4 outbreak did not become established in wild bird populations in the U.S., whereas the current epizootic has persisted across years, suggesting a more durable presence in wild reservoirs. This contrast is a compelling finding and could be framed more clearly. However, broader claims about ecological change should be made with caution, as many aspects of clade 2.3.4.4b’s host range and persistence have already been described in prior work.
9. The practical implications of the findings could be further developed. For instance, do the observed spillover patterns suggest persistent gaps in farm-level biosecurity? Could enhanced real-time wild bird surveillance provide earlier warnings of outbreak risk? Briefly addressing such questions would strengthen the applied value of the study.
10. The final paragraph continues to largely reiterate established facts about global HPAI risk. This could be shortened or replaced with a clearer summary of the study’s main contributions and limitations. e.g., extensive genomic and metadata coverage, limited inference on drivers of transmission, and a relatively short timespan for evaluating true seasonality.
11. Consider citing recent study Signore et al. (Science Advances, 2025; <https://www.science.org/doi/10.1126/sciadv.adu4909>), which addresses related questions of clade 2.3.4.4b in North America using genomic data.

Referee #3

(Remarks to the Author)

The Authors have addressed most of my concerns in this revision and I do not have any additional comments

Dear Editor,

We greatly appreciate the reviewers' comments and suggestions on our work. We have attempted to address all suggested comments and revisions, and believe that the proposed changes have significantly improved the manuscript. We have reframed our introduction, rewritten the discussion section, and performed a series of experiments to bolster the statistical rigor of our findings that resulted in new figure panels in Figures 2-4, 12 new supplemental figures, 4 new supplemental tables, and 14 expansions to existing supplemental tables. In addition, we have revised the manuscript throughout to better articulate the novelty and impact of our results. We have directly responded to the critiques leveled by reviewer 1 in our point by point response, particularly in response to points 2 and 4. These new results provide further support for our results, and have resulted in a much stronger manuscript. We thank the reviewers for their helpful critiques.

Please find a point by point response to each reviewer comment below. Reviewer comments are organized by reviewer, with each comment numbered and italicized. Our responses are written in plain text below each comment.

Referee #1

1) Skyline interpretation - The proportional relationship between $N_e(t)$ and $I(t)$ has been challenged. Frost and Volz 2012 showed that $N_e(t)$ depends on transmission rates (λ) as well as prevalence ($I(t)$), challenging the assumption of proportionality. However - I am unsure if this criticism is valid towards the SkyGrid. Given the poor correlation between HPAI detections and the $\text{Log}(N_e)$ I wonder if the underlying assumption in this analysis and interpretation may be spurious.

We appreciate the reviewers' concern about coalescent methodology as described in Frost and Volz (2010). The authors in that paper discuss that N_e is more accurately related to the ratio of prevalence to transmission rate, rather than prevalence alone. This implies that variation in transmission intensity or population structure can decouple N_e from actual incidence. Non parametric coalescent-based models like the Skygrid estimate N_e using a flexible piecewise constant approach, which allows the model to infer the rate at which lineages coalesce in a phylogeny, which is influenced by the number of infections contributing to transmission over time (Drummond et al., 2005; Gill et al., 2013). Under certain epidemiological conditions such as relatively stable transmission rates and random mixing, N_e is expected to be roughly proportional to prevalence or incidence (Volz et al., 2013). Therefore we interpret N_e as a coarse approximation of relative trends in transmission. To reflect this we have added the following sentences and have removed the phrase "approximates incidence" at **line 207** to better reflect the concerns about direct comparison between N_e and incidence:

Lines 205-211: "To evaluate whether sequence data reflect case detections, we inferred the viral effective population size (N_e), a measure of viral genetic diversity

shown to be mathematically related to disease prevalence and the disease transmission rate²⁵. We inferred N_e using a nonparametric population model (Skygrid), which captures relative changes in genetic diversity and the variability of growth rate in the virus population over time, providing a proxy for epidemic dynamics as previously described^{25,39}.”

2) The molecular epi described in Figure 2 and spread in Figure 3 has been shown previously, Specifically the Atlantic introduction and establishment in North America and subsequent patterns of spread (See Gass et al 2023 for example, ref 6 and 7). Multiple introductions to North America along the pacific coast have also been described (Alkie et al 2022 ref 23). The relationship linking between bird behavior (migratory to sedentary, domestic or mammal) appears to be a novel result.

The reviewer is correct that the referenced papers do describe initial introduction into the Atlantic Flyway (Gass et al) and the Pacific (Alkie et al). However, while the initial introduction of the virus into the Atlantic has been described by Gass et al, we believe that we take a more robust and complete approach to fully describe the number, timing, and breadth of introductions into both flyways, an analysis which to our knowledge, is not published elsewhere. For example, Gass et al use a dataset of 185 sequences from North America and 60 sequences from Europe to describe an initial introduction from Europe. In comparison, we use 1,327 more North American sequences, 240 more European sequences and include an additional 294 from Asia, allowing us to not only recapitulate prior introductions, but also identify an additional novel 5 introductions into the Pacific. This expanded dataset allowed us to more conclusively show that the vast majority of transmission stems from one primary introduction in the Atlantic that occurred in late 2021, and that the Pacific coast receives far more viral flow than previously described. While Alkie et al described limited transmission into the Pacific, currently there are no other studies that have quantified the breath and number of incursions into the Pacific region as we have. Indeed, while Alkie et al highlight 2 Pacific introductions, our larger dataset allowed us to capture 7 independent introductions, indicating substantially higher flow into the Pacific region. Thus, while prior publications have described single introduction events, we believe that our manuscript is the first to use a large and geographically representative dataset to comprehensively measure the numbing and timings of introductions, and subsequent expansion of H5N1 in North America across both coastal flyways. These results have implications for the relative roles of surveillance on each coast for capturing novel introduction events. We believe that our manuscript therefore provides a novel and important analysis of H5N1 transmission across the continent, which has not to our knowledge yet been described.

We have additionally updated the text to better reflect the novelty of our results:

Lines 133-141: “Together, these data pinpoint wild birds as emerging sources for H5N1 virus transmission in North America, capable of rapid, and long-distance viral dispersal and repeated reseeding of outbreaks in agriculture. Our results implicate continuous surveillance in wild aquatic migratory birds (particularly Anseriformes) as critical for contextualizing outbreaks in mammals and agriculture, and suggest that viral evolution

in North America may now be increasingly governed by wild bird movement, ecology, and reassortment. Investment in interventions beyond culling that reduce interactions between domestic and wild animals may now be crucial for limiting future outbreaks in agriculture.”

Lines 229-232: “Though early case reports have described early introductions into the Atlantic and Pacific, a comprehensive and geographically representative analysis of the number and timings of introductions into North America across the continent has not been published^{6,8,40}.”

Lines 258-260: “However, we also infer 7 (median = 7, 95% HPD: (6,8)) additional introductions between February and September 2022 that nest within the diversity of viruses circulating in Asia (Figure 2B-C), 5 of which have not been previously described.”

Lines 269-275: “Though it remains unclear why this HA lineage was not detected from mid-2023 to 2024, the novel introductions documented here, and the eventual outgrowth of one of these lineages, highlights the importance of surveillance in the Pacific region for capturing viral importations. These data suggest that H5N1 viruses were introduced into North America at least 9 times, and that viral flow into the Pacific coast may be far more common than previously documented.”

3) However, the authors should determine the impact of trait bias sampling on this result. This could be tested with a tip-trait swap test (see Edwards et al Current Biology, 2011 - also relevant to analyses presented in Figure 4,5 and 6).

The reviewer brings up an important point regarding trait sampling bias, which is always an important consideration for phylodynamic analysis. In our initial development of this analysis, we attempted to control for sampling bias by analyzing the data using two, distinct sampling regimes. In the first, we included the same number of sequences for each host order, forcing differences in transition rates between groups to be driven by differences in the underlying sequences themselves rather than by sampling (since all host orders were included at the same frequency). We additionally performed an analysis where host orders were sampled proportional to the number of reported cases. For each regime, we performed 3 independent subsamplings to measure how the particular set of sequences included in the analysis impacted the results, and only report findings that were significant across the independent subsamplings. Though we reasoned that these approaches should adequately control for sampling, the tip trait swap test was a useful suggestion for quantifying these effects, even in the presence of the sampling regimes were employed. As suggested, we performed a Bayesian association of Tip states (BaTs) analysis, and developed a modified tip trait shuffle test for all discrete trait models we report in the manuscript. We report the complete results of these tests in **Tables S18 and S19**, and throughout the results sections. We have additionally added this information into the Methods (a new section called “Assessment of Sampling Bias), which additionally describes the modified tip shuffling analysis we

performed. Additional results lines and the new methods section are appended below for convenience:

Methods:

Lines 1326-1392:

Assessment of Sampling Bias

BaTs analysis

To determine if the discrete traits analyzed correlated with shared ancestry in the phylogeny, we employed tip trait association tests implemented in the Bayesian Tip-association Significance (BaTs) program v1.0⁴⁵. This program assesses the phylogenetic structure of discrete traits across viral lineages using three metrics: the association index (AI), parsimony score (PS), and (maximum monophyletic clade size (MC). The AI measures the imbalance of internal nodes of a phylogeny for a given set of traits. The PS calculates the number of state changes in the phylogeny. The MC measures the maximum number of tips belonging to a monophyletic clade for each discrete trait of interest. These metrics are calculated for the phylogeny as tips are randomly swapped to create a null distribution to compare against. Taken together these metrics quantify the degree of clustering within the phylogeny with Lower AI and PS values indicating stronger phylogenetic structure, suggesting that closely related taxa tend to share the same trait, whereas higher values indicate weaker structure and more frequent transitions between trait states. Statistical significance was assessed by comparing observed values against a null distribution generated through randomization, with p-values reported for each test. All discrete trait groupings showed evidence for clustering by trait, supporting the use of trait modeling across the tree.

Tip shuffle analysis

To assess the sensitivity of each of our discrete trait reconstructions to differences in sampling between groups, we implemented a modified version of a tip swap analysis⁵⁵. As originally developed, a tip swap analysis attempts to assess the impact of trait sampling on discrete trait measurements. An operator is implemented within the MCMC chain that randomly picks pairs of tips and swaps their trait values, thus generating a posterior set of trees among which pairs of trait assignments have been randomly swapped. The probability of each state at the root is then computed, and compared to the inferred root state probabilities in the empirical data. Because the root state probabilities in randomized datasets should primarily reflect the frequency of each trait in the analysis, empirical results that differ substantially from this null distribution are interpreted as evidence that the sequence data is informing the analysis beyond what is expected based on trait frequencies alone. Thus, traits for which the root state probability differs considerably from the root state probability in the null data are frequently interpreted as being informed by the data, rather than sampling bias. While this approach has been shown to perform well on small phylogenies^{50,77}, the strategy of swapping single pairs of tips poses challenges for larger trees. In our flyways dataset which includes ~1000 tips, we found that even with extremely high operator values (4000), the traditional tip-swap analysis resulted in a posterior set of trees in which the majority of tips (93.3%) remained assigned to their true state at least 50% of the time,

resulting in a null dataset that was only partially randomized. We believe this is due to the high number of tips in our analysis, resulting in only an extremely small fraction of tips randomized at any given step in the MCMC chain. To overcome this limitation, we instead performed a randomized tip shuffle analysis. Using the empirical set of trees inferred for each discrete trait analysis, we generated 100 null datasets in which we shuffled the trait assignments randomly across the tips. In this approach, we preserve the phylogenetic tree topology and the ratio of samples from each group, but shuffle their assignments at the tips. For each discrete trait analysis, we generated 100 distinct shuffled versions of the empirical trees, reran the analysis, and summarized the resulting posterior distribution by inferring a maximum clade credibility tree. We then computed the root state probabilities for each trait for each mcc tree, and computed the mean root state probability across all 100 replicates. This computed mean is reported in Table S19, in the column labelled “Mean root state probability across 100 datasets with randomly shuffled tip states.” Finally, we compared these null values to the root state probabilities calculated for each group in the empirical data (reported as “Root state probability in Empirical data” in Table S19). As expected, the root state probabilities inferred in the shuffled datasets are proportional to the number of sequences included for each group. For the analyses using an equal sampling regime (Migration, Flyway, Host Orders Equal, and initial titration tests), this leads to approximately equal expected root state probabilities across groups. In contrast, the root state probabilities in the empirical data generally differ significantly from expectation, suggesting that the phylogenetic results are informed by the genetic data rather than from sampling alone.

The results of BaTS analyses and tip shuffling analyses for each discrete trait in this study can be found in the supplemental material (Figure S30-S39), Table S18-19).

We have additionally updated the Results in the following lines:

Lines 361-364: “Wild migratory and partially migratory birds are inferred at the root of the tree with probabilities substantially higher than expected based on their sampling frequencies (Figure S10 and Table S19), indicating a role for these species in sustained transmission across the epizootic.”

Lines 432-439: “Finally, to measure the effects of potential sampling bias on the inferred transition rates, we performed a modified “tip shuffle” analysis. We generated 100 datasets in which the host tip assignments were randomly shuffled, re-inferred the host group at internal nodes, and infer a mean root state probability for each host across the 100 shuffled datasets. We then compared the root state probability in the empirical data to that inferred in the shuffled data as a measure of the impact of sampling on the results as previously described (see Methods for details)⁵⁵.”

Lines 472-475: “Across all replicates in both sampling regimes (6 total analyses), Anseriformes are inferred at the root 2-3 times more frequently than in null, shuffled datasets (Table S19), providing strong support for Anseriformes as critical drivers of epizootic transmission beyond what is expected from their frequency in the data.”

Lines 494-497: “Additionally, Galliformes were inferred at the root less frequently than expected for their sampling frequency (Table S19), and tended to cluster together more strongly than expected by chance ($p=0.0099$, Table S18), consistent with a more limited role in transmission that may have been confined to localized agricultural outbreaks.”

Lines 500-509: “Shorebird sequences were very highly clustered with each other ($AI=8.008$, $null=2.324$, $p=0.00999$), suggesting some degree of separation between viruses circulating in Shorebird populations, consistent with ecological partitioning of low pathogenicity avian influenza viruses in these hosts⁵⁷. Tip shuffle analyses show mixed results for the Shorebirds, indicating a lack of consistent evidence for their role in transmission relative to their sampling frequency in the dataset. These data suggest that while Anseriformes, shorebirds, and Galliformes may all have contributed to transmission events to other species, that Anseriformes were the predominant drivers of longer-term persistence and spread to other hosts in the time period analyzed.”

Lines 519-522: “Tip shuffle results for both sampling regimes indicate that raptors are generally less probable at the root than expected based on their sampling frequency, indicating that the clustering of genetic sequences supports a limited role for epizootic transmission.”

Lines 605-607: “Wild birds are inferred at the root of the tree at a far higher probability than expected from their sampling (posterior probability = 0.895 in empirical data vs. 0.482 in tip shuffled data), while domestic birds are underrepresented (Table S19).”

4) The role of wild birds in the spread and maintenance of HPAI in this current HPAI outbreak is consistent with other observations. This has been assumed to be true. Certainly, most researchers working on influenza would believe this to be an accepted fact, and therefore, this conclusion will not be controversial. However, the speculations at Line 293-299, within the middle of the results, touches on the mechanistic questions that could be impactful on how preparedness and surveillance efforts. Since the results do not touch on this, this section belongs in the discussion.

We agree with this reviewer that these conclusions are not likely to be controversial, and we have moved the specified lines to the discussion section as suggested. We have additionally added further discussion on the global trend towards wild bird transmission of clade 2.3.4.4b viruses as requested by Reviewer 2. However, we would like to note that although these results may be assumed to be true, to our knowledge, no other published work has specifically or rigorously tested the hypothesis that wild birds drove H5N1 transmission during the 2021 epizootic in North America. While earlier studies have shown early case detection in wild birds, or performed region-specific analyses with wild bird data, no other study has provided a unified set of well-controlled analyses to specifically provide evidence for wild bird transmission as a major driver of dissemination and spillover into agriculture. While perhaps not controversial, we believe that these results are important because they indicate a fundamental change in the ecology of highly pathogenic avian influenza viruses in North America. These results suggest that prior conceptions of how to surveil, predict, and prevent outbreaks may

need to shift to reflect that H5N1 risk now directly stems from transmission in migrating wildlife. We have added text throughout the manuscript to highlight the novelty of these conclusions.

Lines 72-107: “Historically, H5N1 transmission has been linked to enzootic transmission in domestic poultry^{17,18}, paired with occasional cross-continental movement by wild birds of the Anseriformes (waterfowl such as ducks and geese) and Charadriiformes (Shorebirds) orders¹⁹⁻²¹. Unlike the North American epizootic in 2014-2015, widespread culling of domestic birds has not halted detections in North America, suggesting that patterns of transmission since 2022 may be distinct from past North American epizootics. Prior work has posited that clade 2.3.4.4b viruses may be better able to infect and transmit among wild bird species, leading to persistent, seasonal circulation in European wild birds and widespread reassortment^{2,11,13}. In Europe, clade 2.3.4.4b viruses caused repeated incursions into European wild and domestic birds from 2016-2020, with a notable shift towards seasonal outbreaks and a broader range in infected wild species since 2020²². Thus, rather than acting as transient hosts that facilitate long-range movement, wild birds are now thought to play a greater role in maintaining and disseminating these viruses across Europe and Asia. However, the role of wild vs. domestic birds in driving transmission in North America has not been robustly or comprehensively studied, limiting informed enactment of surveillance and outbreak intervention strategies.

Previous genomic analysis of the United States epizootic linked outbreaks in poultry to wild birds, though the robustness of these results to differences in sampling between wild and domestic birds was not directly examined¹⁰. Other studies have reported surveillance and localized outbreak data to identify new incursions in wild bird populations^{6,7,8}, but do not address the role of different species in contributing to cross-continental epizootic spread, transmission between species, or outbreaks in agriculture. As of 2025, highly pathogenic avian influenza viruses are classified as foreign animal diseases by the United States and Canada, with outbreak control plans primarily focused on biosecurity to reduce spread between farms, and rapidly culling domestic birds^{23,24}. If the epizootic in North America reflects the changing ecology of clade 2.3.4.4b viruses towards wild bird driven transmission observed elsewhere, then surveillance, policy, and outbreak mitigation strategies may need to be fundamentally reformulated. Surveillance activities should focus on the host groups most important for viral dispersal, while control plans should be formulated to prevent incursions and spread in agriculture. Disentangling which species drove the North American epizootic, and whether transmission was driven by wild birds vs. domestic poultry, are therefore key for formulating effective surveillance and intervention strategies, but currently understudied.”

Lines 887-900: “Together, our results suggest a critical shift in the ecology of highly pathogenic avian influenza viruses in North America. As the primary source of transmission shifts from poultry to wild migratory birds, the ecology of clade 2.3.4.4b viruses in North America may now follow patterns unfolding globally, where evolution is increasingly governed by wild bird movement, ecology, and reassortment. Our results

implicate surveillance in wild Anseriforme species as potentially fruitful targets for ongoing tracking and response, and highlight the necessity of continued surveillance in wild birds for accurate outbreak reconstruction. Finally, our results provide explanations for the rapid expansion across the continent and for why culling domestic birds may no longer be sufficient for preventing outbreaks in agriculture. Instead, layered interventions, like improved biosecurity, separation of wild and domestic birds, and domestic animal vaccination may now be necessary for reducing future spillovers to agriculture, and by extension, humans.”

Lines 902-909: “Our study develops multiple lines of evidence that collectively support wild birds as a critical emerging source of highly pathogenic avian influenza transmission in North America. By directly modeling transitions between host groups based on domestic/wild classification, taxonomic order, and migratory behavior, paired with geographic analyses showing strong dispersal across flyways, we show that wild birds were key drivers of the epizootic. These results imply that continuous surveillance in wild birds may now be key for viral tracking and outbreak reconstruction, with our data pointing to Anseriformes as good potential targets.”

5) *Note - authors state "Previous work has shown that within-flyway transmission occurs far more rapidly than transmission between flyways, which may occur over longer time spans (> 5 years) (38– 40)." Reference 38 contradicts this statement.*

We have rewritten the suggested sentence as follows: “Previous work has shown that within-flyway viral population structure is more observable within short time periods (< 5 years)⁷⁵, while transmission assessed over longer time scales is more likely to span multiple flyways^{76,77}.”

6) *Line 286 please correct "statically"*

We thank the reviewer for finding this typo, and we have corrected this mistake.

7) *The titration analysis is an elegant test and certainly should serve as a warning against inferences made from non-representative datasets. Rather than assuming Turkeys not labeled as wild, did you consider classifying this host as ambiguous where it could be either wild or domestic? Regardless, determining that the outbreak in domestic poultry has been maintained through repeated introductions is an important result.*

We thank the reviewer for the compliment, and agree that it would be an interesting follow-up analysis to perform to determine if the host ambiguity would lead to a consistent result. We opted to investigate turkeys as potentially domestic, rather than perform the suggested analysis for the following reasons. First, because domestic bird sequences are interspersed heavily with wild bird sequences and do not descend from a genetically distinct lineage, we thought that the proposed analysis would be very likely insufficiently informed for accurately inferring sequences as domestic/wild. Instead, we chose to label these sequences as domestic because of strong epidemiologic data that

turkey infections in the United States are overwhelmingly driven by infections reported in domestic turkeys. More than half of all commercial detections between 2021 and 2024 (53.7%) stem from domestic turkeys, and the sequences included in these analyses were sampled during a time period in which almost 70% of commercial farms affected by HPAI were turkey farms (Patyk et al, 2023 *Frontiers in Veterinary Science*). In contrast, during that time period, only 139 infections (1.5% of all wild bird infections) were reported in wild turkeys. Because sequences directly stem from infections sampled and reported to USDA APHIS, we reasoned that the likelihood of a sequence stemming from a domestic or wild turkey should be roughly proportional to the cases reported to and by USDA APHIS from domestic/wild turkeys. While we cannot be sure about the domestic/wild status without additional metadata, we reasoned that epidemiologic information made it far likelier that turkey sequences reflected infections sampled in domestic birds than from wild birds. We have added this additional information to the manuscript in lines 650-655. We have additionally pasted the new information below:

Lines 638-650: “Commercial turkey operations have been heavily impacted during the epizootic, comprising 53.7% of all detections on commercial farms⁶². However, the presence of wild turkeys throughout North America makes categorizing turkey sequences as domestic or wild status ambiguous. 98% of all turkey sequences are not associated with metadata on domestic/wild status, and thus were excluded from the previous analysis. However, epidemiologic data provides some hint that most deposited turkey sequences likely stem from domestic outbreaks. Among case detections during the study period, only 139 were reported in wild turkeys, representing 1.5% of all wild bird detections. In contrast, commercial turkey outbreaks comprised 28.5% of all domestic detections in the study period, suggesting that unlabelled turkey sequences are most likely to have come from domestic birds. While this data is not conclusive, we performed an additional analysis to determine whether our exclusion of turkey sequences (that are likely domestic) may have biased our results.”

Referee #2 (Remarks to the Author):

1. The justification used (both in abstract and introduction) feels outdated. While mentioning novel species infections, it underemphasizes the established understanding that current poultry outbreaks are largely driven by shifted H5N1 ecology (as seen in Eurasia/Africa and previous North/South American studies), increased wild bird adaptation, and repeated introductions. Instead of implying a need to "re-examine" outbreak causes, it might be better to focus on how these known drivers manifest in the 2021-2023 panzootic and their implications for surveillance/control.

The reviewer brings up a fair point, and we agree that the introduction could be better updated to reflect these suggestions. In response, we have substantially revised the introduction to better highlight recently observed changes in H5N1 epidemiology and ecology, and to frame the North American outbreak as an extension of trends observed in Europe in the past few years. We have additionally added callouts to this information in the results and discussion. We have added additional citations for studies describing

the outbreaks in Europe, which are discussed at greater length in the discussion section. We have rewritten the following lines in the introduction:

Lines 72-109: “Historically, H5N1 transmission has been linked to enzootic transmission in domestic poultry^{17,18}, paired with occasional cross-continental movement by wild birds of the Anseriformes (waterfowl such as ducks and geese) and Charadriiformes (Shorebirds) orders^{19–21}. Unlike the North American epizootic in 2014-2015, widespread culling of domestic birds has not halted detections in North America, suggesting that patterns of transmission since 2022 may be distinct from past North American epizootics. Prior work has posited that clade 2.3.4.4b viruses may be better able to infect and transmit among wild bird species, leading to persistent, seasonal circulation in European wild birds and widespread reassortment^{2,11,13}. In Europe, clade 2.3.4.4b viruses caused repeated incursions into European wild and domestic birds from 2016-2020, with a notable shift towards seasonal outbreaks and a broader range in infected wild species since 2020²². Thus, rather than acting as transient hosts that facilitate long-range movement, wild birds are now thought to play a greater role in maintaining and disseminating these viruses across Europe and Asia. However, the role of wild vs. domestic birds in driving transmission in North America has not been robustly or comprehensively studied, limiting informed enactment of surveillance and outbreak intervention strategies.

Previous genomic analysis of the United States epizootic linked outbreaks in poultry to wild birds, though the robustness of these results to differences in sampling between wild and domestic birds was not directly examined¹⁰. Other studies have reported surveillance and localized outbreak data to identify new incursions in wild bird populations^{6,7,8}, but do not address the role of different species in contributing to cross-continental epizootic spread, transmission between species, or outbreaks in agriculture. As of 2025, highly pathogenic avian influenza viruses are classified as foreign animal diseases by the United States and Canada, with outbreak control plans primarily focused on biosecurity to reduce spread between farms, and rapidly culling domestic birds^{23,24}. If the epizootic in North America reflects the changing ecology of clade 2.3.4.4b viruses towards wild bird driven transmission observed elsewhere, then surveillance, policy, and outbreak mitigation strategies may need to be fundamentally reformulated. Surveillance activities should focus on the host groups most important for viral dispersal, while control plans should be formulated to prevent incursions and spread in agriculture. Disentangling which species drove the North American epizootic, and whether transmission was driven by wild birds vs. domestic poultry, are therefore key for formulating effective surveillance and intervention strategies, but currently understudied. Finally, while it is currently thought that cases in mammals likely stem from infections in wild birds, work to formally link infections across species has been sparse.”

We have additionally added the following sentences to the last paragraph of the introduction, lines 133-141:

Lines 133-141: “Together, these data pinpoint wild birds as emerging sources for H5N1 virus transmission in North America, capable of rapid, and long-distance viral dispersal and repeated reseeding of outbreaks in agriculture. Our results implicate continuous surveillance in wild aquatic migratory birds (particularly Anseriformes) as critical for contextualizing outbreaks in mammals and agriculture, and suggest that viral evolution in North America may now be increasingly governed by wild bird movement, ecology, and reassortment. Investment in interventions beyond culling that reduce interactions between domestic and wild animals may now be crucial for limiting future outbreaks in agriculture.”

2. *While providing a strong overview, the Introduction needs clarification on several points:*

o It underemphasizes recent evidence of the increased role of wild birds in H5N1 dissemination, particularly major ecological/evolutionary changes driving global outbreaks. It would be better to align paragraph 3 with the 2020/2021 Eurasian wild bird resurgence (e.g., Ref 35 for instance).

We agree with the comment from the reviewer and have added sentences describing the changes in Eurasian transmission of the virus. See response to point 1.

o Clarify lines 64-69 regarding reassortment with LPAI H5Nx. How was it determined that the gene sources were LPAI H5, and why not other LPAI subtypes?

The reviewer has made an important point and text was clarified to reflect that it may not necessarily reflect just LPAI H5 viruses as shown in the work by Alkie et al.

o The claim of altered mammalian tissue tropism (lines 66-68) requires specific genomic evidence/references. It is unclear how Ref 8 supports "altered tissue tropism." If it is a hypothesis, rephrase as such.

The paper cited demonstrates “viruses appear to be neurotropic in mammals, as seen from the histologic lesions and immunostaining in ferret brain tissues”. We have reworded the sentence as follows:

Lines 56-60: “These viruses were likely first introduced into North America in late 2021 by migratory birds flying across the Arctic Circle from Europe^{6,7}, after which reassortment with low-pathogenicity avian influenza (LPAI) viruses endemic to North America produced a reassortant with altered neurotropism in experimentally infected mammals⁸.”

o The introduction suggests culling halted the 2014–2015 epidemic but overlooks that those viruses failed to sustain in wild birds. Revise to avoid implying a causal link between poultry culling and HPAI's disappearance from North America (also see comments on Discussion below).

This is an important point that we agree should be clarified. We have revised the text to specifically highlight that these viruses failed to establish in wild birds, to avoid making a causing link between culling and disappearance of the virus. We have rewritten this sentence as follows:

Lines 49-52: “While this outbreak substantially impacted the agriculture industry, these viruses did not establish persistent transmission in wild birds. Following aggressive culling of domestic birds, the epizootic was extinguished and North America remained free of HPAI for years.”

We have additionally edited this portion of the Discussion:

Lines 972-974: “Because the viruses circulating in 2014/2015 did not establish in local wild bird populations, that epizootic was well-controlled by culling domestic flocks, and after culling 50.5 million domestic birds, the epizootic died out.”

Lines 979-983: “These relatively short transmission chains likely reflect improved biosecurity plans developed since 2015⁵, and far more rapid and aggressive culling of domestic birds. Compared to 2014/2015, the median time from case detection to depopulation in this epizootic ranged from 4-51 hours⁶⁹, vs. 6 days previously⁵, indicating a far more rapid culling response.”

3. The claim that wild birds constitute an "emerging reservoir" needs careful framing. Is this a long-term HPAI reservoir, or primarily a consequence of ongoing introductions and rapid transmission? A clear definition of "reservoir" (lines 90-92), including any genomic or epidemiological criteria used, will help.

This is a fair concern, and perhaps worthwhile to avoid labelling wild birds as a reservoir at this stage. We have altered the text to reflect that wild birds may be increasing sources of transmission, and now avoid labelling them as a reservoir.

Results

4. While the authors attempt to clarify the geographic origin of the data (North America vs. U.S.), some parts are unclear. For example, in the Methods, "Data for detections of HPAI in North America were collected from USDA APHIS" implies a US focus. If the data is primarily from the US, please revise the claims to reflect this. It would be useful if the numbers from each country were explicitly stated.

We agree with the reviewer and have added the number of sequences from each country in the methods section *Dataset subsampling and definition of discrete traits (lines 1172-1180)* and have revised the text to clarify how much data was included from the US vs. the rest of North America. While we included primarily data from the United States, we do include 224 sequences from Canada in our analyses. We have added the following lines to clarify the scope of the geographic analysis, and to provide additional data on case acquisition in Canada:

Lines 156-163: “Most sequence data from North America represent detections in the United States and Canada (United States: 1590, Canada: 224, Central America: 8), with a “detection” classified as a positive PCR test from a collected sample. In Canada, surveillance in wild and domestic populations are coordinated by a collaborative effort from the Canadian Food Inspection Agency, Environment Canada, the Public Health Agency of Canada, and the Canadian Wildlife Health Centre, which perform year-round surveillance in live and dead wild birds, and case investigation for suspected poultry outbreaks²⁹.”

Lines 202-205: “Sequence data sampled in North America is heavily skewed toward sequences from the United States (United States: 1590, Canada: 224, Central America: 8), and from the first 6 months of the outbreak, with 74% of all available sequences sampled from January-July 2022 (Figure S2).”

Lines 223-225: “Finally, though we retained data from Canada and Central America for all subsequent analyses, our results are likely most informative about transmission within the United States due to the heavy skewing of data towards the United States.”

5. The term "detections" is used frequently at the start. While generally understood, it would be beneficial to explicitly define what a detection constitutes in this context (lines 185-193) (e.g., a positive test result from a single animal/sample) to avoid ambiguity. Although, to be fair, the authors provide more information in a subsequent Results section (lines 462–464): “while each detection in wild birds represents a single infected animal, domestic bird detections usually represent a single infected farm, where the true...”

We have added the following definition of detection to avoid ambiguity:

Lines 156-158: “Most sequence data from North America represent detections in the United States and Canada (United States: 1590, Canada: 224, Central America: 8), with a “detection” classified as a positive PCR test from a collected sample.”

6. Line 233 “failed to disseminate widely”. The duration of persistence (0.024-6.9 months) is quite variable. “Limited dissemination” or “localized persistence” might be more appropriate, as “failed” could imply a lack of any spread.

We agree with the suggested verbiage and have changed it in the manuscript:

Lines 261-263: “These introductions represent infections sampled in Alaska, Oregon, California, Wyoming and British Columbia, Canada that showed limited dissemination and persisted for short periods of time (0.024 – 6.9 months).”

7. Line 249. The text refers to ‘transition rates’ between flyways as a “proxy for transmission.” While transition rates reflect viral movement, they don’t necessarily equate to successful onward transmission... acknowledge that transition rates are a proxy for, but not equivalent to, successful onward transmission.

We have revised the text as follows on **Lines 288-289**: "...and implemented a discrete trait diffusion model to estimate transition rates between flyways as a proxy for viral movement between regions."

8. Lines 323-325. Domestic ducks are missing here, which play a crucial role in facilitating outbreaks, especially in Asia.

This is an important point that we did indeed overlook. We have added a sentence to highlight the importance of domestic ducks in past outbreaks in Asia, but note that there were limited domestic duck detections in North America, as written below:

Lines 378-385: "Previous outbreaks of highly pathogenic H5N1 viruses have been facilitated by wild Anseriformes (waterfowl) and Charadriiformes (shorebirds), and domestic species (Galliformes and Anseriformes)^{18,49-52}, though the role of these hosts varies across outbreaks. While domestic ducks have played critical roles in bridging interactions between wild and domestic populations in past outbreaks in Asia, domestic ducks account for only 2% of all detections in the US, and the vast majority of cases in the North American panzootic have been in wild birds and Gallinaceous poultry (turkeys and chickens), potentially reflecting differences in poultry production between regions⁵³."

Discussion

10. The Discussion extensively reiterates results, adding little new interpretation. It needs to be significantly condensed, focusing on broader implications and avoiding excessive detail already presented in the Results. The current length dilutes the key messages.

Upon rereading, we agree with the reviewer that this could benefit from substantial shortening and refocusing. We have essentially rewritten the Discussion to be shorter and more succinct and to organize the flow around a few key insights we wish to highlight. We believe this now better clarifies our main points and has improved the manuscript. We appreciate the suggestion, and we hope the reviewer agrees that this version is improved.

11. Lines 818-823: "Combined with our analyses showing that most agricultural outbreaks were driven by repeated introductions from wild birds, these findings suggest that infections in these domestic populations were generally transient." This is likely due to extensive culling, and these patterns are bound to change once vaccination strategies are adopted. This needs to be acknowledged.

This is an important point that we overlooked addressing in the initial submission. We have revised this section to include the following information:

Lines 972-974: “Because the viruses circulating in 2014/2015 did not establish in local wild bird populations, that epizootic was well-controlled by culling domestic flocks, and after culling 50.5 million domestic birds, the epizootic died out.”

Lines 979-983: “These relatively short transmission chains likely reflect improved biosecurity plans developed since 2015⁵, and far more rapid and aggressive culling of domestic birds. Compared to 2014/2015, the median time from case detection to depopulation in this epizootic ranged from 4-51 hours⁶⁹, vs. 6 days previously ⁵, indicating a far more rapid culling response.”

12. "Given backyard birds' enhanced likelihood of interaction with wild birds, and our inference of earlier spillovers in those groups, backyard bird populations could be investigated as early warning sentinels for increasing transmission in local wild bird populations." The "early warning signal" claim is weak and potentially misleading. The earlier detection in backyard birds could be due to differences in surveillance rather than a true early warning. Frame this point much more cautiously and acknowledge the limitations of the data. The Mali/Egypt comparison feels speculative and should be removed or significantly toned down. "Putative links between infected locally reared birds and human infections have been made in Mali and Egypt where backyard flocks are common, serving as an indicator for the potential for exposure to occur in North America (85,86)."

We understand the reviewer's concern with the framing of our results regarding the potential of backyard birds, and have removed the Mali/Egypt comparison as suggested. In response to this concern, we have added significant detail regarding how backyard and commercial populations are surveilled in the United States, and added additional information about backyard bird populations, which we hope make our reasoning more clear. Commercial poultry populations in the United States are generally prioritized for rapid testing and surveillance before backyard populations, are more likely to be enrolled in indemnity programs (ensuring more rapid testing and culling responses), and have higher passive surveillance rates than backyard birds. These considerations suggest that our ability to detect earlier infections in backyard birds is likely conservative. Still, we acknowledge that these observations are built on a small dataset, and have added additional significant caveats to these results and conclusions. We have additionally re-focused the findings from this section to focus on the stronger conclusion that outbreaks in backyard birds and commercial poultry were generally seeded independently, which we believe is still an important conclusion. We have revised the following lines accordingly:

Lines 835-864: “Comparison of detections and sequence availability in commercial birds vs backyard birds show no apparent skewing in availability of samples for each group in that time period, suggesting that this pattern is not simply due to an abundance of earlier cases in backyard birds at that time (Figure S27). While this pattern could also arise if cases in backyard birds are systematically reported earlier than commercial poultry, data on testing turnarounds and enrollment in the US indemnity payment register suggest that this is unlikely to be the case. Commercial and backyard bird farms

have almost identical lag times between case reporting and confirmation (2.15 days for commercial birds and 2.4 days for backyard birds)⁶⁸, and commercial poultry operations generally depopulate completely within 24 hours⁶⁹, indicating testing and response in commercial operations that is efficient and slightly earlier than in backyard birds.

Comparison of the proportion of farms that received indemnity payments (as a proxy for the percentage that submit to regular testing) shows that while 511 of 168,048 commercial operations (0.3%) reported cases, that only 656 out of ~12 million backyard bird owners (0.0055%) were enrolled in the program^{64,68,70}. While complete data on the distribution of backyard bird operations within North America are sparse, these data suggest that backyard birds are likely undersampled relative to commercial poultry, with somewhat lagged testing, making the early pattern we observe compelling. Future studies utilizing expanded datasets across future epizootic waves are necessary to confirm this pattern more broadly.

These data confirm that in the first 6 months of the epizootic, outbreaks in backyard bird and commercial bird populations were generally seeded independently, with limited evidence for transmission between them. We show phylogenetic evidence that spillovers into backyard birds may have occurred slightly more frequently and earlier on average than spillovers into commercial poultry, despite epidemiologic data that testing is slower and more limited in backyard birds. Should future studies support these findings more broadly, then backyard bird populations could potentially be investigated as early warning signals for upticks in transmission. Future work will be necessary to investigate the utility of this hypothesis.”

Lines 1001-1005, Discussion: “Using a small dataset from the first 6 months of the epizootic, we find phylogenetic evidence that spillovers into backyard birds may have occurred slightly earlier and more frequently than those into commercial farms. We also show through extensive subsampling experiments that outbreaks on commercial and backyard premises were generally seeded independently, with limited viral flow between them.”

13. "Sampling bias is pervasive across viral outbreak datasets, and no modeling approach can completely overcome inherent biases in data acquisition." While acknowledging sampling bias is important, the discussion of its impact on the results is limited. Discuss how the chosen sampling regimes and experiments address these biases and what limitations remain. "Even so, an important caveat of our work is that our inferences are limited by the availability of sequence data, and results could change if future data become available."

We agree with the reviewer that this could have been more well-described, and we have added significant additional discussion in this paragraph. In response to reviewers 1 and 3, we have also performed follow-up analyses to perform tip-swap and tip shuffle analyses, and to measure Association Indices to quantify potential bias. Please see responses to reviewer 1’s point 3, and reviewer 2’s point 1. We have pasted the new Discussion section paragraph on sampling issues below for clarity:

Lines 1045-1090: “Sampling bias is pervasive across viral outbreak datasets, and no modeling approach can completely overcome inherent biases in data acquisition. In this study, we employed multiple, overlapping analyses to control for sampling biases, providing a framework for performing phylodynamic analyses in the presence of uncertain sampling. Highly pathogenic avian influenza data is rife with sampling issues. In the US, avian influenza sampling of wild birds employs 3 distinct methods (hunter harvest, sick and dead, and live birds), each with unmeasured biases. Only wild Anseriformes are sampled live or hunter harvested, while all other host groups are sampled sick or dead, likely skewing detections towards birds with dedicated rescue services and those located near humans. In domestic birds, detection depends on producer identification of illness and reporting, and subsequent testing, which likely varies across production types, locations, and premises. While we accumulated as much data as possible to contextualize how cases are sampled and sequenced in North America, it is impossible to know the true case burden in either domestic or wild birds. Because of this, we opted to investigate multiple distinct subsampling approaches (equal and proportional), report results that are consistent between them, and to employ statistical tests for trait bias to most accurately estimate transmission dynamics. Equal sampling regimes rely inherently on the diversity in sampled sequences to make inferences about sources of viral diversity, while proportional regimes bring the data in line with model assumptions that sequence numbers are proportional to cases. For analyses of host orders and wild/domestic bird transitions (via titration tests), we employ both. Beginning with an equal sampling regime, we show that the genetic diversity of viruses circulating in Anseriformes and wild birds is sufficient to infer them as source populations, patterns which remained under proportional sampling regimes. In the titration tests, we show that while the central conclusions remained true, that the precise number of transitions between wild and domestic birds depends heavily on sampling numbers, providing a clear argument for continuous surveillance in wildlife, and a warning for over-confidence in estimating the particular numbers of transitions between groups. Finally, we quantified the level of imbalance between sampled traits across each discrete trait analysis, and find limited evidence that sampling biases drove our inferences (Table S17-18). Still, these methods have important caveats. An equal subsampling regime might over-represent a truly under-impacted group, while a proportional subsampling might over-represent a more well sampled group. Though use of replicates allowed us greater confidence in our results, all phylodynamic inferences are limited by sequence data availability, and results could change if future data become available. Additional sequence data could result in inference of more introductions into North America, or altered numbers of transitions between domestic and wild birds. Our analyses only employ HA sequences, a caveat that could result in slightly different numbers of inferred transitions if full genomes were used. Accurately distinguishing between hypotheses of epizootic spread (e.g., whether agricultural outbreaks are driven by introductions from wild birds or by from farm-to-farm spread) depends on adequate sequence data from wild birds, without which transmission inference is impossible. As H5N1 viruses continue to evolve and spread globally, investment in surveillance strategies that capture circulating diversity among wild birds will likely be critical for accurately tracking viral evolution, prioritizing vaccine strains, and contextualizing new emergence events, like the recent outbreaks in dairy cattle.”

14. *The calls for future research are numerous and often generic. It would be better to focus on the most promising and feasible research directions.*

This is a fair critique! We have revised the discussion to be more judicious with our calls for future research.

Referee #3 (Remarks to the Author):

1. The Authors claim to test how virus diversity is structured according to flyways (line 245), but no hypothesis testing is eventually being performed. The discrete trait reconstruction indeed shows an appreciable degree of clustering by flyway, but no null hypothesis is being rejected, nor do we get an idea how much better the flyway hypothesis is compared to others. There are however ways for evaluating this more formally. Using metrics such as the association index (10.1128/JVI.75.23.11686-11699.2001), one could test whether the null hypothesis of random clustering by trait can be rejected, and approaches exist to apply this to a posterior set for trees (<https://doi.org/10.1016/j.meegid.2007.08.001>). To further appreciate the degree of clustering by flyway, one could use these metrics for comparison to other partitions of the data. For instance, a partition in 4 groups of equal size based on latitude could provide an interesting point of comparison. Alternatively, a generalized linear model extension of the discrete diffusion model could be used (10.1371/journal.ppat.1003932.) with a more fine-grained spatial partitioning and including predictors such as pairwise rates representing within or between flyway transitions, and a latitude-based predictor and/or a geographic distance predictor as alternatives.

We appreciate this very thoughtful analysis and response! These are excellent suggestions, and we have implemented the following new analyses in response.

1. To assess the degree with which sequences cluster by flyway, we have performed BaTs analysis and calculated the Association Index, which we find supports a strong association between flyway and the phylogenetic tree topology. These results show that sequences clustered strongly by flyway (AI =10.563, p=0.00199), grouping most closely with those sampled within the same or geographically adjacent flyway (Figure 3A, Table S18). We have added this test to the Methods, and Supplemental Table 18, and describe the results in the following lines:

Lines 289-295: “To determine whether sequences clustered more strongly by flyway than expected by chance, we calculated the Association Index (AI), a measure of how strongly a trait is associated with a phylogenetic tree⁴⁵. Finally, to determine whether movement between flyways was better supported than movement across other adjacent geographic areas, we quantified transitions between 4 North American regions stratified by latitude (see Methods for details).”

Lines 320-322: “Sequences clustered strongly by flyway (AI =10.563, p=0.00199), grouping most closely with those sampled within the same or geographically adjacent flyway (Figure 3A, Table S18).”

2. We also agree that comparison to an alternative geographic sampling scheme by latitude would be insightful. We have now performed an additional analysis in which we stratified North America into 4 categories by latitude, and generated a dataset of sequences in which we sampled the same number of sequences per latitude category. We then performed the same analysis as for the flyways, and quantified the number of jumps between categories (using Markov jumps and rewards), between adjacent and non-adjacent groups, and between Northward and Southward transitions. Our results show higher jumps from North to South than South to North, but that the ratio of these jumps is substantially less than the ratio between jumps from East to West. This analysis also showed that transitions across these latitudinal categories also showed a strong overrepresentation among transitions between adjacent vs. non-adjacent categories, further supporting transitions in all directions that are related to geographic proximity. Together, these analyses show that viral dispersal overwhelmingly occurred between adjacent geographic areas, and with a strong bias towards east to west movement. These results are now described in **Figures 3b-e, Figure 4b and d**, and Supplemental figures S7-S9. We also show the new results below for ease of comparison. We have added the following lines:

Lines 292-295: “Finally, to determine whether movement between flyways was better supported than movement across other adjacent geographic areas, we quantified transitions between 4 North American regions stratified by latitude (see Methods for details).”

Lines 315-334: “To determine whether sequences clustered by flyway, we calculated the Association Index (AI), a measure of how strongly a trait is associated with a phylogenetic tree (39). Sequences clustered strongly by flyway (AI =10.563, $p=0.00199$) grouping most closely with others sampled within the same or geographically adjacent flyway (Figure 3A). Direct quantification of the number of transitions between flyways (via “Markov jumps”) showed that transitions between adjacent flyways were ~10 times more common (mean = 239, 95% HPD: (216, 262)) than those between distant flyways (24, 95% HPD: (12,33))(Figure S7). Analysis of transitions between geographic groups stratified by latitude showed a similar trend towards transitions between adjacent (111, 95% HPD: (97, 125)) vs distant (39, 95% HPD: (31, 46)) regions, though the effect was less pronounced (2.8 times more transitions between adjacent regions). Transitions between flyways were predominantly inferred from East to West (Figure 3C, D, Table S1), with East to West jumps inferred ~4.4 times more frequently (mean = 214, 95% HPD: (196, 232)) than West to East jumps (mean = 49, 95% HPD: (38, 57)) (Figure 3D). Highly supported jump rates (BF > 100) were also inferred from South to North (92, 95% HPD: (73, 111)) and from North to South (57, 95% HPD: (44, 69)) (Figure S9, Table S2), with ~1.6 times more Northward than Southward transitions. Overall, viral dispersal was strongly associated with geographic proximity, with the strongest evidence for rapid viral movement that proceeded westward between adjacent migratory flyways.”

We have also added additional data panels to **Figure 3**, including Markov jumps and rewards in panel B, and summing of markov jumps and rewards in panel D:

2. The transition rate estimates (reported in Fig.3 and Fig. 4) are not very useful in my opinion. First it is not clear whether they are appropriately scaled in units per time using the overall rate scalar of the CTMC model. The scale of the estimates in both figures seems to suggest that only the relative rates are used (and the Methods do not offer further insight about this). Second, credible intervals are only shown in Fig.4 and not in Fig.3. This brings me to the most important issue; such rates are poorly informed by the data (a single discrete trait observation), and they are strongly impacted by (default) prior specification – this is clearly shown by the uncertainty represented for the estimates in Fig.4. It would be more useful to report rates based on the realizations of the CTMC process, which can be obtained by Markov jump counts divided by tree length. Furthermore, Markov jump counts would be useful to more strongly or more directly support specific claims, for example by summarizing the posterior ratio of jumps from East-to-West over jumps from West-to-East, to quantify the asymmetry, or by comparing jumps between adjacent versus non- adjacent flyways in support of transmission being more efficient between adjacent flyways. In the same way, Markov jumps could be useful to demonstrate sink-like behavior for particular host orders (e.g. for Strigiformes, Fig. 4).

This is an excellent suggestion, and we agree that these analyses would enhance the rigor of our conclusions. We have now performed Markov jumps and rewards analyses on the discrete trait analyses for the host orders and flyways categories as suggested. We next summarized the number of jumps between the following categories: east to west, west to east, north to south, south to north, and between all pairwise combinations

of host groups. The results (described in the response to point 1 above) show a strong association between transitions west to east, and some signal of south to north movement. Quantification of Markov jumps and rewards across host categories (now shown in **Figure 4b**) showed that the Anseriformes and Galliformes were most frequently associated with sources of transitions to other species, while non-canonical species (especially Passeriformes and Strigiformes) were substantially less likely to be inferred as sources. We have altered all inferred transition rates to be reflected as Markov jump counts throughout the results, and additionally provide the new results in Supplemental Table We additionally summarized these results with a new panel in **Figure 4 (4d)** in which we quantified the proportion of all inferred Markov jumps involving each host group in which that host was inferred as a source vs. a sink. We additionally report these new results in **Supplemental Figure S14 and Supplemental Tables S4-S11, and S17**. These results show that Anseriformes contributed the highest number of overall jumps to other species, and that canonical hosts were far more likely to act as sources. We have rewritten the flyways and host orders sections as follows:

Lines 313-352: “Tips that descend from viruses circulating in Asia (those in Figure 2B) cluster together as a basal clade inferred in the Pacific flyway (orange cluster at top of tree, posterior probability = 0.98), consistent with introduction into the West Coast. The primary introduction from Europe occurred via the Atlantic Flyway, and subsequently spread rapidly across North America (Figure 3A,C). From the inferred time of introduction in the Atlantic coast (September 9th – October 7th 2021), viruses descending from this introduction had been sampled in every other flyway within ~4.8 months, indicating markedly faster spread than observed for other avian-transmitted viruses⁴⁶. Sequences clustered strongly by flyway (AI =10.563, p=0.00199), grouping most closely with those sampled within the same or geographically adjacent flyway (Figure 3A, Table S18). Transitions (inferred as “Markov jumps”) between adjacent flyways were ~10 times more frequent (mean = 239, 95% HPD: 216, 262) than those between distant flyways (mean = 24, 95% HPD: 12, 33)(Figure 3D, Figure S7), indicating a strong signal of dissemination via geographic proximity. Transitions between geographic groups stratified by latitude showed a similar trend towards transitions between adjacent (mean = 111, 95% HPD: 97, 125) vs distant (mean = 39, 95% HPD: 31, 46) regions, though the effect was less pronounced (Figure S9, Table S2), suggesting transmission that proceed most strongly among flyway regions. Transitions were predominantly inferred from East to West (Figure 3C, D, Table S1), with East to West jumps inferred ~4.4 times more frequently (mean = 214, 95% HPD: (196, 232)) than West to East jumps (mean = 49, 95% HPD: (38, 57)) (Figure 3D), and 2.3-3.8 times more frequently than jumps along the North-South axis (Figure S9, Table S2).

We next calculated the Bayes Factor (BF) support for each between-flyway transition rate, and describe those very high statistical support (BF > 100, see Methods for details). We infer the highest and most well supported rates from the Mississippi to Central flyway (56.301 markov jumps/year, 95% HPD: 47.85, 64.33), Atlantic to Mississippi flyway (37.34 markov jumps/year, 95% HPD: 30.84, 43.065), and Central to Pacific flyway (13.127 markov jumps/year, 95% HPD: 7.975, 18.077)(Figure 3B, Figure

S8, Table S1). Though the Pacific flyway experienced the highest number of introductions during the time period analyzed, transitions from the Pacific flyway elsewhere were inferred with low magnitude and weak support, consistent with the limited transmission we infer from West to East. Indeed, only a single statistically supported rate was inferred from the Pacific flyway to the adjacent Central flyway (BF = 3, 11.236 markov jumps/year, 95% HPD: 7.975, 13.292). Quantification of the length of time that each lineage persisted in each flyway showed slightly longer persistence within the Atlantic and Pacific flyways, though the estimates were variable (Figure 3A). We speculate that this pattern could reflect higher habitat and species richness within coastal flyways⁴⁷, or that coastal flyways each only border 1 other flyway for viruses to transition to, potentially resulting in longer within-flyway persistence.”

Redacted panel A

Figure 4. Anseriformes drove outbreak transmission, while new host species represent dead-end infections. A) Bayesian phylogenetic reconstruction of n=655 sequences subsampled by host order with equal proportions of each host. The color of tips and branches represents taxonomic order, and opacity represents the posterior support for the inferred host group. Thickness of branches correspond to the number of tips descending from a given branch. B) Mean number of Markov jumps and 95% HPD from the host group on the left (labelled “From”) to the host on the right (labelled “To”)

as inferred from the combined results of three equal orders subsamples. The dot represents the mean number of Markov jumps and the black lines (whiskers) represent the 95% HPD. The corresponding bar plot shows the bayes factor support for each jump pair and color of each bar represents the “From” host. The opacity of the bar represents the Bayes factor support for the inclusion of the rate in the diffusion network. White (opacity of 0) represents any Bayes factors inferred to be less than 3, while a full color (opacity of 1) represents any Bayes factors inferred to be greater than or equal to 100. C) Results of the PACT analysis for persistence in each host order for phylogeny shown in panel A. D) For each host, we computed the proportion of Markov jumps involving that host order in which that host was inferred as a source (jump coming from that order) or as a sink (jump going to that order). The bars represent the variability across the 3 replicates of equal orders subsamples.

Lines 461-555: “The first introduction into North America is comprised of infections sampled in great black-backed gulls (large shorebirds cluster, posterior probability = 0.69), consistent with evidence of migratory gulls facilitating transmission from Europe (Figure 4A). This shorebirds cluster contains 6 sequences from harbor seals sampled from outbreaks in New England, resulting in a highly supported transition rate (BF = 537, posterior probability= 0.99) from shorebirds to non-human mammals (6.07 markov jumps/year, 95% HPD: (3.21, 8.91), aligning with suggestions that these outbreaks are linked to scavenging or environmental contamination by infected shore-birds^{1,16}. After this initial cluster of infections, multiple deep, internal nodes on the phylogeny are inferred in Anseriformes with high posterior support (0.99), indicating that Anseriformes played an important role in driving sustained transmission and dispersal across North America. Across all replicates in both sampling regimes (6 total analyses), Anseriformes are inferred at the root 2-3 times more frequently than in null, shuffled datasets (Table S19), providing strong support for Anseriformes as critical drivers of epizootic transmission beyond what is expected from their frequency in the data. We infer Anseriformes as the predominant hosts seeding infections into other species (Figures 4B and 4D, Figure S13, Table S6-11), with the highest rates to Galliformes (17.81 markov jumps/year (95% HPD: (9.27, 26.02), BF = 1691, posterior probability = 0.99) and Strigiformes (13.51 markov jumps/year, 95% HPD: 5.35, 22.87, BF = 232, posterior probability = 0.99). Aligning with speculation following mortality events in bald eagles⁵⁶, we also infer a highly supported transition rate (BF = 127, posterior probability = 0.95) from Anseriformes to Raptors, consistent with putative links between raptors and the waterfowl they predate. Each of these patterns were preserved in each independent subsample in both sampling regimes, indicating high robustness to sampling (Figure S11-12).

We also infer support for transmission from Galliformes to Anseriformes (7.86 markov jumps/year (95% HPD: (0.71, 14.97), BF = 147, posterior probability = 0.96), Strigiformes, and nonhuman mammals. In this dataset, Galliformes primarily represent domesticated poultry (98% of sequences), suggesting that transmission from domestic birds back to wild birds and mammals may also have occurred, a hypothesis we investigate in more depth below. However, lineages in Galliformes tended to be short-lived, persisting for 0.26 years on average (95% HPD: 0.07, 0.33 years). Additionally,

Galliformes were inferred at the root less frequently than expected for their sampling frequency (Table S19), and tended to cluster together more strongly than expected by chance ($p=0.0099$, Table S18), consistent with a more limited role in transmission that may have been confined to localized agricultural outbreaks. In contrast, viral lineages persisted for the longest in Anseriformes (mean persistence time = 0.71 years, 95% HPD: 0.42, 0.88 years) (Figure 4C), and Shorebirds (mean persistence time = 0.654 years, 95% HPD: 0.18, 1.04 years). Shorebird sequences were very highly clustered with each other ($AI=8.008$, $null=2.324$, $p=0.00999$), suggesting some degree of separation between viruses circulating in Shorebird populations, consistent with ecological partitioning of low pathogenicity avian influenza viruses in these hosts⁵⁷. Tip shuffle analyses show mixed results for the Shorebirds, indicating a lack of consistent evidence for their role in transmission relative to their sampling frequency in the dataset. These data suggest that while Anseriformes, shorebirds, and Galliformes may all have contributed to transmission events to other species, that Anseriformes were the predominant drivers of longer-term persistence and spread to other hosts in the time period analyzed.

One surprising result was inference of raptors as a low-frequency, but statistically well supported source population to Anseriformes (5.18 markov jumps/year (95%HPD: (0.36, 9.27), $BF = 39$, posterior probability = 0.87). Previous characterizations of HPAI in Raptors during the 2014/2015 outbreak in North America showed mortality events and neurological symptoms in wild raptors⁵⁸, while serological evidence of infections in bald eagles have indicated exposure to influenza A viruses in 5% of birds tested between 2006 and 2010⁵⁹. In the ongoing panzootic, raptors represent the third most prevalent group in wild bird detections in Europe (12% of detections) and second most detected group in North America (20.3%)^{13,60}. Tip shuffle results for both sampling regimes indicate that raptors are generally less probable at the root than expected based on their sampling frequency, indicating that the clustering of genetic sequences supports a limited role for epizootic transmission. Thus, while raptors may have been both heavily impacted and sampled at a high frequency, these data suggest that raptors were unlikely key drivers of epizootic spread. Future work to better establish the reasons for such high case numbers among raptors and to investigate their potential links to Anseriformes will be necessary for formulating wildlife management strategies.

We found limited support for non-canonical host groups (songbirds, owls, and nonhuman mammals) in seeding infections in other species. Passeriformes (songbirds), Strigiformes (owls), and mammals each primarily served as sinks for viral diversity (Figure 4B,D), with transitions inferred with low magnitude and weak support (Figure 4B). Summing the number of jumps originating from wild canonical (Anseriformes, shorebirds), wild noncanonical (Passeriformes, Strigiformes, raptors, mammals), and Galliforme (domestic) hosts confirm that noncanonical hosts primarily acted sinks that were far likelier to receive virus than propagate it onward (Figure S14), supporting short, terminal transmission chains that did not lead to long-term persistence (Figure 4C and Figure S15). Mammal sequences cluster across the entire diversity of the phylogeny (Figure 4A) and are not associated with one particular cluster of viruses, indicating that mammal infections were not confined to a particular viral lineage, supporting very short

persistence times of 0.22 years (95%HPD:(0.088, 0.328)), and only one strongly supported transition rate to Anseriformes (BF = 53, posterior probability = 0.89). Instead, these findings are most compatible with a model in which wild mammals and other non-canonical species are infected by direct interaction with wild birds, likely related to scavenging and predation behavior¹⁴.

Taken together, these data suggest that despite high case numbers in several unusual wild hosts, non-canonical species generally played minor roles in transmission across the continent or to other species. Instead, epizootic transmission was most strongly supported in Anseriformes. We infer Anseriformes as predominant drivers of longer-term persistence and spread to other hosts across multiple independent analyses using distinct sampling regimes, supporting surveillance in these species for capturing trends in viral diversity and spread. Our data also suggest some role for shorebirds in supporting persistent viral transmission. Future work to disentangle the relationship between viral spread in these two key host groups may further resolve their utilities for surveillance and monitoring.”

3. I find the use of the ‘titration experiments’ to examine transitions between wild, backyard and commercial birds somewhat convoluted. I appreciate that such an approach can be useful to examine sensitivity to sampling bias/heterogeneity, but in this case, hypotheses are being put forward based on the analysis of an incomplete data set that are subsequently tested based on the more complete data set. If the more complete data set (942 sequences) is deemed to be the more realistic/representative, then inference should focus on this, avoiding confusion over first analysis incomplete data.

We appreciate this point, and agree that it is fair. The description of the experiments in the manuscript reflect the true order in which they were conceived, which we hoped would provide a clear and linear rationale for the experiments. We had originally intended to use an equal sampling regime, as is common within the field, as a fair way of assessing transmission between these three groups. We have found that many people are curious about how to choose a sampling regime, particularly in cases where the true underlying case count is unknown, and we felt that this was a reasonable starting place for answering this question. However, this equal sampling regime gave rise to a whole new set of questions, namely, whether the relationship between commercial and backyard birds was real or an artifact of sampling. The subsequent titration includes heavy over-representation from wild birds, with the explicit goal of determining whether commercial and backyard bird outbreaks were linked. The initial pattern observed in the first titration was what guided the additional experiments, which we believe highlights a pitfall of using equal sampling for testing particular hypotheses. Given that Reviewer 1 liked the titration tests, we have opted to keep them in the manuscript, but have added additional information about the rationale and interpretation to the text. We hope that this clarifies the rationale, but are open to further modifying if necessary. We have added the following information below:

Lines 584-592: “The goal of this analysis was two-fold. First, we aimed to determine whether either domestic or wild birds would be inferred as the primary source population, and whether that inference would be robust to our choice in sampling regime. Second, by titrating in sequences at varying degrees, we hoped to assess whether the inferred number of transitions between hosts stabilized at a certain ratio. If inferred transitions increase linearly with sequences and never stabilize, this would indicate that more surveillance data is necessary. If not, this provides evidence that adding additional data does not result in altered results, suggesting that the currently available data may be sufficient for estimating dynamics within this time period.”

Minor comments

1. *Could the Authors clarify what they consider to be the trunk in the phylogeny (Fig.6A&D)? This concept has been useful for ladder-like trees with strong lineage turnover, but that is not the case here as there are many co-circulating lineages. I also do not really understand how Markov rewards along the trunk would be proxies for when transmission occurred in each group.*

We agree that this might be confusing and have thus reworded as follows:

Lines 819-823: The Markov Reward proportion is calculated as the proportion of the phylogeny at a given time being a given discrete state. By looking at the proportion a given state is over time across the phylogeny we can provide a proxy for how long transmission has occurred in each group between transition events.

2. *Line 288-290: “Quantification of the length of times that lineages persisted in each flyway showed slightly longer persistence within the Atlantic and Pacific flyways, potentially due to the habitat and species richness in each flyway allowing for greater interaction of hosts”. This could also have a simple technical explanation in terms of border effects: in these two flyways, viruses can only transition to one adjacent flyway, whereas in the Central and Mississippi flyway, they can transition to two adjacent flyways.*

We agree this is possible and have added that as an additional possibility in the text.

Lines 350-352: “We speculate that this pattern could reflect higher habitat and species richness within coastal flyways⁴⁷, or that coastal flyways each only border 1 other flyway for viruses to transition to, potentially resulting in longer within-flyway persistence.”

3. *Using metrics of association would also be helpful in demonstrating that there is sufficient viral genetic structure by host order group for performing discrete trait reconstructions (Fig. 4).*

We agree that these metrics would be very important and have performed a BaTs analysis to address this concern for each of the discrete trait datasets in this paper. We have added the results of all association tests in **Table S18**, and pasted the results

below for clarity. We have additionally added the following line to the Method section, and call out specific Association test results throughout the Results:

Lines 1329-1345: “BaTs analysis

To determine if the discrete traits analyzed correlated with shared ancestry in the phylogeny, we employed tip trait association tests implemented in the Bayesian Tip-association Significance (BaTs) program v1.0⁴⁵. This program assesses the phylogenetic structure of discrete traits across viral lineages using three metrics: the association index (AI), parsimony score (PS), and (maximum monophyletic clade size (MC). The AI measures the imbalance of internal nodes of a phylogeny for a given set of traits. The PS calculates the number of state changes in the phylogeny. The MC measures the maximum number of tips belonging to a monophyletic clade for each discrete trait of interest. These metrics are calculated for the phylogeny as tips are randomly swapped to create a null distribution to compare against. Taken together these metrics quantify the degree of clustering within the phylogeny with Lower AI and PS values indicating stronger phylogenetic structure, suggesting that closely related taxa tend to share the same trait, whereas higher values indicate weaker structure and more frequent transitions between trait states. Statistical significance was assessed by comparing observed values against a null distribution generated through randomization, with p-values reported for each test. All discrete trait groupings showed evidence for clustering by trait, supporting the use of trait modeling across the tree.”

Results of BaTs analysis:

Analysis	Statistic	observed mean	lower 95% CI	upper 95% CU	null mean	lower 95% CI	upper 95% CI	significance
Geographic introduction	AI	2.221	1.585	2.946	105.122	101.926	108.689	0.000599
	PS	19.596	19.000	21.000	550.334	544.913	556.562	0.000599
	Europe	162.900	160.000	180.000	2.921	2.486	3.426	0.000599
	Asia	82.552	80.000	80.000	2.778	2.379	3.181	0.000599
	North America	437.691	268.000	550.000	11.443	10.113	13.497	0.000599
Flyway	AI	10.563	9.345	11.880	78.911	76.250	82.111	0.00199
	PS	95.375	92.000	100.000	540.419	528.531	551.669	0.00199

	Atlantic flyway	41.305	41.000	43.000	3.288	2.808	4.048	0.00199
	Mississippi flyway	26.024	18.000	38.000	3.332	2.796	4.202	0.00199
	central flyway	18.707	11.000	23.000	3.316	2.792	4.056	0.00199
	pacific flyway	27.798	22.000	42.000	3.164	2.597	4.000	0.00199
Migration	AI	59.277	55.857	62.574	80.351	78.342	82.289	0.00399
	PS	407.488	397.000	417.000	510.881	502.818	519.850	0.00399
	domestic	8.310	8.000	10.000	3.784	3.278	4.916	0.00399
	migratory	6.544	6.000	8.000	4.487	3.924	5.414	0.00399
	nonhuman-mammal	3.232	3.000	5.000	1.667	1.324	2.030	0.00399
	partially migratory	5.656	5.000	8.000	2.816	2.378	3.348	0.00399
	sedentary	3.036	3.000	3.000	1.785	1.458	2.082	0.00399
Host Order	AI	42.505	39.799	45.411	61.018	59.291	62.755	0.00999
	PS	334.122	325.000	342.000	437.302	429.890	445.370	0.00999
	Galliformes	6.000	6.000	6.000	2.321	2.052	3.020	0.00999
	Anseriformes	2.592	2.000	4.000	2.318	2.058	3.006	0.00999
	nonhuman-mammal	4.922	4.000	6.000	2.306	2.050	3.012	0.00999

	raptors	4.106	3.000	6.000	2.321	2.056	3.018	0.00999
	shorebirds	8.008	8.000	8.000	2.324	2.054	3.030	0.00999
	Strigiformes	3.664	3.000	5.000	2.316	2.048	3.020	0.00999
	Passeriformes	10.000	10.000	10.000	1.847	1.414	2.156	0.00999
domestic wild turkey	AI	20.709	18.436	23.018	57.149	54.852	59.336	0.00199
	PS	144.599	135.000	155.000	318.740	313.372	323.543	0.00199
	turkey	9.784	7.000	15.000	2.631	2.294	3.097	0.00199
	domestic	11.565	11.000	14.000	2.613	2.283	3.123	0.00199
	wild	27.688	19.000	37.000	9.602	8.294	12.082	0.00199
domestic wild	AI	15.148	13.665	16.980	43.950	41.232	46.378	0.00199
	PS	108.882	104.000	115.000	241.071	234.363	246.802	0.00199
	Wild	38.178	37.000	44.000	12.628	10.511	16.301	0.00199
	domestic	23.096	19.000	28.000	3.390	2.924	4.156	0.00199
domestic wild backyard bird	AI	10.854	9.014	12.775	30.252	28.751	31.921	0.00999
	PS	84.263	78.000	93.000	164.462	161.920	166.440	0.00999
	backyard bird	7.297	6.000	9.000	2.007	1.693	2.267	0.00999
	domestic	11.243	10.000	13.000	2.012	1.693	2.320	0.00999

	wild	64.840	63.00 0	68.000	16.6 74	14.14 3	20.30 0	0.009 99
--	------	--------	------------	--------	------------	------------	------------	-------------

4. Line 204: *that including* -> *that includes*

Thank you! We have changed this.

We thank the reviewers for their helpful comments. We have addressed the remaining comments below. Italic text indicates reviewer comments, and regular text indicates our response.

Referee #1 (Remarks to the Author):

In this revision the authors have addressed my previous concerns. My opinion on the novelty of the study remains. We have known the role of wild birds in the spread and persistence of HPAI since it was identified in bar-headed geese in 2005, since lineage 2.3 was described in 2006(?). Spillover events in Asia, the Middle East, and Europe have been linked to migratory bird populations. The current outbreak was mediated by migratory birds, and multiple outbreaks have been linked to spill-over events from wild birds. That said, this paper does provide a comprehensive analysis summarizing the multiple continental incursions and spill-overs to domestic populations and spread by wild birds.

• *“If the epizootic were spread predominantly by wild, migratory birds, we reasoned that viruses sampled from the same, or neighboring flyways, should cluster together more closely than viruses sampled from non-adjacent flyways.” Why? Flyways are administrative units, informed by a few species (primarily Mallards). It is questionable that they could truly be generalized to all birds within an Order (Buhnerkempe et al 2016).*

We have removed this sentence from the Results.

o *“Alternatively, the failure of Pacific incursions to spread onward could be explained by ecological isolation of the Pacific flyway, potentially due to land features like the Rocky Mountains.” While this assumption/observation is intuitive, it should be tested*

We have removed this line from the Results, and instead placed this text into the Discussion on **lines 444-454**: “We infer 5 previously undescribed incursions⁶⁰ into the Pacific that mostly persisted transiently, suggesting frequent viral flow between Asia and the Pacific coast of North America. Limited transmission from the Pacific flyway could be explained by differential fitness of the lineages introduced into the Pacific vs. Atlantic flyways, ecological isolation of the Pacific flyway⁶¹⁻⁶³, differences in host distributions at the locations and times of these incursions, or simply due to chance. While future work is necessary to differentiate among these hypotheses, these data support the Pacific coast as an important region for capturing viral transmission between Asia and North America.” We hope this clarifies that we are speculating, and that future work is required to test this hypothesis.

• *“Together, our results suggest a critical shift in the ecology of highly pathogenic avian influenza viruses in North America.” I disagree with this statement. The outbreak in 2015 seems to be atypical. In that outbreak, the introduction failed to become established in the wild bird population. In contrast, the outbreak did become entrenched in domestic animals and spread among farms. Stamping out practices controlled this outbreak in a relatively short time. For the*

current outbreak, the spread is more typical of avian influenza in wild birds. This has been the story since the virus became established in wild birds.

We have reworked the Discussion to reframe according to this suggestion and those from Reviewer 2. See example Discussion reframing sections below:

Lines 425-439: “Our study collectively supports wild birds as critical sources of the North American H5N1 epizootic. By directly modeling transitions between host groups based on domestic/wild classification, taxonomic order, and migratory behavior, paired with strong dispersal across flyways, we show that wild birds were key drivers of epizootic transmission and introductions into agriculture. These results imply that continuous surveillance in wild birds, particularly Anseriformes⁵⁶, may now be critical for viral tracking and outbreak reconstruction. As the primary source of transmission shifts from poultry to wild migratory birds, the ecology of clade 2.3.4.4b viruses in North America may now follow patterns unfolding globally, where evolution is increasingly governed by wild bird movement, ecology, and reassortment. Recent modeling of HPAI risk in Europe identified *Anatinae* and *Anserinae* Anseriformes prevalence as consistent predictors of HPAI detection⁷², supporting wildlife surveillance for outbreak forecasting and risk assessment. Future work investigating the utility of real-time tracking of wild bird abundance and movement for forecasting outbreaks may be fruitful for formulating new approaches to prevention.”

Lines 505-516: “Taken together, we show that wild birds played the central role in dispersal of the 2021-2023 H5N1 epizootic. Transmission in wild birds provides an explanation for the rapid cross-continental spread and continued agricultural outbreaks despite aggressive culling. Our results highlight the utility of wild bird surveillance for accurately distinguishing hypotheses of epizootic spread, and suggest continuous surveillance as critical for preventing and dissecting future outbreaks. Our data underscore that continued establishment of H5N1 in North American wildlife may necessitate a shift in risk management and mitigation, with interventions focused on reducing risk within the context of enzootic circulation in wild birds. At the time of writing, outbreaks in dairy cattle highlight the critical importance of modeling ecological interactions that drive spillovers between wildlife and domestic production to inform biosecurity, outbreak response, and vaccine strain selection.”

• *“Our study develops multiple lines of evidence that collectively support wild birds as a critical emerging source of highly pathogenic avian influenza transmission in North America. By directly modeling transitions between host groups based on domestic/wild classification, taxonomic order, and migratory behavior, paired with geographic analyses showing strong dispersal across flyways, we show that wild birds were key drivers of the epizootic. These results imply that continuous surveillance in wild birds may now be key for viral tracking and outbreak reconstruction, with our data pointing to Anseriformes as good potential targets.” 100% agree with this general conclusion. This has been stated repeatedly. Unfortunately, this has not been supported in policy. Could this point be strengthened?*

Yes! We have made sure to highlight the discrepancy for the need for improved surveillance and current policy. These amendments have been added throughout the Discussion, but particularly in the sections below:

Lines 429-439: “These results imply that continuous surveillance in wild birds, particularly Anseriformes⁵⁶, may now be critical for viral tracking and outbreak reconstruction. As the primary source of transmission shifts from poultry to wild migratory birds, the ecology of clade 2.3.4.4b viruses in North America may now follow patterns unfolding globally, where evolution is increasingly governed by wild bird movement, ecology, and reassortment. Recent modeling of HPAI risk in Europe identified *Anatinae* and *Anserinae* Anseriformes prevalence as consistent predictors of HPAI detection⁷², supporting wildlife surveillance for outbreak forecasting and risk assessment. Future work investigating the utility of real-time tracking of wild bird abundance and movement for forecasting outbreaks may be fruitful for formulating new approaches to prevention.”

Lines 455-474: “We find that outbreaks in agriculture were seeded by repeated introductions from wild birds, a pattern that held true regardless of sampling regime, and that aligns with global observations that clade 2.3.4.4b viruses are increasingly spread by wild birds^{63,64}. These findings contrast with the epizootic in 2014/2015, in which a small number of introductions spread efficiently between commercial poultry operations^{2,12}. Because the viruses circulating in 2014/2015 did not establish in local wild bird populations, that epizootic subsided following aggressive culling. Since 2014/2015, biosecurity plans have improved^{12,47} and depopulation occurs more rapidly⁵⁴⁻¹², potentially contributing to the shorter domestic persistence and limited transmission back to wild birds we observe. Despite these improvements, efficient transmission in wild birds likely allowed for rapid dispersal and continuous outbreak re-seeding, making this epizootic far more challenging to control. US and Canadian policy currently classify H5N1 as a foreign animal disease, meaning that biosecurity to reduce spread between farms and rapid culling^{65,66} are prioritized for outbreak control. Though these control measures will likely remain important, our results suggest that reducing future spillovers into agriculture may now necessitate changes in management priorities. The repeated spillovers we identify suggest that gaps in farm biosecurity remain that could be enhanced to reduce outbreak risk. Finally, layered approaches, including enhanced wild bird monitoring, novel methods to separate wild and domestic birds, and potentially domestic animal vaccination, may necessitate exploration.”

• *“Finally, our results provide explanations for the rapid expansion across the continent and for why culling domestic birds may no longer be sufficient for preventing outbreaks in agriculture. Instead, layered interventions, like improved biosecurity, separation of wild and domestic birds, and domestic animal vaccination may now be necessary for reducing future spillovers to agriculture, and by extension, humans” The study does not support this conclusion. No work has been done to support the effects of implementing control strategies. A statement like “culling domestic birds may no longer be sufficient for preventing outbreaks...” needs to be tempered and stated very carefully so that is not taken out of context. I would suggest a statement like “current control strategies are not sufficient” rather than calling out culling alone.*

We agree that this is perhaps an over-interpretation. We have amended the text in the Discussion as follows:

Lines 455-474: “We find that outbreaks in agriculture were seeded by repeated introductions from wild birds, a pattern that held true regardless of sampling regime, and that aligns with global observations that clade 2.3.4.4b viruses are increasingly spread by wild birds^{63,64}. These findings contrast with the epizootic in 2014/2015, in which a small number of introductions spread efficiently between commercial poultry operations^{2,12}. Because the viruses circulating in 2014/2015 did not establish in local wild bird populations, that epizootic subsided following aggressive culling. Since 2014/2015, biosecurity plans have improved^{12,47} and depopulation occurs more rapidly⁵⁴⁻¹², potentially contributing to the shorter domestic persistence and limited transmission back to wild birds we observe. Despite these improvements, efficient transmission in wild birds likely allowed for rapid dispersal and continuous outbreak re-seeding, making this epizootic far more challenging to control. US and Canadian policy currently classify H5N1 as a foreign animal disease, meaning that biosecurity to reduce spread between farms and rapid culling^{65,66} are prioritized for outbreak control. Though these control measures will likely remain important, our results suggest that reducing future spillovers into agriculture may now necessitate changes in management priorities. The repeated spillovers we identify suggest that gaps in farm biosecurity remain that could be enhanced to reduce outbreak risk. Finally, layered approaches, including enhanced wild bird monitoring, novel methods to separate wild and domestic birds, and potentially domestic animal vaccination, may necessitate exploration.”

Lines 506-513: “Transmission in wild birds provides an explanation for the rapid cross-continental spread and continued agricultural outbreaks despite aggressive culling. Our results highlight the utility of wild bird surveillance for accurately distinguishing hypotheses of epizootic spread, and suggest continuous surveillance as critical for preventing and dissecting future outbreaks. Our data underscore that continued establishment of H5N1 in North American wildlife may necessitate a shift in risk management and mitigation, with interventions focused on reducing risk within the context of enzootic circulation in wild birds.”

Referee #2 (Remarks to the Author):

The authors have substantially improved the manuscript. The dataset is impressive, with extensive genomic and metadata coverage across host species and geographies. The analysis is rigorous and well-interpreted. This work provides valuable insight into the role of wild birds in the North American H5N1 epizootic, particularly in contrast to earlier outbreaks.

However, some framing issues remain. The manuscript would benefit from a clearer distinction between well-established knowledge and novel contributions. A few suggestions:

1. The concluding statement that “wild birds are an emerging source” is misleading. Wild birds have long been recognized as key vectors of HPAI. The main contribution here is the evidence

for sustained transmission and repeated spillovers from wild birds into poultry during this outbreak, unlike the more limited role in 2014–15. Consider rephrasing accordingly.

We have amended our writing to remove this phrase, and the reframe accordingly:

Abstract: “We pinpoint wild birds as critical drivers of the epizootic, implying enhanced surveillance in wild birds and strategies that reduce transmission at the wild agriculture interface as key for future tracking and outbreak prevention.”

Introduction, Lines 59-65: “We used Bayesian phylogeographic approaches to trace the introduction and spread of highly pathogenic H5N1 viruses during the first 18 months in North America. We identify multiple incursions into the continent and subsequent spread by wild, migrating birds that drove repeated introductions into agriculture. These data pinpoint wild birds as important drivers of epizootic spread, and implicate enhanced wildlife surveillance and interventions at the wild domestic interface as key for future viral tracking and spillover prevention.”

Discussion, Lines 425-439: “Our study collectively supports wild birds as critical sources of the North American H5N1 epizootic. By directly modeling transitions between host groups based on domestic/wild classification, taxonomic order, and migratory behavior, paired with strong dispersal across flyways, we show that wild birds were key drivers of epizootic transmission and introductions into agriculture. These results imply that continuous surveillance in wild birds, particularly Anseriformes⁵⁶, may now be critical for viral tracking and outbreak reconstruction. As the primary source of transmission shifts from poultry to wild migratory birds, the ecology of clade 2.3.4.4b viruses in North America may now follow patterns unfolding globally, where evolution is increasingly governed by wild bird movement, ecology, and reassortment. Recent modeling of HPAI risk in Europe identified *Anatinae* and *Anserinae* Anseriformes prevalence as consistent predictors of HPAI detection⁷², supporting wildlife surveillance for outbreak forecasting and risk assessment. Future work investigating the utility of real-time tracking of wild bird abundance and movement for forecasting outbreaks may be fruitful for formulating new approaches to prevention.”

Discussion, Lines 505-516: “Taken together, we show that wild birds played the central role in dispersal of the 2021-2023 H5N1 epizootic. Transmission in wild birds provides an explanation for the rapid cross-continental spread and continued agricultural outbreaks despite aggressive culling. Our results highlight the utility of wild bird surveillance for accurately distinguishing hypotheses of epizootic spread, and suggest continuous surveillance as critical for preventing and dissecting future outbreaks. Our data underscore that continued establishment of H5N1 in North American wildlife may necessitate a shift in risk management and mitigation, with interventions focused on reducing risk within the context of enzootic circulation in wild birds. At the time of writing, outbreaks in dairy cattle highlight the critical importance of modeling ecological interactions that drive spillovers between wildlife and domestic production to inform biosecurity, outbreak response, and vaccine strain selection.”

2. *Introduction, Lines 46–50. The claim that reassortment led to “altered neurotropism” is not well-supported by Ref. 8. Moreover, neurotropism and neuropathogenesis have been observed across a wide range of influenza viruses, including seasonal human strains, and are not unique to clade 2.3.4.4b. If retained, this claim needs stronger support and clearer wording. Please see Bauer, Lisa et al. Trends in Neurosciences, Volume 46, Issue 11, 953 - 970 The neuropathogenesis of highly pathogenic avian influenza H5Nx viruses in mammalian species including humans.*

We have removed this sentence from the Introduction to avoid confusion.

3. *Lines 67–70. Consider citing Xie et al. (Ref. 17), which directly supports the point about increased transmission potential in wild birds by formally comparing clade 2.3.4.4b with other H5Nx clades.*

We have made sure to cite this paper, which is now cited in the following lines:

Lines 29-33: “Since emerging in 1996, highly pathogenic H5N1 viruses of the A/goose/Guangdong lineage have spread globally via enzootic transmission in domestic poultry in Asia and Africa, paired with occasional cross-continental movement by wild birds of the Anseriformes (ducks, geese, swans) and Charadriiformes (shorebirds) orders³⁻⁹.”

Lines 47-50: “In Europe, clade 2.3.4.4b virus incursions into wild and domestic birds has led to seasonal outbreaks²³, frequent reassortment²⁴, and a broader range of affected wild bird species since 2020, and recent analyses suggest that wild birds may now play a greater role in global viral maintenance and dissemination^{8,25}.”

4. *The Introduction covers many topics—from viral evolution to mammalian spillover to policy implications—but this breadth sometimes obscures the paper’s central narrative. The key question seems to be: To what extent did wild birds sustain the North American H5N1 epizootic of 2021–23, and how does this differ from past outbreaks? A more focused structure around this aim would help orient readers, especially those less familiar with HPAI ecology.*

We have substantially shortened the Introduction, which now focuses on this question. We hope the Editor and reviewers agree.

Results

5. *Section: “Viral sequence data capture seasonal variation of HPAI detections”. The observed epidemic waves in Figure 1 are described as “seasonal,” but it’s unclear whether these represent true seasonality or reflect episodic viral incursions. Given the relatively short timeframe (18–24 months), caution is warranted in interpreting these patterns as seasonal.*

We have altered the title of this section to “Sequences reflect HPAI cases over time” and have altered the text on **lines 75-77** to read “Case detections peaked in the fall and spring, coinciding roughly with seasonal migration timing for birds migrating between North and South

America^{26,27}. Continued monitoring is necessary to determine whether these patterns persist in future years.”

6. Section: “Highly pathogenic H5N1 was introduced multiple times from Europe and Asia” This section is mostly descriptive and could be tightened. Lines 219–222, for example, could be removed or condensed to focus on core findings. While the number of introductions is informative, its significance is not clearly framed in the Introduction, and the implications remain underexplored. Briefly clarifying why this matters, e.g, for understanding barriers to establishment or guiding surveillance priorities, would strengthen the narrative.

We agree, and have cut this section as suggested and significantly tightened up the text in this section of the results.

Discussion

7. The Discussion is generally well-structured and thoughtfully contextualizes the findings. However, some claims overstate the novelty of the work. The global role of wild birds in the dissemination of HPAI has been well established for nearly two decades. What this study adds is the scale and resolution of genomic surveillance during the North American epizootic, and the insight that repeated wild bird–to–poultry spillovers, rather than sustained poultry-to-poultry transmission, characterized the 2021–23 outbreak. This distinction should be more clearly emphasized.

We agree, and have reformulated the discussion to focus on this contribution.

8. Some broad statements, such as “a critical shift in the ecology of highly pathogenic avian influenza viruses in North America”, are difficult to interpret and may be misleading. For example, the 2014–15 clade 2.3.4.4 outbreak did not become established in wild bird populations in the U.S., whereas the current epizootic has persisted across years, suggesting a more durable presence in wild reservoirs. This contrast is a compelling finding and could be framed more clearly. However, broader claims about ecological change should be made with caution, as many aspects of clade 2.3.4.4b's host range and persistence have already been described in prior work.

This is a fair point. In response to this comment and to comments from Reviewer 1, we have reframed the Abstract, Introduction, and Discussion to avoid claiming a critical shift in ecology. We instead focus on the point that the 2022 epizootic in North America was distinct from 2015, and the implications of this. See response to Reviewer 1 above.

9. The practical implications of the findings could be further developed. For instance, do the observed spillover patterns suggest persistent gaps in farm-level biosecurity? Could enhanced real-time wild bird surveillance provide earlier warnings of outbreak risk? Briefly addressing such questions would strengthen the applied value of the study.

We agree that this is an interesting point. We have reformulated parts of the Discussion, and added a few sentences dedicated to the practical value of the work:

Lines 425-439: “Our study collectively supports wild birds as critical sources of the North American H5N1 epizootic. By directly modeling transitions between host groups based on domestic/wild classification, taxonomic order, and migratory behavior, paired with strong dispersal across flyways, we show that wild birds were key drivers of epizootic transmission and introductions into agriculture. These results imply that continuous surveillance in wild birds, particularly Anseriformes⁵⁶, may now be critical for viral tracking and outbreak reconstruction. As the primary source of transmission shifts from poultry to wild migratory birds, the ecology of clade 2.3.4.4b viruses in North America may now follow patterns unfolding globally, where evolution is increasingly governed by wild bird movement, ecology, and reassortment. Recent modeling of HPAI risk in Europe identified *Anatinae* and *Anserinae* Anseriformes prevalence as consistent predictors of HPAI detection⁷², supporting wildlife surveillance for outbreak forecasting and risk assessment. Future work investigating the utility of real-time tracking of wild bird abundance and movement for forecasting outbreaks may be fruitful for formulating new approaches to prevention.”

Lines 455-474: “We find that outbreaks in agriculture were seeded by repeated introductions from wild birds, a pattern that held true regardless of sampling regime, and that aligns with global observations that clade 2.3.4.4b viruses are increasingly spread by wild birds^{63,64}. These findings contrast with the epizootic in 2014/2015, in which a small number of introductions spread efficiently between commercial poultry operations^{2,12}. Because the viruses circulating in 2014/2015 did not establish in local wild bird populations, that epizootic subsided following aggressive culling. Since 2014/2015, biosecurity plans have improved^{12,47} and depopulation occurs more rapidly^{54,12}, potentially contributing to the shorter domestic persistence and limited transmission back to wild birds we observe. Despite these improvements, efficient transmission in wild birds likely allowed for rapid dispersal and continuous outbreak re-seeding, making this epizootic far more challenging to control. US and Canadian policy currently classify H5N1 as a foreign animal disease, meaning that biosecurity to reduce spread between farms and rapid culling^{65,66} are prioritized for outbreak control. Though these control measures will likely remain important, our results suggest that reducing future spillovers into agriculture may now necessitate changes in management priorities. The repeated spillovers we identify suggest that gaps in farm biosecurity remain that could be enhanced to reduce outbreak risk. Finally, layered approaches, including enhanced wild bird monitoring, novel methods to separate wild and domestic birds, and potentially domestic animal vaccination, may necessitate exploration.”

10. The final paragraph continues to largely reiterate established facts about global HPAI risk. This could be shortened or replaced with a clearer summary of the study's main contributions and limitations. e.g., extensive genomic and metadata coverage, limited inference on drivers of transmission, and a relatively short timespan for evaluating true seasonality.

We have reworked the last 2 paragraphs of the Discussion, which now overview limitations and reframe the final paragraph:

Lines 488-516: “Sampling bias is pervasive across viral outbreak datasets, and no modeling approach can completely overcome biases in data acquisition. In the US, only wild Anseriformes are sampled live or hunter harvested, while all other host groups are sampled sick or dead. Detections in domestic birds depend on producer reporting and testing, which likely varies across production types, locations, and premises. To account for this variability we used multiple subsampling approaches, reported results that were consistent, and employed statistical tests to measure the impact of sampling on our results. The titration tests we employ show that the precise number of transitions between wild and domestic birds depends on sampling numbers, providing a clear argument for continuous surveillance in wildlife, and a warning for over-confidence in estimating the transitions between groups. Still, all phylodynamic inferences are limited by sequence data availability, and results could change if future data become available. Our analyses only employ HA sequences, meaning that differences between reassortants could not be compared²⁵. Finally, while we retain data from across North America for all analyses, our results are likely most informative transmission within the United States during the first 6 months of the epizootic.

Taken together, we show that wild birds played the central role in dispersal of the 2021-2023 H5N1 epizootic. Transmission in wild birds provides an explanation for the rapid cross-continental spread and continued agricultural outbreaks despite aggressive culling. Our results highlight the utility of wild bird surveillance for accurately distinguishing hypotheses of epizootic spread, and suggest continuous surveillance as critical for preventing and dissecting future outbreaks. Our data underscore that continued establishment of H5N1 in North American wildlife may necessitate a shift in risk management and mitigation, with interventions focused on reducing risk within the context of enzootic circulation in wild birds. At the time of writing, outbreaks in dairy cattle highlight the critical importance of modeling ecological interactions that drive spillovers between wildlife and domestic production to inform biosecurity, outbreak response, and vaccine strain selection.”

11. Consider citing recent study Signore et al. (*Science Advances*, 2025; <https://www.science.org/doi/10.1126/sciadv.adu4909>), which addresses related questions of clade 2.3.4.4b in North America using genomic data.

We have now cited this paper throughout our manuscript.